

**Comparison of six different soft computing methods in modeling**
**evaporation in different climates**
Lunche Wang[1], Ozgur Kisi[2], Mohammad Zounemat-Kermani[3], Yiqun Gan[4]
[1]Laboratory of Critical Zone Evolution, School of Earth Sciences, China University of
Geosciences, Wuhan 430074, China;
[2]Canik Basari University, Faculty of Architecture and Engineering, Civil Engineering
Department, Samsun, Turkey;
[3]Department of Water Engineering, Shahid Bahonar University of Kerman, Kerman, Iran;
[4]School of Enviromental studies, China University of Geosciences, Wuhan 430074, China;
Corresponding author: Lunche Wang, Laboratory of Critical Zone Evolution, School of Earth
Sciences, China University of Geosciences, Lumo road 388, Hongshan District, Wuhan
430074, China; Tel.: +86 13349889828; E-mail address: wang@cug.edu.cn





**Abstract**: Evaporation plays important roles in regional water resources management, terrestrial ecological process and regional climate change. This study investigated the abilities of six different soft computing methods, Multi-layer perceptron (MLP), generalized regression neural network (GRNN), fuzzy genetic (FG), least square support vector machine (LSSVM), multivariate adaptive regression spline (MARS), adaptive neuro-fuzzy inference systems with grid partition (ANFIS-GP), and two regression methods, multiple linear regression (MLR) and Stephens and Stewart model (SS) in predicting monthly $Ep$. Long-term climatic data at eight stations in different climates, air temperature ($Ta$), solar radiation ($Rg$), sunshine hours ($Hs$), relative humidity ($RH$) and wind speed ($Ws$) during 1961-2000 are used for model development and validation. The first part of applications focused on testing and comparing the model accuracies using different local input combinations. The results showed that the models have different accuracies in different climates and the MLP model performed superior to the other models in predicting monthly $Ep$ at most stations, while GRNN model performed better in Tibetan Plateau. The accuracies of above models ranked as: MLP, GRNN, LSSVM, FG, ANFIS-GP, MARS and MLR. Generalized models were also developed and tested with data of eight stations. The overall results indicated that the soft computing techniques generally performed better than the regression methods, but MLR and SS models can be more preferred at some climatic zones instead of complex nonlinear models, for example, the BJ, CQ and HK stations.

**Keywords**: Pan evaporation; soft computing techniques; regression methods; model comparison

## 1. Introduction

Evaporation is the process of conversion of liquid water to water vapor, which depends on the differences in vapor pressure and air between the surface and surrounding atmosphere





(Penman 1948; Kisi, 2013; Kim et al., 2015). Pan evaporation ($Ep$), which is a major
component of hydrological cycle, plays important roles in scheduling water resources and
designing of irrigation systems. It has been widely used as an index of lake and reservoir
evaporation, potential or reference crop evapotranspiration and irrigation (Snyder 1993).
There are many factors influencing the rates of $Ep$, including solar radiation ($Rg$), air
temperature ($Ta$), relative humidity ($RH$), sunshine hours ($Hs$) and wind speed ($Ws$). The
quantitative effects of different climatic parameters on $Ep$ variations in different regions is
still one of the less understood aspects in the hydrologic cycle. Therefore, proper estimation
and prediction of $Ep$ is of great importance to integrated water resources management and
modeling studies.
The direct measurements of $Ep$ are spatially and temporally limited due to some instrumental
and practical issues (Shirsath and Singh, 2010; Shiri et al., 2014). Many researchers have tried
to estimate the evaporation through indirect methods using climatic variables, for example,
many empirical or semi-empirical equations have been developed for estimating $Ep$ as a
function of meteorological data (Stephens and Stewart 1963; Piri et al., 2009), but some of
these techniques require the data which are often incomplete or not always available for many
locations (Sharda et al., 2008; Majidi et al., 2015). Recently,  the advanced soft computing
techniques (such as artificial neural network, ANN) have been successfully applied for
modeling $Ep$ due to its ability to learn complex and non-linear relationships that are difficult
to model with conventional techniques (Sudheer et al., 2002;  Kisi and Ozturk, 2007;  Kim
and Kim, 2008; Kisi, 2009b; Rahimikhoob, 2009; Guven and Kisi, 2011; Kim et al., 2013;
Goyal et al., 2014; Shiri et al., 2015; Kisi et al., 2016), for example, Kisi (2009a) investigated
the abilities of three different ANN techniques and it was found that the MLP and radial basis
neural network (RBNN) computing techniques could be employed successfully to model the
evaporation process using the available climatic data; Piri et al. (2009) improved the ANN





model by incorporating an autoregressive external input (ARX) component and  evaluated the
models for $Ep$ estimation at a site in hot and dry climate of Southeast Iran. The results showed
that NNARX is better than the ANN and Marciano method, the models with inputs of wind
and vapor pressures performed much better than the ones with temperature and dew point;
Chang et al. (2010) proposed a self-organizing map neural network (SOMN) to assess the
variability of daily evaporation based on meteorological variables, the results demonstrated
that the topological structures of SOMN could give a meaningful map to present the clusters
of meteorological variables and the networks could well estimate the daily evaporation (Kim
et al., 2015). Kim et al. (2012) applied multilayer perceptron-neural networks (MLP),
generalized regression neural networks (GRNN) and support vector machine-neural networks
(SVM) to estimate $Ep$ in temperate and arid climatic zones and the results indicated that these
ANN models performed better than the emprical Linacre model and MLR model. Goyal et al.
(2014) investigated the abilities of ANN, Least Squares Support Vector Machine (LSSVM),
Fuzzy Logic (FG), Adaptive Neuro-Fuzzy Inference System (ANFIS) techniques, Hargreaves
and Samani method (HGS), as well as the Stephens-Stewart (SS) method to improve the
accuracy of daily $Ep$ estimation in sub-tropical climates of India. The results showed that the
above soft computing models outperformed the HGS and SS methods, and the LSSVM and
FG models produced the highest accuracies. Kisi (2015) investigated the accuracy of LSSVM,
multivariate adaptive regression splines (MARS) and M5 Model Tree (M5Tree) in modeling
$Ep$ at Mersin and Antalya stations in Mediterranean region of Turkey, which indicated that the
LSSVM model could be successfully used for estimating $Ep$ using local input and output data
while the MARS model performed better than the LSSVM model in case of without local
input and outputs. Several studies have also been performed in order to compare and assess
the $Ep$ models with limited data around the world (Kisi and Cengiz, 2013; Majidi et al., 2015).
On the contrary, only a few studies have been conducted to find the most appropriate methods



to estimate *Ep*, and most of these studies focused on comparing only two or three models.
Therefore, there is no clear consensus on which methods are better to employ when lacking
important long term measured data such as radiation and heat fluxes. Meanwhile, the *Ep*
models are only tested at few number of stations in literature, for example, Keskin et al. (2004)
only compared the FG model with empirical Penman method at Lake Eğirdir in Turkey;
Sanikhani et al. (2012) compared two different ANFIS models including grid partitioning (GP)
and subtractive clustering (SC), in modeling *Ep* at San Francisco and San Diego in California,
however, there are almost no studies using large number of stations (> 3) for obtaining more
generalized conclusions. In addition, there are not any studies in literature that compare
different methods in estimating *Ep* at different climates (for example, the arid continental
climate, desert climate, semi humid monsoon climate, plateau climate and the tropical
maritime monsoon climate), which provided an impetus for the present investigation for
revealing a more robust and applicable *Ep* estimation model.
Considering the importance of the evaporation in either irrigation management or
hydrological modeling, the aim of this study is to investigate capability and usability of six
different soft computing methods, ANFIS-GP, FG, GRNN, LSSVM, MARS and MLP, and
two regression methods, MLR and SS, in modeling *Ep* using different climatic input
combinations of *Rg, Ta, Hs, RH* and *Ws*. Data from eight stations in different climatic zones are
used for training and testing above models. The model performances will be compared and
discussed through: (i) estimating *Ep* of each station using different local input combinations;
(ii) estimating *Ep* of eight stations using generalized ANFIS-GP, FG, GRNN, LSSVM,
MARS, MLP, MLR and SS models. To the knowledge of the authors, no similar studies have
been reported using above mentioned methods for modeling *Ep*, this will be the first study to
compare the accuracy of multiple soft computing models for *Ep* estimation in different
climates.



## 2. Methods and materials

*2.1. Modeling strategies*

*2.1. 1. Multi-layer perceptron neural network*

Fig. 1

MLP is well-known and efficient neural network widely used for a variety of problems such as classification, time series modeling and regression. MLPs are organized hierarchically networks by several layers, including an input layer, hidden layer(s) and an output layer (Zounemat-Kermani et al., 2013; Wang et al., 2016b). There are one or more hidden layers between the input and output layers which are connected by neurons (including synaptic weights, biases and activation or transfer functions). Each neuron receives its input value(s) from the input vector (or the antecedent hidden layer's output) and then calculates a weighted sum of input values passing through the transfer function, which generates the output of the neuron (Fig. 1a). MLPs are feed-forward networks, using the error back-propagation (BP) algorithm for network training. In the BP algorithm, an iterative process changes the weights and biases of the network to optimize the solution by reducing the overall error between the output and target (generally the observed parameters) values. More details about the MLP model can be found in Kisi (2009b) and Zounemat-Kermani (2012).

*2.1. 2. Generalized regression neural network*

The GRNN model has a parallel structure, but they do not use an iterative process for learning procedure between the input and output variables. The structure of GRNN consists of four consequent layers, namely the 1) input, 2) pattern, 3) summation, and 4) output layers (see Fig.1b for a schematic diagram of a GRNN network). In the first layer, the total number of input variables is equal to the number of input units. Input data are linked to the second layer where each neuron presents a training pattern. The second layer sends processed information



to the third (summation) layer through the pattern neurons. In the summation layer, there are
two types of S-summation and D-summation neurons, which are connected to the pattern
layer unit (Zounemat-Kermani, 2014). The sum of the weighted responses of the second layer
is calculated by the S-summation neurons, while the D-summation neurons compute the un-
weighted outputs. Finally, in the output layer, the division of the output of each S-summation
neuron by D-summation neuron gives the output value. More descriptions about the GRNN
model can be seen from Cigizoglu and Alp (2006).
*2.1. 3. Grid partitioning adaptive neuro-fuzzy inference system*

Fig. 2

ANFIS refers to a multi-layer adaptive network combined with neural network analogy with
the fuzzy inference system. It consists of five consecutive layers (fuzzification, product,
normalization, de-fuzzification and output) in an implementation procedure of different node
functions to learn and adjust the parameters in a fuzzy inference system (Fig. 2). A hybrid
learning algorithm, including forward and backward passes is utilized for reducing calculated
errors and training phase. With the calculation of the least squared error, the consequent
parameters are updated, whereas, the premise parameters are fixed. Hence, in the backward
pass the consequent parameters are fixed and the premise parameters are updated through the
gradient descent algorithm (Kisi and Ozturk, 2007; Zounemat-Kermani and Teshnehlab,
2008; Khayam et al., 2012).
Membership functions and fuzzy inference parameters are identified according to the
adjustment of premise and consequent parameters by an iterative process of the forward and
backward passes. Construction of ANFIS models is based on the partitioning of the input-
output data for establishing the rule base system. In this respect, various approaches such as
ANFIS-GP, subtractive clustering and fuzzy c-means methods can be applied.





In this study the Sugeno model, which is the most commonly used system, along with the grid
partitioning method is applied for modelling evapotranspiration. The reader can find out more
details about ANFIS-GP in several available publications (Jang, 1993; Terzi et al., 2006;
Khayam et al., 2012).
*2.1. 4. Fuzzy-genetic algorithm*
The hybrid FG algorithm combines a meta-heuristic algorithm (genetic algorithm) and an
adaptive fuzzy inference system (AFIS). In the AFIS, the input vectors along with the
corresponding output(s) are introduced to the fuzzy system which is established based on the
fuzzy logic approach. For further information about fuzzy logic, the reader is referred to
related reports (Zounemat-Kermani and Scholz, 2013).
Genetic algorithms (GAs) are stochastic search algorithm based on the mechanics of natural
genetics and natural selection which can be used for optimization problems. Getting the
advantage of using the evolutionary mechanism, they are capable of searching large solution
spaces efficiently. GA is composed of three main stages, namely, population initialization,
GA operators (reproduction, crossover, and mutation) and evaluation (Kisi and Tombul,
2013). In this study, the proposed hybrid FG model is based on a model wherein the
membership functions' parameters (e.g. center and width of Gaussian MFs) are optimized
using a GA algorithm. The objective function of the genetic algorithm optimizer is the
minimization of the error criterion (e.g. RMSE) of prediction made by an AFIS model (Kisi,
2009; Ganjidoost et al., 2015).
*2.1. 5. Least-Squares support vector machine*
The SVM is based on a statistical learning theory which projects the input data classes to a
higher dimensional feature space. The aim of the SVM algorithm is searching for an optimum
hyper-plane with the minimum distance to the observed values. The algorithm is efficient,





quick and converging procedure to the global optimum (Mesbah et al., 2015; Lu et al., 2016).
However, the SVM algorithm has been modified and improved, referring to the Least-
Squares-SVM (LSSVM) (Suykens and Vandewalle, 1999). In addition to have all the merits
of the original SVM, LSSVM has become simpler and more rapid. This issue is caused by the
structure of the LSSVM algorithm which solves a group of linear equations instead of solving
a quadratic programming problem in the SVM method. LSSVM gets the advantage of the
applying equality constraints (in exchange for traditional inequality constraints of SVM) and
implements the sum of squared regression errors in the training process. Further details about
the main equations and complete explanations of this subject can be found in Suykens and
Vandewalle (1999) and Kisi (2015).
*2.1. 6. Multivariate adaptive regression splines*
MARS, introduced by Friedman (1991), is categorized as a nonparametric regression method.
MARS divides the space of each independent variable into split various regions called sub-
regions. For each sub-region a unique mathematical regression equation is defined. A
relationship is developed for each sub-region of the independent variable to the output
(response) of the system based on the attained mathematical equation. This whole process is
conducted by a stepwise procedure consists of backward and forward steps. In the forward
step, a set of appropriate input variables is selected and split. However, split process in the
forward step might generate an over-fitted complex model resulting in poor performance.
Thereafter, in the backward step unnecessary variables will be eliminated (Adamowski et al.,
2012; Kisi, 2015). For more detailed information on the development of the MARS models
used throughout in this research the reader can refer to Shardaet al. (2008) and Kisi (2015).
*2.1. 7. Multiple Linear Regressions*



MLR is a technique utilized to model the linear relationship between a dependent variable and
one or more independent variables. The dependent variable is sometimes additionally called
the predictors. MLR is depends on least squares: the model is fit such that the sum of squares
of     differences     of     estimated     and     observed     values     is     minimized
(http://tree.ltrr.arizona.edu/webhome/dmeko/geos585a.html#cLesson11).  MLR  is  probably
the most widely used method in hydrology and climatology for developing models to
reconstruct or analysis the long-term variations of climatic factors in literature. In this study,
MLR models are developed using the same data set which was used to train and test the above
soft computing models.

*2.1. 8. Stephens and Stewart Model*

Stephens and Stewart (SS) model is an empirical linear equation and requires only radiation
and temperature data (Stephens and Stewart 1963). This model was reported as the best
among 23 models (Al-Shalan and Salih, 1987; Sudheer et al., 2002; Shirsath and Singh, 2010),
which can be expressed as $Ep = Rg$ (a + b×$Ta$), where *a* and *b* are fitting constants
(determined on the training data through least square fitting).

*2.2. Case study and data*

Fig. 3

In this study, monthly climatic data at eight stations of China Meteorological Administration
(CMA) were used for developing and testing *Ep* models in different climates. Fig.3 shows the
detailed geographical locations of above stations, which are named as HEB (latitude 45°45′ N,
longitude 126°46′E,  142.3 masl (m above sea level), ALT (47°44′ N, 188° 05′E, 735.3 masl),
MQ (38°38′ N, 103°05′E,  1367 masl), BJ (39°48′ N, 116°28′E, 31.3 masl), LSA (29°40′ N,
91°08′E, 3648.7 masl), CQ (29°35′ N, 106°28′E, 259.1 masl), HZ (30°14′ N, 120°10′E, 41.7
masl) and HK (20°02′ N, 110°21′E, 13.9 masl). It should be noted that above stations are



243 located at  different climatic zones, for example, the HEB station is in the Northeast China

244 with long and cold winter (semi humid temperate continental climate); the ALT station is in

245 the Northwest China with arid continental climate; the MQ station is surrounded by the

246 Tengger and Badan Jilin desert, which is characterized by continental desert climate with hot

247 summer and cold winter, enough light and little rainfall amount; BJ is characterized by typical

248 north temperate semi humid continental monsoon climate; LSA is in the zone of semi-arid

249 plateau climate, which is called the sunlight city due to the sufficient sunshine resources in

250 Tibetan Plateau; CQ is characterized by subtropical monsoon humid climate with more

251 cloudy and foggy conditions; the HZ station is known as one of China's "four ovens" cities

252 where summertime temperatures can reach to $40\,^{o}C$, which is characterized by hot and rainy

253 in summer, cold and dry in winter due to the effects of East Asian atmospheric circulation, the

254 terrain of Qinghai-Tibet Plateau and the North Pacific Ocean; the HK station is located at the

255 northern margin of the low latitude tropics, which belongs to the tropical maritime monsoon

256 climate. The detailed information about the geographical, climatic and hydrological

257 conditions in this region can also be seen in Zhai et al. (1999) and Ding et al. (2006).

258            Fig. 4

259 The data used in this research cover 40 years (1961-2000) of monthly records of air

260 temperature ($Ta$), solar radiation ($Rg$), sunshine durations ($Hs$), relative humidity ($RH$), wind

261 speed ($Ws$) and pan evaporation ($Ep$). For each station, 50% of the whole data were randomly

262 chosen for training the $Ep$ models and the remaining data were used for testing the models.

263 The annual variations of $Ep$ and associated climatic factors are shown clearly in Fig. 4, it is

264 clear that $Ep$ at MQ and LSA are generally higher than those at other stations, there are

265 decreasing trends of $Ep$ for ALT, BJ and HK stations during 1960-2000, and the most

266 significant increasing trends of $Ep$ are observed for LSA station. The $Hs$ at CQ and HZ

267 stations are much lower than those at other stations and the annual mean $Hs$ is the highest





among the eight stations. There are also slight increasing trends for *Hs* at most stations except
BJ, HK and HZ stations. The annual mean *Rg* is obviously higher at LSA station and *Rg* at
CQ is the lowest, the *Rg* generally decreased from 1961 to 1990 and then increased for most
stations. The *RH* is generally larger than 75% at HZ, CQ and HK stations and lower than 50%
for MQ and LSA stations. There are distinct differences for annual mean *Ta* at above eight
stations, for example, *Ta* at HK is generally higher than 23 $^{\circ}$C, while the highest annual mean
*Ta* at HEB is lower than 5 $^{\circ}$C. However, *Ws* is highest at HEB and lowest at CQ station, and
*Ws* is decreasing from 1960s to 2000s at most stations.

Table 1

Fig. 5

Table 1 showed the monthly statistics of the climatic parameters, $x_{mean}$, $S_x$, $C_v$, $C_x$, $x_{min}$ and
$x_{max}$ denote the mean, standard deviation, variation coefficient, skewness, minimum and
maximum values, respectively. It is clear that the monthly mean *Ep* is 4.35, 4.72, 7.26, 5.09,
6.35, 2.86, 3.65 and 5.00 mm for station HEB, ALT, MQ, BJ, LSA, CQ, HZ and HK,
respectively. The mean *Rg* at LSA, MQ and ALT (20.41, 16.41 and 15.13MJ m$^{-2}$) are higher
than those at other stations; the mean *Ta* at HK station is 24.08$^{\circ}$C, which is highest among all
the stations. The *Hs* shows low variations for the MQ, BJ and LSA stations (see Cv values in
Table1) and the monthly mean *Hs* at CQ (2.83 hour) is much lower than the other station. The
monthly *RH* is 65.44%, 57.99%, 44.82%, 57.29%, 44.39%, 79.15%, 78.04% and 84.14% for
HEB, ALT, MQ, BJ, LSA, CQ, HZ and HK, respectively, which indicates that *RH* is
generally higher at lower latitudes. The *Ws* at HEB station is higher than other station in each
month and the lowest monthly *Ws* (1.36 m s$^{-1}$) is observed at CQ station. The monthly *Ep*, *Ta*,
*Hs* and *Rg* are generally higher in summer and lower in winter months (Fig.5); the *RH* is also
lower in spring months for some stations such as HEB and MQ; the *Ws* is higher at spring and
lower in summer months for most stations (Fig.5). For the HEB station, *RG* shows low





skewed distribution and has a relatively higher correlation with $Ep$ ($R$ =0.89); the $RH$ and $Ws$
data have the lowest ($R$ = -0.30, 0.26) correlation with $Ep$. For the ALT station, the $Ta$ data
have a high skewness ($C_v$ =3.07) and high correlation with $Ep$ ($R$ =0.93). For the MQ station,
the $Hs$ shows the lower skewed distribution ($C_v$ =0.13) and has a positive correlation with $Ep$
($R$ = 0.72); the $Ta$ data have a higher skewness ($C_v$ =1.36) and the highest correlation with $Ep$
($R$ =0.93). In similar, the $Rg$ and $Hs$ data show relatively higher skewness and correlations
with $Ep$ for the BJ, CQ and HK stations. At some cases, the correlations between $Ta$ and $Ep$
are also higher than those with $Hs$, for example, the LSA ($R$ =0.75) and HZ ($R$ =0.88) stations.
It is clear from the statistical indices in Table 1 that each climatic variable have different
correlations with $Ep$, and $Rg$, $Hs$ and $Ta$ variables seem to be the most effective parameters
for predicting $Ep$ with respect to the correlations.
*2.3. Evaluation criteria*
In this study, the ANFIS-GP, FG, GRNN, LSSVM, MARS, MLP, MLR and SS models were
evaluated and compared with each other utilizing the mean absolute errors (MAE), root mean
square errors (RMSE) and determination coefficient ($R^2$), which can be expressed as
$$RMSE = \sqrt{\frac{1}{N} \sum_{i=1}^{N} \left( Ep_{m,i} - Ep_{o,i} \right)^2} \qquad (1)$$

$$MAE = \frac{1}{N} \sum_{i=1}^{n} \left| Ep_{m,i} - Ep_{o,i} \right| \qquad (2)$$

$$R^2 = \frac{\left( \sum_{i=1}^{n} \left( Ep_{m,i} - \overline{Ep_m} \right)\left( Ep_{o,i} - \overline{Ep_o} \right) \right)^2}{\sum_{i=1}^{n} \left( Ep_{m,i} - \overline{Ep_m} \right)^2 \sum_{i=1}^{n} \left( Ep_{o,i} - \overline{Ep_o} \right)^2} \qquad (3)$$

where $N$ and bar respectively indicate the number of data and mean of the variable, $Ep_m$ and
$Ep_o$ are the modeled and observed pan evaporation.





**3.  Results and discussion**
This study compares six different soft computing methods, ANFIS-GP, FG, GRNN, LSSVM,
MARS and MLP, and two empirical methods, MLR and SS, in modeling $Ep$ using climatic
inputs of $Rg, Ta, Hs, RH$ and $Ws$. Data from eight stations, HEB, ALT, MQ, BJ, LSA, CQ, HZ
and HK were utilized in the applications. The input combinations used for each model are
provided in Table 2, the numbers after each model indicates the input combination. Two
Gaussian membership functions were utilized for each ANFIS-GP and FG model. Different
regularization constants and RBF kernel widths were tried for the LSSVM models and the
optimal models that provided the least RMSE error in testing stage were obtained. For the
GRNN models, different spread constants were tried. Different hidden node numbers were
tried and the optimal ones were obtained for each MLP model.

Table 2

Table 3

Table 4

The training and testing results of the ANFIS-GP, FG, GRNN, LSSVM, MARS, MLP, SS
and MLR models in predicting $Ep$ of HEB station are shown in Table 3. It is clear from the
table that the models with full weather data ($Rg, Ta, Hs, RH$ and $Ws$) have the best accuracy.
The MLP7 model performs superior to the other models in predicting $Ep$ at HEB. The
accuracy ranks of the applied soft computing models in testing period are: MLP, ANFIS-GP,
FG, GRNN, MARS, LSSVM and MLR. It is clear from the first three input combinations that
there is a slight difference between $Rg$ and $Ta$ and they are much more effective on modeling
$Ep$ at HEB station than the other variables. This is also confirmed by the $R^2$ values given in
Table 1. Comparisons of the simple two-input ($Rg$ and $Ta$) models clearly indicate that the
ANFIS-GP4, GRNN4 and LSSVM4 models have better accuracy than those of the SS model
while the FG4, MARS4 and MLP4 models give inferior results in testing period. Table 4





gives the accuracy of the applied models in predicting *Ep* at ALT station. Similar to the HEB
station, the models comprising whole weather inputs generally provide the best accuracy and
the optimal MLP7 model outperforms the other models in predicting *Ep* at ALT station. The
accuracy ranks of the models in testing stage are: MLP, LSSVM, ANFIS-GP, FG, MARS,
GRNN and MLR. There is a slight difference between the *Rg, Ta* and *Hs* parameters and
these are also parallel to the correlations given in Table 1. Two-input soft computing models
seem to have a better accuracy than the SS model in predicting *Ep* at ALT station in the
testing stage.

Table 5

Table 6

The training and testing statistics of the soft computing models, SS and MLR in predicting *Ep*
of MQ station are provided in Table 5. In this station, five-input models also have the best
performance and the MLP7 model performs superior to the other models. The accuracy ranks
of the 5-input models are: MLP, FG, ANFIS-GP, GRNN, MARS, MLR and LSSVM. From
the first three inputs, it is clear that the *Rg* and *Ta* variables have more effects on *Ep* than the
*Hs* in MQ station. The correlations in Table 1 also confirm these results. It is apparent from
Table 5 that the SS model provides inferior results in comparison with the 2-input soft
computing models at MQ station. Table 6 reports the training and testing results of the applied
models in predicting *Ep* at BJ station. From the table, it is obvious that the models with full
weather data generally have the best accuracy. The MLP model provides better performance
than the other models with respect to MAE, RMSE and $R^2$. The ranks of the applied models in
testing accuracy are: MLP, LSSVM, GRNN, ANFIS-GP, FG, MARS and MLR. It is clear
from the first three input combinations that the *Rg* which has a higher correlation with *Ep* (see
Table 1) is much more effective on *Ep* than the *Ta* and *Hs* at BJ. Simple SS model seems to





have better accuracy than the applied 2-input soft computing models in predicting *Ep* at BJ
station in testing stage.

Table 7

Table 8

Table 7 compares the accuracy of the applied soft computing models in predicting *Ep* at LSA
station. Similar to the previous stations, the best accuracies were generally obtained from five-
input models and the GRNN model performs better than the other models with respect to
MAE and RMSE statistics. The accuracies of the applied models in testing stage rank as:
GRNN, MLP, LSSVM, MARS, FG, ANFIS-GP and MLR. The *Ta* seems to be the most
effective parameter in predicting *Ep* at this station, which is also confirmed by the high
correlation between *Ta* and *Ep* given in Table 1. The models with *Hs* input generally provide
worse results than those comprising *Rg* input parameter. Two-input LSSVM4, ANFIS-GP4
and SS models have similar accuracy and they perform inferior to the FG4, GRNN4, MARS4
and MLP4 models. The accuracies of the ANFIS-GP, FG, GRNN, LSSVM, MARS, MLP, SS
and MLR models in both training and testing stages are given in Table 8 for predicting *Ep* of
CQ station. Unlike the previous stations, four-input models generally provide the best
performance in this station. This implies that adding *Ws* input generally decreases the model
accuracies even though it does not have a low correlation (R=0.58) with *Ep* at CQ station. The
MLP6 and GRNN6 models have similar accuracies and they perform superior to the other
models. The accuracy ranks of the above applied models in testing stage are: MLP, GRNN,
FG, LSSVM, MARS, ANFIS-GP and MLR. Similar to the BJ station, the *Rg* input seems to
have more effects on *Ep* than the *Ta* and *Hs* input at CQ station than the other variables even
though the *Hs* has a higher correlation with *Ep*. Comparison of two-input models obviously
shows that the SS model has a better accuracy than those of the other two-input soft
computing models in testing period.



Table 9

Table 10

The training and testing accuracy of the soft computing models, SS and MLR in predicting *Ep*
of HZ station are provided in Table 9. In this station, four- and five-input models also have
the best accuracies. The optimal MLP and GRNN models have similar performance and they
perform superior to the other models in predicting *Ep* at HZ station. The performance ranks of
the optimal models are: MLP, GRNN, MARS, FG, LSSVM, ANFIS-GP and MLR. Similar to
the BJ station, the *Rg* variable which has a higher correlation with *Ep* (see Table 1) is much
more effective on estimating *Ep* than the *Ta* and *Hs* at HZ. *Ta* variable also provides better
accuracy than the *Hs* in predicting *Ep*. Comparisons of simple two-input models clearly
indicates that the MARS4 and MLP4 models have better accuracy than those of the SS model
while the ANFIS-GP4, FG4, GRNN4 and LSSVM4 provide inferior results in testing stage.
Table 10 compares the accuracy of the models in predicting *Ep* at HK station. From the table,
it is clear that the best accuracies were obtained from five-input models in predicting *Ep*. The
MLP model performs superior to the other models with respect to MAE, RMSE and $R^2$
statistics. The accuracies of the applied models in testing period rank as: MLP, MLR, GRNN,
MARS, LSSVM, ANFIS-GP and FG. The accuracies of the MLR and MLP model are close
to each other. Therefore, simple MLR model can be preferred instead of more complex soft
computing models in predicting *Ep* at HK station. Unlike the previous stations, the models
comprising *Hs* input provide better accuracy than those which use only *Rg* or *Ta* input. The
*Rg* variable also seems to be more effective on *Ep* than the *Ta* at HZ station. The difference
among the two-input models is very small and the GRNN4 and SS models perform slightly
better than the other two-input models in predicting *Ep* at HK station.
It can be seen from above analysis that adding *RH* or *Ws* inputs into the applied models
generally increased their accuracies in predicting *Ep* in all stations even though these




parameters had the lowest correlation with *Ep* (see Table 1). This indicates the non-linear
relationship between *RH* (*Ws*) and *Ep* and linear $R^2$ indicator cannot show this phenomenon.
General accuracies of the applied models are compared in Table 11. It is obvious that the
MLP model provides much better scores than the other methods in predicting *Ep* at above
eight stations (and data from all station) and the final accuracy ranks of the above models are:
MLP, GRNN, LSSVM, FG, ANFIS-GP, MARS and MLR. In some stations (e.g., BJ, CQ),
simple SS model performed superior to the two-input soft computing models and it can be
preferred in these stations where *Hs, RH* and *Ws* parameters are not available.
Table 11
Figs.6-13 illustrates the estimates of the optimal models in testing phase for eight stations in
the form of scatterplot. For HEP station, the fit line of the MARS model seems to be closer to
the ideal line (y=x) while the MLP model has the highest $R^2$ means less scattered estimates
than the other models. All the soft computing models provide close estimates to the
corresponding observed ones in ALT station while MLR generally tends to overestimation.
For MQ station, the LSSVM, MARS and MLR models provide more scattered estimates than
the ANN (MLP and GRNN) and fuzzy based ANFIS-GP and FG models. All the models
generally have good estimates at the BJ, CQ, HK and HZ stations. In LSA station, the MLP,
GRNN, LSSVM and MARS provide less scattered estimates than the fuzzy based ANFIS-GP,
FG and MLR models. From Figs.6-13, it is clear that the MLP model generally provided less
scattered estimates than the other models in all stations. The models generally provided the
worst accuracy in LSA station. One of the main reasons of this may be the fact that the *Ep* has
low correlations with the climatic input data at LSA in comparison with other stations. It is
clear from Fig. 12 that the SS model provides less scattered estimates for the ALT, CQ and
HZ in comparison with other stations.
Fig. 6





Fig. 7

Fig. 8

Fig. 9

Fig. 10

Fig. 11

Fig. 12

Fig. 13

The *Ep* data at all stations are further estimated using single generalized MLP, GRNN,
MARS, LSSVM, ANFIS-GP, FG, MLR and SS models. The optimal models are obtained
using training and testing data of above eight stations, the training and testing statistics of the
applied models are compared in Table 12. Similar to the previous results, the best accuracies
are obtained from the five-input models and the MLP model performs better than the other
models. The accuracies of the applied models in testing stage rank as: MLP, GRNN, LSSVM,
FG, ANFIS-GP, MARS and MLR. The *Rg* variable seems to be the most effective parameter
in predicting *Ep* for data from all station, and the models with *Hs* input generally provides
slightly better results than those comprising *Ta* input parameter. Two-input MARS4 and
ANFIS-GP4 models perform inferior to the FG4, GRNN4, LSSVM4 and MLP4 models; the
SS model has a lower accuracy than those of the other two-input soft computing models in
testing period (Table 12). It is also observed that only *Ta* or *Hs* input seems to be insufficient
for obtaining an accurate generalized *Ep* model and the model performances generally
increase with input numbers, which implies that all above climatic parameters have positive
effects on estimating *Ep* for most stations in different climates. The estimates of the
generalized models are illustrated in Figs.6-13 (see the last scatterplot in each figure). It is
clearly observed from the figures that the generalization significantly decreases models
accuracy in estimating *Ep* at all stations. However, all the soft computing models generally





have good generalization ability. Some underestimations of the high Ep values are clearly
seen for the generalized models. Different data ranges in training and test stages may be the
reason of this.

465                      Table 12


**4. Conclusion**
This study investigated and compared the abilities of six different soft computing techniques,
MLP, GRNN, LSSVM, FG, ANFIS-GP, MARS, and two regression methods, MLR and SS,
in modeling *Ep* using different climatic input combinations of *Rg, Ta, Hs, RH* and *Ws*. The
climatic data obtained from eight stations in different climatic zones were used as inputs for
training and testing above models. In the first part of applications, the models with different
local input combinations were compared with each other in estimating *Ep* at each station,
separately. The results showed that the models with more inputs generally have better
accuracies and the MLP model performed superior to the other models in predicting monthly
*Ep* at most stations, however, the GRNN model performed better than the other models at
LSA station with respect to MAE and RMSE statistics. The *Rg* and *Ta* variables are more
effective on modeling *Ep* at most stations, while *Ta* seems to be the most important parameter
in predicting *Ep* at LSA, and adding *Ws* to the input combinations even decreases the model
accuracies. Sometimes, MLR model can be used for predicting *Ep* in tropic climate instead of
more complex soft computing models, and SS model can also be adopted for some stations in
regions of subtropical humid climate or temperate continental climate. The second part of
applications focused on estimating *Ep* of all stations using generalized models, which could
be successfully used for predicting *Ep* using different input combinations. The accuracies of
the applied models rank as: MLP, GRNN, LSSVM, FG, ANFIS-GP, MARS and MLR. The



*Rg* and *Hs* variables seem to be the most effective parameters in predicting *Ep* for data from
all stations.
In summary, it is revealed in this study that the MLP models are the most appropriate for
predicting *Ep* using limited climatic inputs in different climates. The present applications can
be practically adopted in the field of water resources management for accurately mapping
regional and global distributions of evaporation and related water resource storages.
**Acknowledgement**
This work was financially supported by the Special Fund for Basic Scientific Research of
Central Colleges, China University of Geosciences, Wuhan (No.CUG150631), and the 111
Project (grant No. B08030). We would like to thank the China Meteorological Administration
(CMA) for providing the meteorological and radiation data.

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





Table 1. Monthly statistical parameters of each data set for each station

| Station | Dataset | $x_{mean}$ | $S_x$ | $C_v$ | $C_x$ | $x_{min}$ | $x_{max}$ | $R$ | |
|---|---|---|---|---|---|---|---|---|---|
| HEB | *Rg* | 12.98 | 5.35 | 0.41 | 0.00 | 3.68 | 28.71 | 0.89 | |
| | *Ta* | 4.17 | 14.52 | 3.48 | -0.25 | -24.71 | 25.25 | 0.86 | |
| | *Hs* | 7.02 | 1.59 | 0.23 | -0.25 | 2.82 | 10.89 | 0.79 | |
| | *RH* | 65.44 | 11.01 | 0.17 | -0.44 | 36.23 | 85.06 | -0.36 | |
| | *Ws* | 3.69 | 0.97 | 0.26 | 0.61 | 1.88 | 6.69 | 0.26 | |
| | *Ep* | 4.35 | 3.27 | 0.75 | 0.44 | 0.16 | 12.96 | 1 | |
| ALT | *Rg* | 15.13 | 7.21 | 0.48 | -0.06 | 2.34 | 27.69 | 0.92 | |
| | *Ta* | 4.54 | 13.95 | 3.07 | -0.25 | -25.08 | 24.87 | 0.93 | |
| | *Hs* | 8.2 | 2.52 | 0.31 | -0.25 | 1.92 | 12.66 | 0.90 | |
| | *RH* | 57.99 | 13.41 | 0.23 | 0 | 30.1 | 86.77 | -0.89 | |
| | *Ws* | 2.40 | 0.99 | 0.41 | 0.05 | 0.31 | 5.46 | 0.69 | |
| | *Ep* | 4.72 | 3.84 | 0.81 | 0.33 | 0.15 | 13.79 | 1 | |
| MQ | *Rg* | 16.41 | 4.98 | 0.30 | 0.07 | 7.21 | 26.9 | 0.92 | |
| | *Ta* | 8.33 | 11.32 | 1.36 | -0.19 | -15.46 | 25.72 | 0.93 | |
| | *Hs* | 8.37 | 1.12 | 0.13 | 0.30 | 5.47 | 11.38 | 0.72 | |
| | *RH* | 44.82 | 9.06 | 0.2 | 0.12 | 24.3 | 74.58 | -0.29 | |
| | *Ws* | 2.68 | 0.55 | 0.20 | 0.08 | 1.23 | 4.32 | 0.55 | |
| | *Ep* | 7.26 | 4.45 | 0.61 | 0.10 | 0.42 | 15.89 | 1 | |
| BJ | *Rg* | 14.61 | 4.94 | 0.34 | 0.05 | 5.14 | 25.59 | 0.91 | |
| | *Ta* | 12.20 | 10.74 | 0.88 | -0.17 | -7.6 | 29.56 | 0.75 | |
| | *Hs* | 7.41 | 1.42 | 0.19 | 0.06 | 3.79 | 11.21 | 0.76 | |
| | *RH* | 57.29 | 13.70 | 0.24 | 0.02 | 21.86 | 85.52 | 0.09 | |
| | *Ws* | 2.50 | 0.67 | 0.27 | 0.49 | 1.07 | 4.65 | 0.14 | |
| | *Ep* | 5.09 | 2.83 | 0.56 | 0.70 | 0.85 | 15.63 | 1 | |
| LSA | *Rg* | 20.41 | 4.20 | 0.21 | 0.11 | 10.39 | 30.69 | 0.68 | |
| | *Ta* | 7.82 | 6.37 | 0.81 | -0.21 | -5.16 | 18.19 | 0.75 | |
| | *Hs* | 8.19 | 0.96 | 0.12 | -0.59 | 4.66 | 10.55 | 0.18 | |
| | *RH* | 44.39 | 15.10 | 0.34 | 0.30 | 15.36 | 76.61 | 0.19 | |
| | *Ws* | 1.90 | 0.46 | 0.24 | 0.30 | 0.92 | 3.41 | 0.34 | |
| | *Ep* | 6.35 | 2.23 | 0.35 | 0.36 | 2.15 | 13.28 | 1 | |
| CQ | *Rg* | 8.80 | 4.69 | 0.53 | 0.43 | 0 | 21.32 | 0.92 | |
| | *Ta* | 17.93 | 7.46 | 0.42 | -0.10 | 0.64 | 30.90 | 0.85 | |
| | *Hs* | 2.83 | 2.02 | 0.71 | 0.91 | 0 | 9.19 | 0.94 | |
| | *RH* | 79.15 | 8.55 | 0.11 | -4.66 | 6.97 | 90.30 | -0.40 | |
| | *Ws* | 1.36 | 0.34 | 0.25 | -0.12 | 0.64 | 2.13 | 0.58 | |
| | *Ep* | 2.86 | 1.94 | 0.68 | 0.87 | 0.54 | 9.32 | 1 | |
| HZ | *Rg* | 11.63 | 4.20 | 0.36 | 0.54 | 3.93 | 24.83 | 0.94 | |
| | *Ta* | 16.45 | 8.46 | 0.51 | -0.06 | -0.01 | 31.03 | 0.88 | |
| | *Hs* | 4.99 | 1.74 | 0.35 | 0.63 | 1.19 | 11.25 | 0.80 | |
| | *RH* | 78.04 | 5.63 | 0.07 | -0.80 | 53.74 | 90.42 | -0.04 | |
| | *Ws* | 2.24 | 0.43 | 0.19 | 0.05 | 1.01 | 3.58 | 0.13 | |
| | *Ep* | 3.65 | 1.94 | 0.53 | 0.84 | 0.74 | 11.33 | 1 | |
| HK | *Rg* | 13.86 | 4.33 | 0.31 | -0.05 | 4.06 | 24.34 | 0.90 | |
| | *Ta* | 24.08 | 4.07 | 0.17 | -0.55 | 13.21 | 29.83 | 0.81 | |



| | | | | | | |
|---|---|---|---|---|---|---|
| **Hs** | 5.83 | 1.96 | 0.34 | -0.26 | 0.47 | 9.94 | 0.89 |
| **RH** | 84.14 | 3.61 | 0.04 | -0.52 | 71.39 | 94.46 | -0.41 |
| **Ws** | 2.65 | 0.66 | 0.25 | 0.61 | 1.33 | 4.98 | 0.04 |
| **$E_P$** | 5.00 | 1.59 | 0.32 | 0.08 | 1.37 | 9.97 | 1 |

The unit of *Rg, Ta, Pa, Ws* and $E_P$ are MJ m$^{-2}$, ℃,hPa,ms$^{-1}$ and mm, respectively; $x_{mean}$, $S_x$, $C_v$, $C_x$, $x_{min}$ and
$x_{max}$ denote the mean, standard deviation, variation coefficient, skewness, minimum and maximum values,
respectively.
Table 2.The input combinations for different artificial intelligence techniques.

| Models | | | | | | Input combinations |
|---|---|---|---|---|---|---|
| ANFIS-GP | FG | GRNN | LSSV | MARS | MLP | |
| ANFIS-GP1 | FG1 | GRNN1 | LSSV1 | MARS1 | MLP1 | *Rg* |
| ANFIS-GP2 | FG2 | GRNN2 | LSSV2 | MARS2 | MLP2 | *Ta* |
| ANFIS-GP3 | FG3 | GRNN3 | LSSV3 | MARS3 | MLP3 | *Hs* |
| ANFIS-GP4 | FG4 | GRNN4 | LSSV4 | MARS4 | MLP4 | *Rg, Ta* |
| ANFIS-GP5 | FG5 | GRNN5 | LSSV5 | MARS5 | MLP5 | *Rg, Ta, Hs* |
| ANFIS-GP6 | FG6 | GRNN6 | LSSV6 | MARS6 | MLP6 | *Rg, Ta, Hs, RH* |
| ANFIS-GP7 | FG7 | GRNN7 | LSSV7 | MARS7 | MLP7 | *Rg, Ta, Hs, RH, Ws* |






Table 3. Comparisons of different models for predicting *Ep* at HEB.

| HEB | MAE | RMSE | $R^2$ | MAE | RMSE | $R^2$ |
|---|---|---|---|---|---|---|
| ANFIS-GP1 | 1.062 | 1.411 | 0.815 | 1.044 | 1.431 | 0.819 |
| ANFIS-GP2 | 1.226 | 1.68 | 0.737 | 1.082 | 1.471 | 0.797 |
| ANFIS-GP3 | 1.589 | 2.05 | 0.609 | 1.496 | 1.834 | 0.726 |
| ANFIS-GP4 | 0.865 | 1.225 | 0.86 | 0.781 | 1.089 | 0.894 |
| ANFIS-GP5 | 0.785 | 1.167 | 0.873 | 0.645 | 0.907 | 0.923 |
| ANFIS-GP6 | 0.429 | 0.601 | 0.966 | 0.517 | 0.751 | 0.956 |
| ANFIS-GP7 | 0.378 | 0.521 | 0.975 | 0.431 | 0.6 | 0.967 |
| FG1 | 1.031 | 1.371 | 0.825 | 1.031 | 1.507 | 0.816 |
| FG2 | 1.151 | 1.632 | 0.752 | 1.077 | 1.502 | 0.786 |
| FG3 | 1.528 | 2.008 | 0.625 | 1.354 | 1.798 | 0.74 |
| FG4 | 0.719 | 1.071 | 0.893 | 0.688 | 1.178 | 0.891 |
| FG5 | 0.67 | 1.002 | 0.907 | 0.673 | 1.059 | 0.897 |
| FG6 | 0.39 | 0.56 | 0.971 | 0.474 | 0.69 | 0.961 |
| FG7 | 0.305 | 0.421 | 0.983 | 0.435 | 0.661 | 0.959 |
| GRNN1 | 1.057 | 1.428 | 0.819 | 1.039 | 1.403 | 0.817 |
| GRNN2 | 1.155 | 1.632 | 0.753 | 1.057 | 1.475 | 0.796 |
| GRNN3 | 1.519 | 2 | 0.628 | 1.379 | 1.816 | 0.738 |
| GRNN4 | 0.729 | 1.089 | 0.892 | 0.733 | 1.116 | 0.886 |
| GRNN5 | 0.703 | 1.042 | 0.901 | 0.652 | 0.988 | 0.908 |
| GRNN6 | 0.405 | 0.579 | 0.97 | 0.492 | 0.785 | 0.943 |
| GRNN7 | 0.343 | 0.483 | 0.979 | 0.499 | 0.745 | 0.951 |
| LSSV1 | 1.035 | 1.375 | 0.824 | 1.025 | 1.426 | 0.82 |
| LSSV2 | 1.131 | 1.619 | 0.756 | 1.062 | 1.491 | 0.79 |
| LSSV3 | 1.685 | 2.135 | 0.604 | 1.557 | 1.951 | 0.712 |
| LSSV4 | 0.675 | 1.007 | 0.906 | 0.703 | 1.099 | 0.89 |
| LSSV5 | 0.901 | 1.267 | 0.853 | 0.761 | 1.031 | 0.9 |
| LSSV6 | 0.556 | 0.812 | 0.941 | 0.591 | 0.864 | 0.934 |
| LSSV7 | 0.808 | 1.092 | 0.901 | 0.84 | 1.082 | 0.903 |
| MARS1 | 1.038 | 1.371 | 0.825 | 1.064 | 1.581 | 0.805 |
| MARS2 | 1.067 | 1.523 | 0.784 | 1.098 | 1.584 | 0.767 |
| MARS3 | 1.537 | 2.01 | 0.624 | 1.369 | 1.795 | 0.744 |
| MARS4 | 0.659 | 0.972 | 0.912 | 0.806 | 1.39 | 0.861 |
| MARS5 | 0.659 | 0.972 | 0.912 | 0.806 | 1.39 | 0.861 |
| MARS6 | 0.548 | 0.72 | 0.952 | 0.596 | 0.959 | 0.931 |
| MARS7 | 0.507 | 0.641 | 0.962 | 0.581 | 0.763 | 0.949 |
| MLP1 | 1.044 | 1.374 | 0.824 | 1.03 | 1.483 | 0.818 |
| MLP2 | 1.082 | 1.567 | 0.771 | 1.03 | 1.49 | 0.792 |
| MLP3 | 1.135 | 1.618 | 0.757 | 1.04 | 1.46 | 0.798 |
| MLP4 | 0.655 | 0.963 | 0.914 | 0.716 | 1.148 | 0.892 |
| MLP5 | 0.608 | 0.908 | 0.923 | 0.584 | 0.879 | 0.928 |
| MLP6 | 0.314 | 0.458 | 0.98 | 0.409 | 0.607 | 0.97 |
| MLP7 | 0.279 | 0.398 | 0.985 | 0.314 | 0.405 | 0.988 |
| SS | 0.954 | 1.327 | 0.838 | 0.822 | 1.152 | 0.886 |
| MLR | 0.825 | 1.05 | 0.897 | 0.874 | 1.16 | 0.875 |




638 Table 4. Comparisons of different models for predicting $Ep$ at ALT.

| ALT | MAE | RMSE | $R^2$ | MAE | RMSE | $R^2$ |
|---|---|---|---|---|---|---|
| ANFIS-GP1 | 1.19 | 1.597 | 0.841 | 1.003 | 1.268 | 0.896 |
| ANFIS-GP2 | 1.19 | 1.506 | 0.859 | 1.11 | 1.435 | 0.884 |
| ANFIS-GP3 | 1.345 | 1.763 | 0.807 | 1.214 | 1.601 | 0.844 |
| ANFIS-GP4 | 0.535 | 0.786 | 0.962 | 0.707 | 1.013 | 0.973 |
| ANFIS-GP5 | 0.494 | 0.737 | 0.966 | 0.691 | 1.012 | 0.977 |
| ANFIS-GP6 | 0.286 | 0.411 | 0.99 | 0.398 | 0.586 | 0.984 |
| ANFIS-GP7 | 0.241 | 0.351 | 0.992 | 0.371 | 0.545 | 0.987 |
| FG1 | 1.079 | 1.398 | 0.878 | 0.994 | 1.3 | 0.891 |
| FG2 | 0.78 | 1.065 | 0.929 | 0.953 | 1.29 | 0.928 |
| FG3 | 1.052 | 1.375 | 0.882 | 1.002 | 1.328 | 0.92 |
| FG4 | 0.45 | 0.703 | 0.969 | 0.67 | 1.027 | 0.971 |
| FG5 | 0.49 | 0.717 | 0.968 | 0.697 | 1.043 | 0.971 |
| FG6 | 0.266 | 0.38 | 0.991 | 0.391 | 0.575 | 0.987 |
| FG7 | 0.253 | 0.343 | 0.993 | 0.394 | 0.57 | 0.988 |
| GRNN1 | 1.172 | 1.516 | 0.865 | 0.881 | 1.113 | 0.906 |
| GRNN2 | 0.875 | 1.223 | 0.913 | 0.882 | 1.187 | 0.927 |
| GRNN3 | 1.173 | 1.552 | 0.868 | 0.885 | 1.213 | 0.91 |
| GRNN4 | 0.532 | 0.84 | 0.957 | 0.663 | 0.936 | 0.977 |
| GRNN5 | 0.527 | 0.809 | 0.96 | 0.69 | 0.952 | 0.982 |
| GRNN6 | 0.192 | 0.301 | 0.994 | 0.47 | 0.657 | 0.989 |
| GRNN7 | 0.158 | 0.251 | 0.996 | 0.476 | 0.679 | 0.986 |
| LSSV1 | 1.077 | 1.421 | 0.875 | 0.928 | 1.212 | 0.9 |
| LSSV2 | 0.776 | 1.058 | 0.93 | 0.957 | 1.294 | 0.927 |
| LSSV3 | 1.063 | 1.381 | 0.881 | 0.986 | 1.299 | 0.92 |
| LSSV4 | 0.514 | 0.763 | 0.964 | 0.669 | 0.957 | 0.973 |
| LSSV5 | 0.481 | 0.732 | 0.967 | 0.669 | 0.974 | 0.978 |
| LSSV6 | 0.303 | 0.435 | 0.989 | 0.369 | 0.528 | 0.987 |
| LSSV7 | 0.39 | 0.587 | 0.98 | 0.487 | 0.647 | 0.986 |
| MARS1 | 1.032 | 1.371 | 0.883 | 0.956 | 1.269 | 0.899 |
| MARS2 | 0.748 | 1.029 | 0.934 | 0.927 | 1.28 | 0.929 |
| MARS3 | 1.043 | 1.356 | 0.886 | 1.043 | 1.367 | 0.916 |
| MARS4 | 0.437 | 0.657 | 0.973 | 0.641 | 0.996 | 0.975 |
| MARS5 | 0.438 | 0.658 | 0.973 | 0.644 | 1.0 | 0.975 |
| MARS6 | 0.29 | 0.411 | 0.989 | 0.428 | 0.655 | 0.985 |
| MARS7 | 0.276 | 0.382 | 0.991 | 0.403 | 0.622 | 0.987 |
| MLP1 | 1.03 | 1.363 | 0.884 | 0.951 | 1.268 | 0.897 |
| MLP2 | 0.787 | 1.07 | 0.929 | 0.951 | 1.282 | 0.93 |
| MLP3 | 0.752 | 1.039 | 0.933 | 0.93 | 1.28 | 0.929 |
| MLP4 | 0.445 | 0.689 | 0.97 | 0.667 | 1.033 | 0.971 |
| MLP5 | 0.521 | 0.769 | 0.963 | 0.659 | 1.017 | 0.974 |
| MLP6 | 0.234 | 0.34 | 0.993 | 0.348 | 0.523 | 0.989 |
| MLP7 | 0.161 | 0.211 | 0.989 | 0.19 | 0.265 | 0.989 |
| SS | 0.539 | 0.761 | 0.964 | 0.681 | 1.053 | 0.963 |
| MLR | 0.712 | 0.89 | 0.951 | 0.74 | 0.861 | 0.969 |




Table 5. Comparisons of different models for predicting *Ep* at MQ.

| MQ | MAE | RMSE | $R^2$ | MAE | RMSE | $R^2$ |
|---|---|---|---|---|---|---|
| ANFIS-GP1 | 1.337 | 1.76 | 0.85 | 1.133 | 1.396 | 0.941 |
| ANFIS-GP2 | 1.33 | 1.698 | 0.86 | 1.203 | 1.587 | 0.863 |
| ANFIS-GP3 | 2.467 | 3.11 | 0.53 | 2.453 | 3.045 | 0.55 |
| ANFIS-GP4 | 0.83 | 1.178 | 0.933 | 0.868 | 1.22 | 0.952 |
| ANFIS-GP5 | 0.828 | 1.165 | 0.828 | 0.882 | 1.229 | 0.951 |
| ANFIS-GP6 | 0.648 | 0.886 | 0.962 | 0.608 | 0.81 | 0.981 |
| ANFIS-GP7 | 0.474 | 0.66 | 0.979 | 0.512 | 0.646 | 0.987 |
| FG1 | 1.297 | 1.735 | 0.854 | 1.112 | 1.412 | 0.926 |
| FG2 | 1.263 | 1.638 | 0.87 | 1.198 | 1.555 | 0.87 |
| FG3 | 2.447 | 3.057 | 0.546 | 2.373 | 2.953 | 0.58 |
| FG4 | 0.828 | 1.178 | 0.933 | 0.854 | 1.196 | 0.952 |
| FG5 | 0.795 | 1.13 | 0.938 | 0.923 | 1.335 | 0.942 |
| FG6 | 0.608 | 0.81 | 0.968 | 0.636 | 0.805 | 0.978 |
| FG7 | 0.456 | 0.614 | 0.983 | 0.435 | 0.574 | 0.99 |
| GRNN1 | 1.427 | 1.814 | 0.854 | 1.071 | 1.315 | 0.925 |
| GRNN2 | 1.225 | 1.593 | 0.877 | 1.148 | 1.504 | 0.876 |
| GRNN3 | 2.663 | 3.15 | 0.542 | 2.381 | 2.821 | 0.596 |
| GRNN4 | 0.733 | 1.056 | 0.946 | 0.78 | 1.089 | 0.954 |
| GRNN5 | 0.647 | 0.944 | 0.957 | 0.815 | 1.161 | 0.951 |
| GRNN6 | 0.329 | 0.486 | 0.989 | 0.634 | 0.892 | 0.972 |
| GRNN7 | 0.248 | 0.392 | 0.993 | 0.548 | 0.74 | 0.981 |
| LSSV1 | 1.343 | 1.758 | 0.85 | 1.094 | 1.368 | 0.935 |
| LSSV2 | 1.274 | 1.643 | 0.869 | 1.187 | 1.548 | 0.869 |
| LSSV3 | 2.46 | 3.057 | 0.547 | 2.383 | 2.942 | 0.574 |
| LSSV4 | 1.06 | 1.365 | 0.925 | 0.93 | 1.107 | 0.951 |
| LSSV5 | 0.815 | 1.143 | 0.937 | 0.9 | 1.225 | 0.945 |
| LSSV6 | 0.934 | 1.193 | 0.94 | 0.891 | 1.089 | 0.962 |
| LSSV7 | 0.888 | 1.113 | 0.95 | 0.767 | 0.927 | 0.97 |
| MARS1 | 1.291 | 1.728 | 0.855 | 1.092 | 1.404 | 0.928 |
| MARS2 | 1.076 | 1.462 | 0.896 | 1.078 | 1.471 | 0.888 |
| MARS3 | 2.419 | 3.039 | 0.552 | 2.436 | 2.996 | 0.564 |
| MARS4 | 0.815 | 1.144 | 0.936 | 0.922 | 1.23 | 0.947 |
| MARS5 | 0.807 | 1.126 | 0.938 | 0.97 | 1.29 | 0.95 |
| MARS6 | 0.668 | 0.87 | 0.963 | 0.735 | 0.929 | 0.973 |
| MARS7 | 0.546 | 0.72 | 0.975 | 0.627 | 0.826 | 0.977 |
| MLP1 | 1.297 | 1.735 | 0.854 | 1.107 | 1.408 | 0.928 |
| MLP2 | 1.057 | 1.458 | 0.897 | 1.113 | 1.492 | 0.888 |
| MLP3 | 1.139 | 1.524 | 0.887 | 1.108 | 1.488 | 0.884 |
| MLP4 | 0.724 | 1.026 | 0.949 | 0.797 | 1.074 | 0.96 |
| MLP5 | 0.742 | 1.064 | 0.945 | 0.821 | 1.113 | 0.959 |
| MLP6 | 0.538 | 0.738 | 0.974 | 0.538 | 0.716 | 0.981 |
| MLP7 | 0.384 | 0.532 | 0.986 | 0.358 | 0.489 | 0.99 |
| SS | 0.922 | 1.281 | 0.92 | 1.039 | 1.389 | 0.944 |
| MLR | 0.77 | 0.967 | 0.955 | 0.784 | 0.921 | 0.972 |






Table 6. Comparisons of different models for predicting *Ep* at BJ.

| BJ | MAE | RMSE | $R^2$ | MAE | RMSE | $R^2$ |
|---|---|---|---|---|---|---|
| ANFIS-GP1 | 0.872 | 1.205 | 0.826 | 0.749 | 0.956 | 0.922 |
| ANFIS-GP2 | 1.439 | 1.907 | 0.564 | 1.294 | 1.554 | 0.662 |
| ANFIS-GP3 | 1.431 | 1.818 | 0.603 | 1.482 | 1.88 | 0.561 |
| ANFIS-GP4 | 0.846 | 1.189 | 0.831 | 0.717 | 0.923 | 0.921 |
| ANFIS-GP5 | 0.742 | 1.071 | 0.862 | 0.688 | 0.972 | 0.909 |
| ANFIS-GP6 | 0.464 | 0.735 | 0.935 | 0.384 | 0.51 | 0.965 |
| ANFIS-GP7 | 0.424 | 0.657 | 0.948 | 0.361 | 0.48 | 0.971 |
| FG1 | 0.835 | 1.127 | 0.848 | 0.823 | 1.075 | 0.914 |
| FG2 | 1.416 | 1.891 | 0.571 | 1.256 | 1.544 | 0.665 |
| FG3 | 1.387 | 1.733 | 0.64 | 1.483 | 1.846 | 0.561 |
| FG4 | 0.742 | 1.063 | 0.864 | 0.688 | 0.997 | 0.922 |
| FG5 | 0.721 | 1.052 | 0.867 | 0.679 | 0.959 | 0.926 |
| FG6 | 0.451 | 0.721 | 0.938 | 0.394 | 0.484 | 0.971 |
| FG7 | 0.431 | 0.655 | 0.949 | 0.431 | 0.586 | 0.963 |
| GRNN1 | 0.819 | 1.114 | 0.852 | 0.811 | 1.062 | 0.916 |
| GRNN2 | 1.379 | 1.852 | 0.589 | 1.23 | 1.52 | 0.678 |
| GRNN3 | 1.374 | 1.727 | 0.647 | 1.491 | 1.843 | 0.564 |
| GRNN4 | 0.626 | 0.924 | 0.898 | 0.657 | 0.939 | 0.904 |
| GRNN5 | 0.665 | 0.998 | 0.882 | 0.639 | 0.947 | 0.914 |
| GRNN6 | 0.32 | 0.533 | 0.966 | 0.391 | 0.513 | 0.962 |
| GRNN7 | 0.166 | 0.301 | 0.989 | 0.356 | 0.473 | 0.968 |
| LSSV1 | 0.842 | 1.139 | 0.845 | 0.824 | 1.062 | 0.911 |
| LSSV2 | 1.519 | 1.986 | 0.552 | 1.441 | 1.658 | 0.647 |
| LSSV3 | 1.386 | 1.734 | 0.64 | 1.483 | 1.841 | 0.562 |
| LSSV4 | 0.743 | 1.069 | 0.864 | 0.69 | 0.977 | 0.922 |
| LSSV5 | 0.823 | 1.184 | 0.839 | 0.692 | 0.958 | 0.911 |
| LSSV6 | 0.736 | 1.078 | 0.875 | 0.622 | 0.827 | 0.925 |
| LSSV7 | 0.486 | 0.76 | 0.933 | 0.338 | 0.444 | 0.973 |
| MARS1 | 0.829 | 1.118 | 0.85 | 0.854 | 1.071 | 0.915 |
| MARS2 | 1.364 | 1.832 | 0.597 | 1.282 | 1.607 | 0.659 |
| MARS3 | 1.355 | 1.717 | 0.647 | 1.477 | 1.846 | 0.565 |
| MARS4 | 0.705 | 0.979 | 0.885 | 0.68 | 0.976 | 0.909 |
| MARS5 | 0.687 | 0.972 | 0.887 | 0.687 | 0.991 | 0.903 |
| MARS6 | 0.52 | 0.767 | 0.929 | 0.5 | 0.603 | 0.963 |
| MARS7 | 0.478 | 0.717 | 0.938 | 0.427 | 0.527 | 0.971 |
| MLP1 | 0.784 | 1.075 | 0.861 | 0.813 | 1.045 | 0.914 |
| MLP2 | 1.325 | 1.803 | 0.61 | 1.249 | 1.595 | 0.677 |
| MLP3 | 1.401 | 1.875 | 0.578 | 1.236 | 1.523 | 0.678 |
| MLP4 | 0.675 | 0.968 | 0.888 | 0.66 | 0.974 | 0.911 |
| MLP5 | 0.653 | 0.962 | 0.889 | 0.62 | 0.907 | 0.904 |
| MLP6 | 0.417 | 0.692 | 0.943 | 0.312 | 0.394 | 0.982 |
| MLP7 | 0.337 | 0.506 | 0.969 | 0.314 | 0.428 | 0.979 |
| SS | 0.89 | 1.263 | 0.816 | 0.647 | 0.921 | 0.897 |
| MLR | 0.614 | 0.879 | 0.907 | 0.514 | 0.648 | 0.946 |






Table 7. Comparisons of different models for predicting $Ep$ at LSA.

| LSA | MAE | RMSE | $R^2$ | MAE | RMSE | $R^2$ |
|---|---|---|---|---|---|---|
| ANFIS-GP1 | 1.327 | 1.718 | 0.411 | 1.072 | 1.424 | 0.594 |
| ANFIS-GP2 | 1.245 | 1.523 | 0.536 | 1.192 | 1.417 | 0.601 |
| ANFIS-GP3 | 1.821 | 2.218 | 0.017 | 1.796 | 2.148 | 0.055 |
| ANFIS-GP4 | 1.149 | 1.471 | 0.568 | 1.046 | 1.304 | 0.651 |
| ANFIS-GP5 | 0.966 | 1.223 | 0.701 | 0.875 | 1.082 | 0.761 |
| ANFIS-GP6 | 0.529 | 0.675 | 0.909 | 0.73 | 0.907 | 0.896 |
| ANFIS-GP7 | 0.478 | 0.61 | 0.926 | 0.816 | 1.038 | 0.875 |
| FG1 | 1.324 | 1.715 | 0.413 | 1.073 | 1.415 | 0.6 |
| FG2 | 1.151 | 1.465 | 0.571 | 1.159 | 1.392 | 0.621 |
| FG3 | 1.803 | 2.169 | 0.06 | 1.771 | 2.093 | 0.118 |
| FG4 | 1.044 | 1.381 | 0.619 | 0.987 | 1.201 | 0.725 |
| FG5 | 0.968 | 1.215 | 0.705 | 0.896 | 1.099 | 0.757 |
| FG6 | 0.499 | 0.631 | 0.921 | 0.767 | 0.925 | 0.903 |
| FG7 | 0.491 | 0.61 | 0.926 | 0.729 | 0.886 | 0.914 |
| GRNN1 | 1.296 | 1.692 | 0.429 | 1.094 | 1.436 | 0.587 |
| GRNN2 | 1.025 | 1.336 | 0.647 | 1.072 | 1.288 | 0.679 |
| GRNN3 | 1.783 | 2.152 | 0.077 | 1.762 | 2.08 | 0.134 |
| GRNN4 | 0.841 | 1.131 | 0.751 | 0.817 | 1.032 | 0.795 |
| GRNN5 | 0.639 | 0.844 | 0.862 | 0.714 | 0.937 | 0.828 |
| GRNN6 | 0.33 | 0.427 | 0.965 | 0.533 | 0.65 | 0.926 |
| GRNN7 | 0.326 | 0.417 | 0.967 | 0.459 | 0.592 | 0.933 |
| LSSV1 | 1.376 | 1.754 | 0.41 | 1.211 | 1.508 | 0.599 |
| LSSV2 | 1.22 | 1.499 | 0.554 | 1.213 | 1.422 | 0.606 |
| LSSV3 | 1.811 | 2.209 | 0.027 | 1.791 | 2.144 | 0.07 |
| LSSV4 | 1.163 | 1.476 | 0.573 | 1.078 | 1.31 | 0.663 |
| LSSV5 | 0.987 | 1.253 | 0.69 | 0.894 | 1.085 | 0.777 |
| LSSV6 | 0.462 | 0.601 | 0.933 | 0.646 | 0.799 | 0.916 |
| LSSV7 | 0.47 | 0.609 | 0.932 | 0.591 | 0.713 | 0.928 |
| MARS1 | 1.316 | 1.713 | 0.414 | 1.072 | 1.412 | 0.602 |
| MARS2 | 1.012 | 1.318 | 0.653 | 1.098 | 1.299 | 0.683 |
| MARS3 | 1.82 | 2.182 | 0.049 | 1.766 | 2.089 | 0.12 |
| MARS4 | 0.917 | 1.23 | 0.698 | 0.947 | 1.176 | 0.735 |
| MARS5 | 0.94 | 1.227 | 0.699 | 0.913 | 1.135 | 0.746 |
| MARS6 | 0.501 | 0.641 | 0.918 | 0.762 | 0.929 | 0.91 |
| MARS7 | 0.528 | 0.66 | 0.913 | 0.697 | 0.85 | 0.92 |
| MLP1 | 1.308 | 1.707 | 0.418 | 1.073 | 1.413 | 0.596 |
| MLP2 | 0.992 | 1.307 | 0.659 | 1.111 | 1.313 | 0.675 |
| MLP3 | 0.994 | 1.316 | 0.654 | 1.108 | 1.312 | 0.675 |
| MLP4 | 0.883 | 1.187 | 0.719 | 0.918 | 1.123 | 0.754 |
| MLP5 | 0.686 | 0.91 | 0.835 | 0.728 | 0.958 | 0.825 |
| MLP6 | 0.397 | 0.503 | 0.949 | 0.629 | 0.771 | 0.928 |
| MLP7 | 0.522 | 0.681 | 0.907 | 0.53 | 0.638 | 0.936 |
| SS | 1.198 | 1.577 | 0.515 | 0.969 | 1.307 | 0.652 |
| MLR | 0.628 | 0.795 | 0.874 | 0.656 | 0.789 | 0.906 |






Table 8. Comparisons of different models for predicting *Ep* at CQ.

| CQ | MAE | RMSE | $R^2$ | MAE | RMSE | $R^2$ |
|---|---|---|---|---|---|---|
| ANFIS-GP1 | 0.466 | 0.859 | 0.815 | 0.28 | 0.397 | 0.958 |
| ANFIS-GP2 | 0.82 | 1.189 | 0.645 | 0.693 | 0.959 | 0.748 |
| ANFIS-GP3 | 0.539 | 0.722 | 0.869 | 0.537 | 0.679 | 0.876 |
| ANFIS-GP4 | 0.416 | 0.786 | 0.845 | 0.316 | 0.398 | 0.959 |
| ANFIS-GP5 | 0.369 | 0.492 | 0.939 | 0.242 | 0.329 | 0.968 |
| ANFIS-GP6 | 0.225 | 0.29 | 0.979 | 0.224 | 0.312 | 0.976 |
| ANFIS-GP7 | 0.187 | 0.244 | 0.985 | 0.203 | 0.3 | 0.978 |
| FG1 | 0.467 | 0.805 | 0.837 | 0.294 | 0.375 | 0.963 |
| FG2 | 0.611 | 0.881 | 0.805 | 0.571 | 0.691 | 0.873 |
| FG3 | 0.474 | 0.672 | 0.887 | 0.479 | 0.607 | 0.905 |
| FG4 | 0.385 | 0.704 | 0.876 | 0.303 | 0.384 | 0.96 |
| FG5 | 0.297 | 0.402 | 0.959 | 0.273 | 0.19 | 0.944 |
| FG6 | 0.275 | 0.359 | 0.968 | 0.3 | 0.407 | 0.955 |
| FG7 | 0.195 | 0.25 | 0.984 | 0.182 | 0.28 | 0.981 |
| GRNN1 | 0.437 | 0.746 | 0.861 | 0.284 | 0.374 | 0.963 |
| GRNN2 | 0.574 | 0.845 | 0.823 | 0.507 | 0.651 | 0.883 |
| GRNN3 | 0.453 | 0.652 | 0.893 | 0.473 | 0.61 | 0.902 |
| GRNN4 | 0.328 | 0.645 | 0.897 | 0.285 | 0.37 | 0.962 |
| GRNN5 | 0.221 | 0.308 | 0.976 | 0.24 | 0.327 | 0.968 |
| GRNN6 | 0.145 | 0.203 | 0.99 | 0.177 | 0.24 | 0.983 |
| GRNN7 | 0.227 | 0.308 | 0.977 | 0.234 | 0.297 | 0.975 |
| LSSV1 | 0.714 | 1.028 | 0.81 | 0.6 | 0.734 | 0.961 |
| LSSV2 | 0.552 | 0.825 | 0.829 | 0.503 | 0.65 | 0.888 |
| LSSV3 | 0.687 | 0.862 | 0.887 | 0.625 | 0.765 | 0.906 |
| LSSV4 | 0.543 | 0.873 | 0.833 | 0.449 | 0.58 | 0.94 |
| LSSV5 | 0.336 | 0.48 | 0.942 | 0.292 | 0.372 | 0.959 |
| LSSV6 | 0.314 | 0.496 | 0.94 | 0.219 | 0.284 | 0.977 |
| LSSV7 | 0.317 | 0.49 | 0.942 | 0.22 | 0.292 | 0.976 |
| MARS1 | 0.451 | 0.709 | 0.874 | 0.28 | 0.441 | 0.943 |
| MARS2 | 0.555 | 0.822 | 0.83 | 0.498 | 0.651 | 0.889 |
| MARS3 | 0.453 | 0.664 | 0.889 | 0.466 | 0.599 | 0.904 |
| MARS4 | 0.363 | 0.624 | 0.902 | 0.33 | 0.441 | 0.95 |
| MARS5 | 0.336 | 0.48 | 0.942 | 0.292 | 0.372 | 0.959 |
| MARS6 | 0.273 | 0.426 | 0.954 | 0.219 | 0.299 | 0.974 |
| MARS7 | 0.267 | 0.417 | 0.956 | 0.25 | 0.323 | 0.956 |
| MLP1 | 0.419 | 0.733 | 0.865 | 0.27 | 0.371 | 0.96 |
| MLP2 | 0.55 | 0.81 | 0.835 | 0.509 | 0.658 | 0.887 |
| MLP3 | 0.568 | 0.845 | 0.82 | 0.502 | 0.637 | 0.893 |
| MLP4 | 0.334 | 0.65 | 0.894 | 0.266 | 0.355 | 0.966 |
| MLP5 | 0.252 | 0.348 | 0.97 | 0.218 | 0.296 | 0.975 |
| MLP6 | 0.185 | 0.239 | 0.986 | 0.167 | 0.23 | 0.985 |
| MLP7 | 0.161 | 0.211 | 0.989 | 0.189 | 0.265 | 0.985 |
| SS | 0.379 | 0.786 | 0.847 | 0.226 | 0.307 | 0.973 |
| MLR | 0.389 | 0.534 | 0.928 | 0.317 | 0.398 | 0.955 |






Table 9. Comparisons of different models for predicting *Ep* at HZ station.

|  | MAE | RMSE | $R^2$ | MAE | RMSE | $R^2$ |
|---|---|---|---|---|---|---|
| ANFIS-GP1 | 0.532 | 0.698 | 0.87 | 0.451 | 0.605 | 0.903 |
| ANFIS-GP2 | 0.72 | 1.001 | 0.734 | 0.728 | 0.965 | 0.754 |
| ANFIS-GP3 | 0.937 | 1.164 | 0.64 | 0.991 | 1.178 | 0.694 |
| ANFIS-GP4 | 0.377 | 0.521 | 0.928 | 0.333 | 0.448 | 0.948 |
| ANFIS-GP5 | 0.357 | 0.482 | 0.938 | 0.311 | 0.397 | 0.961 |
| ANFIS-GP6 | 0.272 | 0.356 | 0.966 | 0.329 | 0.427 | 0.965 |
| ANFIS-GP7 | 0.242 | 0.312 | 0.974 | 0.347 | 0.453 | 0.949 |
| FG1 | 0.519 | 0.686 | 0.875 | 0.438 | 0.59 | 0.908 |
| FG2 | 0.62 | 0.817 | 0.822 | 0.626 | 0.786 | 0.837 |
| FG3 | 0.943 | 1.151 | 0.648 | 1.01 | 1.188 | 0.699 |
| FG4 | 0.358 | 0.485 | 0.938 | 0.299 | 0.397 | 0.959 |
| FG5 | 0.344 | 0.462 | 0.943 | 0.29 | 0.373 | 0.965 |
| FG6 | 0.269 | 0.347 | 0.968 | 0.295 | 0.375 | 0.974 |
| FG7 | 0.26 | 0.36 | 0.966 | 0.278 | 0.369 | 0.964 |
| GRNN1 | 0.519 | 0.68 | 0.878 | 0.457 | 0.607 | 0.904 |
| GRNN2 | 0.556 | 0.733 | 0.859 | 0.581 | 0.736 | 0.86 |
| GRNN3 | 0.926 | 1.127 | 0.664 | 1.02 | 1.197 | 0.705 |
| GRNN4 | 0.322 | 0.438 | 0.949 | 0.314 | 0.409 | 0.957 |
| GRNN5 | 0.238 | 0.327 | 0.972 | 0.295 | 0.404 | 0.961 |
| GRNN6 | 0.232 | 0.3 | 0.977 | 0.275 | 0.346 | 0.969 |
| GRNN7 | 0.223 | 0.295 | 0.978 | 0.335 | 0.445 | 0.956 |
| LSSV1 | 0.593 | 0.801 | 0.87 | 0.572 | 0.731 | 0.903 |
| LSSV2 | 0.715 | 0.984 | 0.778 | 0.733 | 0.97 | 0.799 |
| LSSV3 | 0.996 | 1.214 | 0.638 | 1.074 | 1.267 | 0.678 |
| LSSV4 | 0.413 | 0.594 | 0.924 | 0.399 | 0.548 | 0.94 |
| LSSV5 | 0.398 | 0.554 | 0.929 | 0.376 | 0.509 | 0.953 |
| LSSV6 | 0.278 | 0.378 | 0.964 | 0.3 | 0.372 | 0.968 |
| LSSV7 | 0.292 | 0.406 | 0.959 | 0.338 | 0.441 | 0.957 |
| MARS1 | 0.52 | 0.69 | 0.874 | 0.443 | 0.601 | 0.904 |
| MARS2 | 0.534 | 0.686 | 0.875 | 0.524 | 0.673 | 0.881 |
| MARS3 | 0.915 | 1.125 | 0.664 | 0.999 | 1.189 | 0.698 |
| MARS4 | 0.339 | 0.449 | 0.946 | 0.273 | 0.362 | 0.966 |
| MARS5 | 0.335 | 0.437 | 0.949 | 0.282 | 0.358 | 0.966 |
| MARS6 | 0.286 | 0.37 | 0.964 | 0.318 | 0.393 | 0.976 |
| MARS7 | 0.27 | 0.358 | 0.966 | 0.276 | 0.361 | 0.967 |
| MLP1 | 0.529 | 0.691 | 0.873 | 0.449 | 0.598 | 0.906 |
| MLP2 | 0.523 | 0.68 | 0.877 | 0.523 | 0.674 | 0.881 |
| MLP3 | 0.908 | 1.124 | 0.664 | 0.992 | 1.181 | 0.698 |
| MLP4 | 0.334 | 0.65 | 0.894 | 0.266 | 0.355 | 0.966 |
| MLP5 | 0.333 | 0.446 | 0.947 | 0.279 | 0.348 | 0.968 |
| MLP6 | 0.247 | 0.326 | 0.972 | 0.318 | 0.405 | 0.978 |
| MLP7 | 0.244 | 0.319 | 0.973 | 0.263 | 0.34 | 0.977 |
| SS | 0.35 | 0.487 | 0.938 | 0.291 | 0.388 | 0.96 |
| MLR | 0.32 | 0.427 | 0.952 | 0.395 | 0.486 | 0.942 |




Table 10. Comparisons of different models for predicting *Ep* at HK.

| HK | MAE | RMSE | $R^2$ | MAE | RMSE | $R^2$ |
|---|---|---|---|---|---|---|
| ANFIS-GP1 | 0.528 | 0.688 | 0.814 | 0.669 | 0.8 | 0.854 |
| ANFIS-GP2 | 0.741 | 0.964 | 0.634 | 0.802 | 0.97 | 0.742 |
| ANFIS-GP3 | 0.619 | 0.798 | 0.749 | 0.482 | 0.61 | 0.851 |
| ANFIS-GP4 | 0.488 | 0.646 | 0.836 | 0.66 | 0.796 | 0.861 |
| ANFIS-GP5 | 0.46 | 0.597 | 0.86 | 0.494 | 0.609 | 0.891 |
| ANFIS-GP6 | 0.388 | 0.501 | 0.901 | 0.809 | 0.93 | 0.919 |
| ANFIS-GP7 | 0.286 | 0.379 | 0.943 | 0.428 | 0.555 | 0.925 |
| FG1 | 0.506 | 0.661 | 0.828 | 0.662 | 0.792 | 0.858 |
| FG2 | 0.716 | 0.914 | 0.671 | 0.793 | 0.94 | 0.784 |
| FG3 | 0.612 | 0.768 | 0.768 | 0.503 | 0.63 | 0.85 |
| FG4 | 0.471 | 0.626 | 0.846 | 0.659 | 0.786 | 0.875 |
| FG5 | 0.451 | 0.591 | 0.863 | 0.485 | 0.596 | 0.895 |
| FG6 | 0.39 | 0.496 | 0.903 | 0.718 | 0.849 | 0.92 |
| FG7 | 0.381 | 0.494 | 0.904 | 0.452 | 0.566 | 0.886 |
| GRNN1 | 0.505 | 0.666 | 0.829 | 0.673 | 0.81 | 0.854 |
| GRNN2 | 0.699 | 0.902 | 0.681 | 0.786 | 0.929 | 0.776 |
| GRNN3 | 0.6 | 0.759 | 0.775 | 0.511 | 0.642 | 0.845 |
| GRNN4 | 0.452 | 0.605 | 0.859 | 0.65 | 0.771 | 0.879 |
| GRNN5 | 0.405 | 0.535 | 0.889 | 0.484 | 0.589 | 0.892 |
| GRNN6 | 0.408 | 0.538 | 0.894 | 0.539 | 0.651 | 0.916 |
| GRNN7 | 0.241 | 0.342 | 0.956 | 0.415 | 0.512 | 0.917 |
| LSSV1 | 0.51 | 0.671 | 0.826 | 0.659 | 0.791 | 0.859 |
| LSSV2 | 0.717 | 0.924 | 0.665 | 0.788 | 0.934 | 0.78 |
| LSSV3 | 0.614 | 0.781 | 0.766 | 0.519 | 0.643 | 0.852 |
| LSSV4 | 0.481 | 0.64 | 0.841 | 0.661 | 0.789 | 0.87 |
| LSSV5 | 0.446 | 0.583 | 0.867 | 0.483 | 0.596 | 0.891 |
| LSSV6 | 0.414 | 0.528 | 0.891 | 0.625 | 0.748 | 0.919 |
| LSSV7 | 0.313 | 0.41 | 0.935 | 0.419 | 0.529 | 0.918 |
| MARS1 | 0.505 | 0.662 | 0.828 | 0.665 | 0.79 | 0.862 |
| MARS2 | 0.664 | 0.862 | 0.708 | 0.858 | 1.023 | 0.766 |
| MARS3 | 0.603 | 0.758 | 0.774 | 0.5 | 0.632 | 0.845 |
| MARS4 | 0.438 | 0.581 | 0.867 | 0.733 | 0.899 | 0.869 |
| MARS5 | 0.426 | 0.547 | 0.882 | 0.536 | 0.691 | 0.891 |
| MARS6 | 0.407 | 0.517 | 0.895 | 0.682 | 0.807 | 0.917 |
| MARS7 | 0.322 | 0.414 | 0.932 | 0.397 | 0.515 | 0.927 |
| MLP1 | 0.512 | 0.671 | 0.823 | 0.657 | 0.793 | 0.855 |
| MLP2 | 0.686 | 0.878 | 0.697 | 0.822 | 0.979 | 0.792 |
| MLP3 | 0.707 | 0.903 | 0.679 | 0.821 | 0.973 | 0.79 |
| MLP4 | 0.47 | 0.623 | 0.847 | 0.657 | 0.779 | 0.878 |
| MLP5 | 0.421 | 0.542 | 0.884 | 0.485 | 0.594 | 0.897 |
| MLP6 | 0.431 | 0.554 | 0.88 | 0.671 | 0.786 | 0.916 |
| MLP7 | 0.34 | 0.444 | 0.923 | 0.386 | 0.491 | 0.930 |
| SS | 0.523 | 0.683 | 0.827 | 0.64 | 0.773 | 0.823 |
| MLR | 0.328 | 0.431 | 0.927 | 0.396 | 0.505 | 0.927 |




Table 11. Accuracy ranks of the soft computing models in estimating *Ep*.

| Stations | ANFIS-GP | FG | GRNN | LSSVR | MARS | MLP | MLR |
|----------|----------|-----|------|-------|------|-----|-----|
| HEB | 2 | 3 | 4 | 6 | 5 | 1 | 7 |
| ALT | 3 | 4 | 6 | 2 | 5 | 1 | 7 |
| MQ | 3 | 2 | 4 | 7 | 5 | 1 | 6 |
| BJ | 4 | 5 | 3 | 2 | 6 | 1 | 7 |
| LSA | 6 | 5 | 1 | 3 | 4 | 2 | 7 |
| CQ | 6 | 3 | 2 | 4 | 5 | 1 | 7 |
| HZ | 6 | 4 | 2 | 5 | 3 | 1 | 7 |
| HK | 6 | 7 | 3 | 5 | 4 | 1 | 2 |
| ALL | 4 | 5 | 2 | 3 | 6 | 1 | 7 |
| **Total** | **40** | **38** | **27** | **37** | **43** | **10** | **57** |






Table 12. Comparisons of different models for predicting *Ep* at all stations.

|  | MAE | RMSE | $R^2$ | MAE | RMSE | $R^2$ |
|---|---|---|---|---|---|---|
| ANFIS-GP1 | 1.204 | 1.681 | 0.739 | 1.022 | 1.378 | 0.804 |
| ANFIS-GP2 | 1.906 | 2.522 | 0.412 | 1.768 | 2.345 | 0.437 |
| ANFIS-GP3 | 1.913 | 2.377 | 0.478 | 1.877 | 2.262 | 0.475 |
| ANFIS-GP4 | 0.994 | 1.446 | 0.807 | 0.88 | 1.228 | 0.847 |
| ANFIS-GP5 | 0.917 | 1.341 | 0.834 | 0.782 | 1.113 | 0.872 |
| ANFIS-GP6 | 0.606 | 0.846 | 0.934 | 0.601 | 0.833 | 0.933 |
| ANFIS-GP7 | 0.517 | 0.738 | 0.95 | 0.486 | 0.666 | 0.957 |
| FG1 | 1.208 | 1.676 | 0.74 | 1.028 | 1.377 | 0.805 |
| FG2 | 1.883 | 2.511 | 0.417 | 1.741 | 2.332 | 0.443 |
| FG3 | 1.8 | 2.221 | 0.544 | 1.812 | 2.148 | 0.524 |
| FG4 | 0.936 | 1.378 | 0.824 | 0.821 | 1.154 | 0.865 |
| FG5 | 0.883 | 1.294 | 0.845 | 0.753 | 1.072 | 0.882 |
| FG6 | 0.589 | 0.834 | 0.936 | 0.607 | 0.842 | 0.931 |
| FG7 | 0.518 | 0.744 | 0.949 | 0.495 | 0.678 | 0.956 |
| GRNN1 | 1.193 | 1.669 | 0.743 | 1.013 | 1.373 | 0.806 |
| GRNN2 | 1.859 | 2.49 | 0.427 | 1.716 | 2.311 | 0.453 |
| GRNN3 | 1.772 | 2.216 | 0.549 | 1.773 | 2.127 | 0.532 |
| GRNN4 | 0.819 | 1.234 | 0.86 | 0.733 | 1.075 | 0.884 |
| GRNN5 | 0.724 | 1.114 | 0.886 | 0.642 | 0.963 | 0.905 |
| GRNN6 | 0.458 | 0.674 | 0.958 | 0.489 | 0.723 | 0.947 |
| GRNN7 | 0.265 | 0.425 | 0.984 | 0.364 | 0.573 | 0.967 |
| LSSV1 | 1.198 | 1.667 | 0.743 | 1.017 | 1.371 | 0.807 |
| LSSV2 | 1.85 | 2.495 | 0.425 | 1.703 | 2.312 | 0.453 |
| LSSV3 | 1.854 | 2.314 | 0.506 | 1.858 | 2.215 | 0.493 |
| LSSV4 | 0.935 | 1.386 | 0.823 | 0.806 | 1.149 | 0.866 |
| LSSV5 | 0.933 | 1.369 | 0.827 | 0.8 | 1.134 | 0.867 |
| LSSV6 | 0.824 | 1.148 | 0.879 | 0.774 | 1.023 | 0.893 |
| LSSV7 | 0.494 | 0.719 | 0.952 | 0.476 | 0.657 | 0.958 |
| MARS1 | 1.198 | 1.666 | 0.744 | 1.021 | 1.373 | 0.806 |
| MARS2 | 1.793 | 2.428 | 0.455 | 1.676 | 2.268 | 0.476 |
| MARS3 | 1.782 | 2.209 | 0.549 | 1.788 | 2.131 | 0.532 |
| MARS4 | 1.025 | 1.439 | 0.808 | 0.929 | 1.235 | 0.845 |
| MARS5 | 0.925 | 1.324 | 0.838 | 0.804 | 1.113 | 0.873 |
| MARS6 | 0.783 | 1.032 | 0.902 | 0.76 | 0.963 | 0.909 |
| MARS7 | 0.692 | 0.933 | 0.920 | 0.654 | 0.829 | 0.932 |
| MLP1 | 1.196 | 1.663 | 0.744 | 1.02 | 1.373 | 0.806 |
| MLP2 | 1.835 | 2.485 | 0.429 | 1.689 | 2.304 | 0.457 |
| MLP3 | 1.842 | 2.491 | 0.426 | 1.695 | 2.302 | 0.458 |
| MLP4 | 0.836 | 1.256 | 0.854 | 0.74 | 1.086 | 0.882 |
| MLP5 | 0.774 | 1.181 | 0.871 | 0.649 | 0.98 | 0.902 |
| MLP6 | 0.529 | 0.758 | 0.947 | 0.531 | 0.77 | 0.943 |
| MLP7 | 0.279 | 0.398 | 0.985 | 0.314 | 0.405 | 0.988 |
| SS | 1.107 | 1.544 | 0.785 | 1.007 | 1.336 | 0.823 |
| MLR | 0.905 | 1.235 | 0.859 | 0.86 | 1.091 | 0.88 |






**Figure captions**:
Fig.1. Schematic architecture of: a) MLP neural network; b) GRNN.
Fig.2. Schematic architecture of network-based ANFIS.
Fig.3. The geographical locations of the stations in different climatic zones.
Fig.4. The annual variations of $Ep$ and associated climatic parameters in each station.
Fig.5. Monthly variations of $Ep$ and associated climatic parameters in each station.
Fig.6. Comparison of the observed and estimated $Ep$ using the optimal ANFIS-GP model
during the testing period.
Fig.7. Comparison of the observed and estimated $Ep$ using the optimal FG model during the
testing period.
Fig.8. Comparison of the observed and estimated $Ep$ using the optimal GRNN model during
the testing period.
Fig.9. Comparison of the observed and estimated $Ep$ using the optimal LSSVM model during
the testing period.
Fig.10. Comparison of the observed and estimated $Ep$ using the optimal MARS model during
the testing period.
Fig.11. Comparison of the observed and estimated $Ep$ using the optimal MLP model during
the testing period.
Fig.12. Comparison of the observed and estimated $Ep$ using the optimal SS model during the
testing period.
Fig.13. Comparison of the observed and estimated $Ep$ using the optimal MLR model during
the testing period.





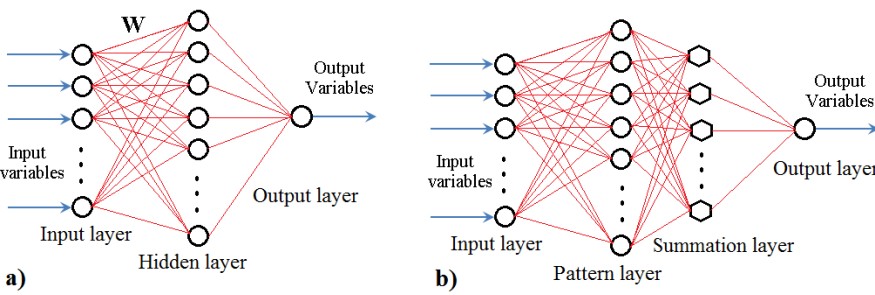


Fig. 1. Schematic architecture of: a) MLP neural network; b) GRNN.




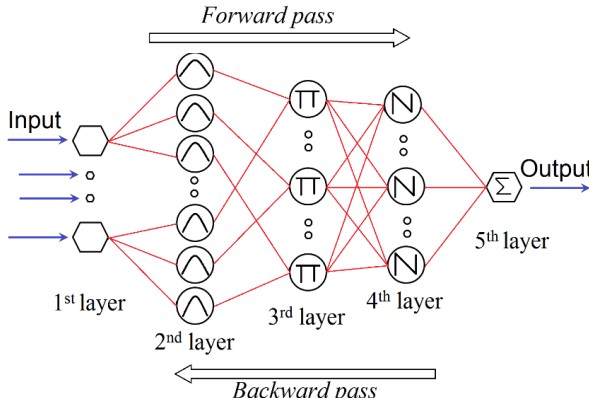


Fig. 2. Schematic architecture of network-based ANFIS.




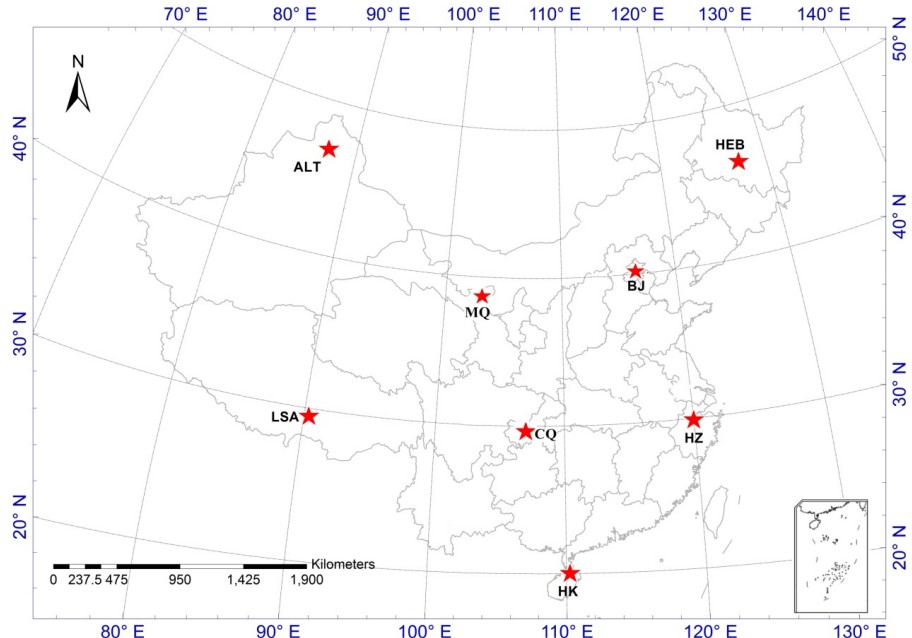

Fig.3. The geographical locations of the stations in different climatic zones.




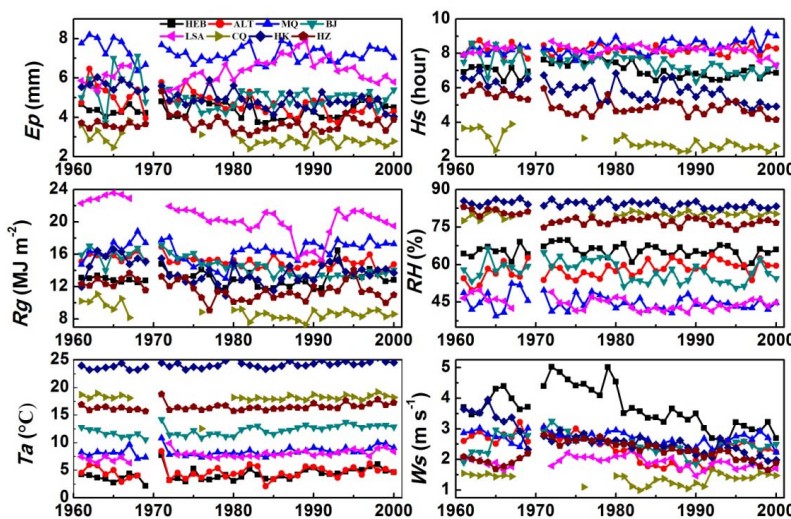


Fig.4. The annual variations of *Ep* and associated climatic parameters in each station.






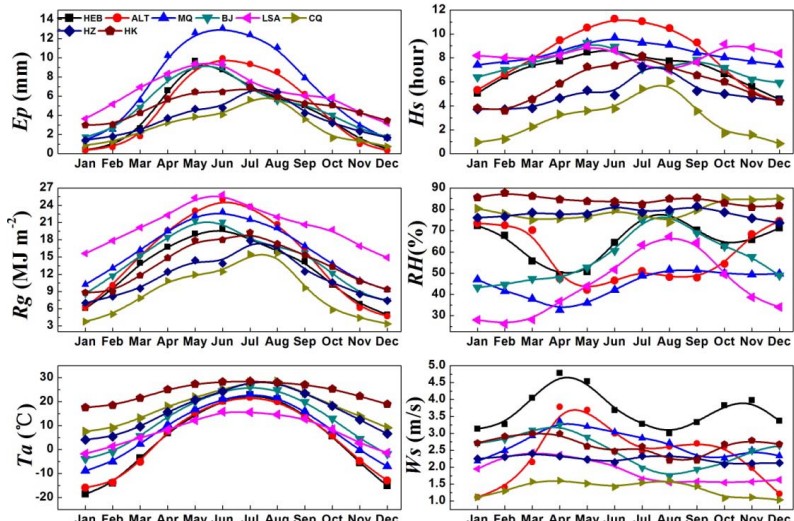


Fig.5. Monthly variations of *Ep* and associated climatic parameters in each station.





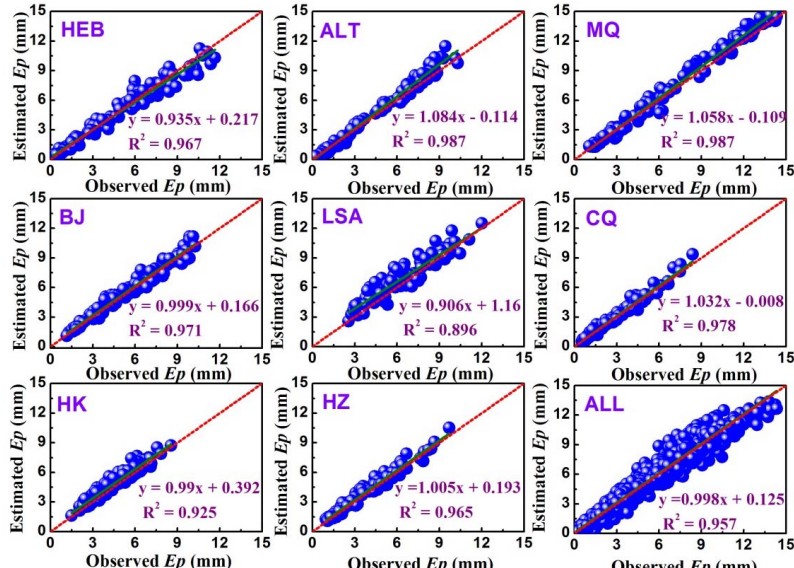


Fig.6. Comparison of the observed and estimated *Ep* using the optimal ANFIS-GP model
during the testing period.

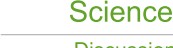
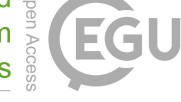


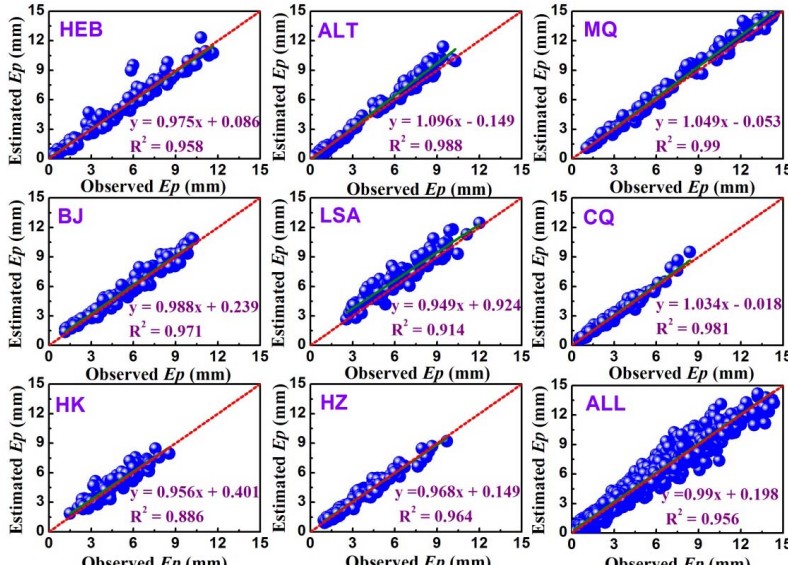


Fig.7. Comparison of the observed and estimated *Ep* using the optimal FG model during the
testing period.




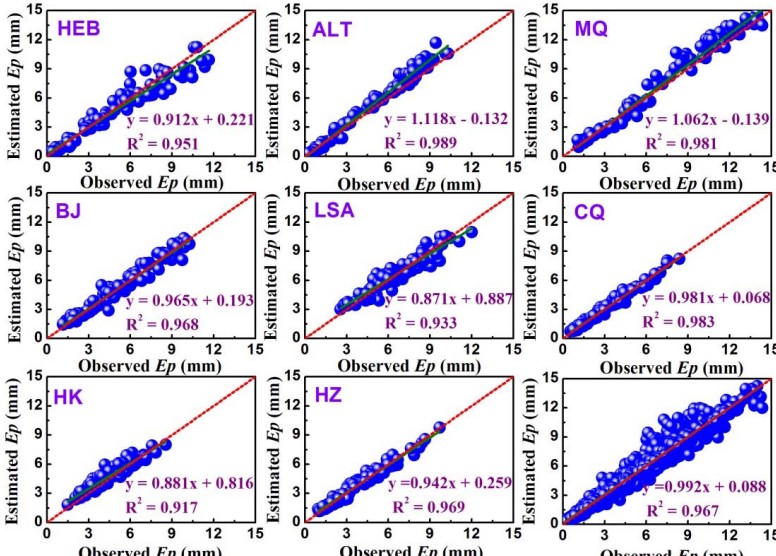


Fig.8. Comparison of the observed and estimated *Ep* using the optimal GRNN model during
the testing period.






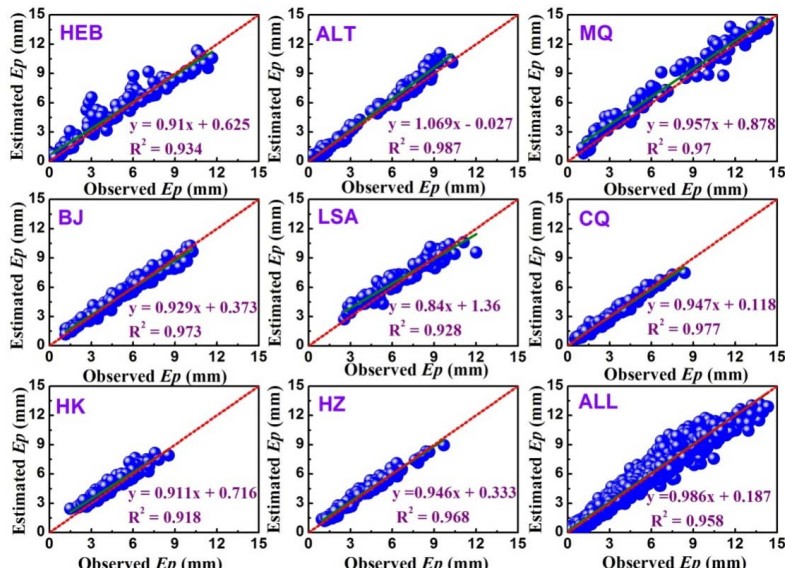


Fig.9. Comparison of the observed and estimated *Ep* using the optimal LSSVM model during
the testing period.





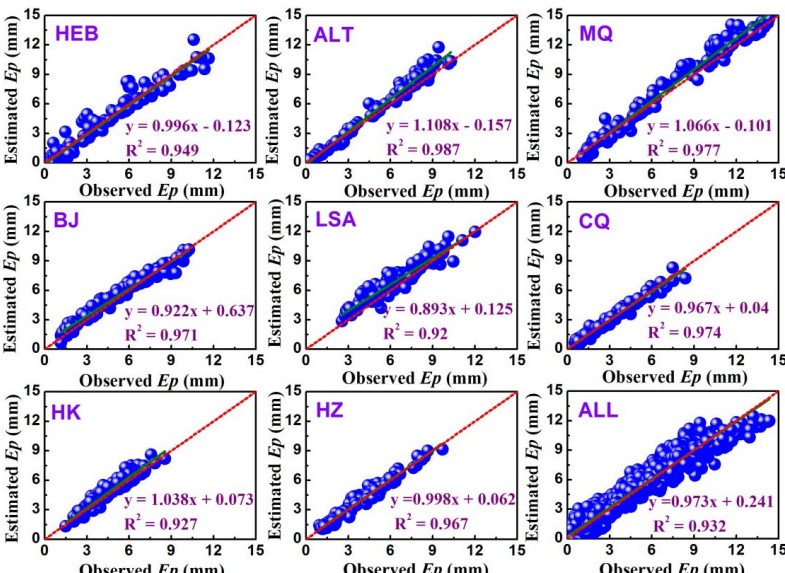


Fig.10. Comparison of the observed and estimated $Ep$ using the optimal MARS model during
the testing period.






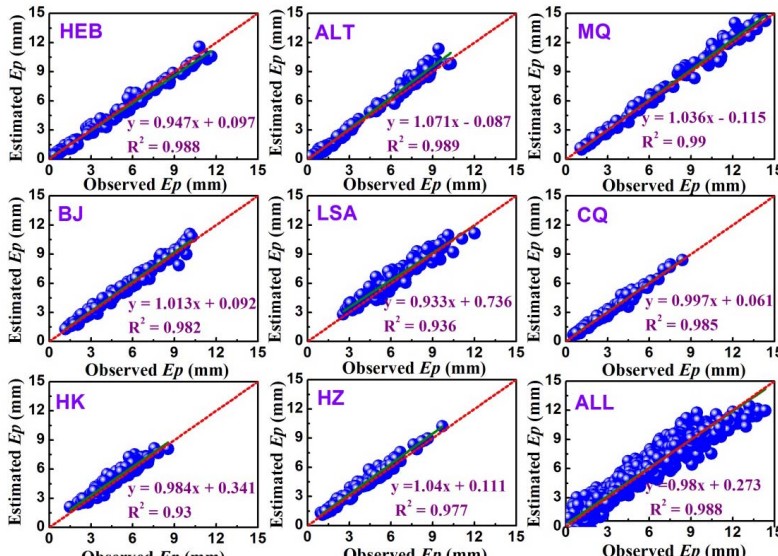


Fig.11. Comparison of the observed and estimated $Ep$ using the optimal MLP model during
the testing period.




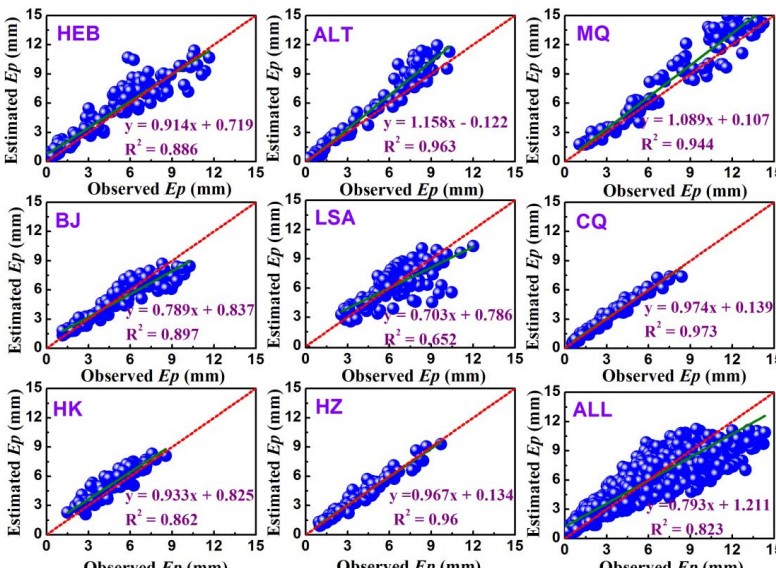


Fig.12. Comparison of the observed and estimated *Ep* using the optimal SS model during the
testing period.






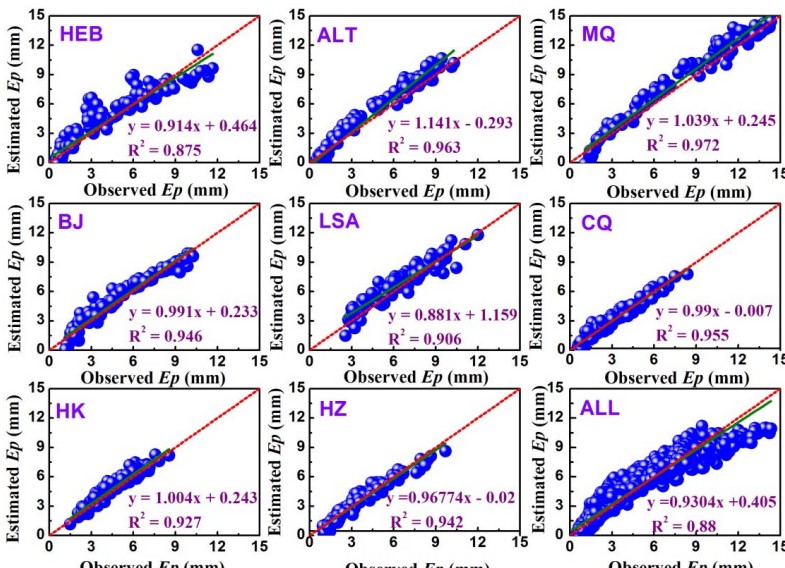

Fig.13. Comparison of the observed and estimated *Ep* using the optimal MLR model during
the testing period.
