# Peer review of "Comparison of six different soft computing methods in modeling evaporation in different climates"

_Hydrology and Earth System Sciences, 2016_

## Referee Comment (RC1) · N. Haughton (Referee) · 1 Jun 2016

To the Editor,

Thank you for the opportunity to review this paper.

The paper implements 8 different empirical model structures over for predicting pan evaporation at a variety of sites in China, and then evaluates the performances of these models against each other. The paper is not revolutionary, but does provide useful information about the performance of a number of advanced empirical models in a hydrological setting.

Over all, the paper is well written, and the intent and methods are very clear. However, the paper suffers from trying to show too much detail in the results section, and the

actual intercomparison results are not very clear. I think that these problems could be resolved but summarising the results in a neater fashion. Some specific suggestions are given below.

Cheers

ned haughton

**Major comments**

There is no mention of local optima problems associated with many soft computing methods. This should at least be acknowledged, any methods for avoiding these problems should be explained in the methods sections for each of the models.

The results section is long and repetitive. It would be good to try to summarise the data as much as possible, and draw a bit of a narrative through the results. What is the key message you're trying to communicate here? Some specific suggestions are given below in the Tables and Figures sections.

There is no discussion section. It would be good to have some general discussion of the generalisability of these results, and the implications for others working in the field.

**Minor comments**

l112-5: There is no justification given for the choice of these 8 models. Many more models are mentioned in the introduction. Why choose these 8 specifically?

l226: This sentence mentions a dataset, but this isn't actually described above.

l280-303: The important thing is that the sites are diverse. You are not trying to describe the sites, just use them to evaluate modes, yet lot of these statistics are just descriptive stats repeated from the table. Better would be to quote ranges (of means, variance, and extremes), and maybe try to relate those to global- or china-wide ranges, to show that the sites are representative.

Section 2.3: No rationale is given for the choice of metrics. All three metrics are highly correlated (all r»0.9), from what I can see, and therefore two metrics don't provide much more information after the first. Consider using alternate metrics, such as the Nash-Sutcliffe model efficiency coefficient, normalised mean error, correlation, or some of the metrics mentioned in Pachepsky et al. (2016)

l314-323: This section should be split up and moved into the relevant Methods subsections.

l331 (and below): How are these "accuracy ranks" calculated? They are not mentioned in the methods section at all. Perhaps they should be included in the tables?

l333-4: It is not clear that Ta and Rg are better at modelling Ep than RH or Ws, because RH and Ws are only included as fourth and fifth variables. If the inputs are highly correlated, then RH and Ws may also perform reasonably by themselves.

l413: the $R^2$ is not the same things as Pearson's Correlation Coefficient, except in the simple case of univariate linear regression. Also, the some of the correlations between Ws and Ep (I assume the R column in Table 1) are reasonably high, so it would be reasonable to assume some predictive power.

l445: The description of the generalised model should be moved to the methods section and expanded.

l477-480: Performance is not additive, especially when the predictor variables have significant covariance, so it is almost inevitable that RH and Ws will appear to be worse predictors relative to Rg and Ta, when they have only been included in models with multiple other variables.

l479: Ws doesn't decrease *all* simulation metric results, and again, it is not clear how this variable would perform as a predictor in the absence of other inputs, which are likely correlated.

**Tables**

Tables 1, 3-10, and 12: There is a *LOT* of data in all of the tables. It is very difficult to read information laid out like this. Consider using summary plots (possibly small multiples of parallel coordinate plots) instead and moving the tables to supplemental material, or colouring the table cells to give a clearer indication of performance (normalise colours per column).

Tables 3-10: There is no explanation given anywhere as to why there are two columns in tables 3-10 for each of the three metrics. Explain in-text, and in the table captions.

**Figures**

Figure 3: Colour the stars in the same colours as in Figure 4.

Figure 4: Put the legend outside above the graphs, make it larger.

Figures 6-13:

- The paper is about comparison between models, not sites. But I have to scroll between 8 pages to compare all of the models. It would be better to have a grid for each site, that included all 8 models.

- If you remove the x- and y-axis tags from all but the first row and column, you can save significant space, and probably fit all models on 2x4 grid, allowing more plots per page.

- The bubble effect only adds unnecessary visual detail. Remove the 3d effect, and use smaller circles, so the detail in the scatter plot can be seen properly.

- I guess that the scatter plots include the seasonal cycle. It may be useful to have corresponding residuals plots, to show under which conditions the modes are performing poorly.

- Units should be mm/day, I think.

**Technical notes**

l48: remove "and air".

l49-50: Pan evaporation is a measurement, it doesn't play a role in the ecosystem. Remove clause, or move to previous sentence.

l56: "..less _well_ understood.."

l65: remove "the"

l70: Full stop before "For example.."

l95: ".. in case of without local inputs and outputs" doesn't make sense. Re-word.

l97: remove first "the"

l98: "On the contrary" probably should be "In contrast"

l102: "at a few number of stations" makes no sense, re-write sentence, split at "for example".

l128: "_The_ MLP is _a_ well-known ..."

l129-30: "_hierarchical_ networks _consisting of_ several layers.."

l132: The neurons are the nodes. The connections are the synapses. Re-word sentence.

l149: "two types of neurons, S-summation and D-summation, which...".

l188: "MF" - abbreviation undefined.

l190: "RMSE" - abbreviation undefined.

l220: "variables are"

l225: "analyse"

l280: remove "It is clear that"

l296: "has lower skewness", I think.

l397: "indicate"

l475: "..MLP _performance was_ superior to.."

l476: Full stop after "stations".

l479: "_Decreased_" (past tense)

Table 1: Headers misaligned. R metric needs to be explained in footer. Also, it is probably better to sort by variable first, and then by station, so that stations can be compared. If you do this, add minor grid lines between variables.

**References**

Best, Martin J., Gab Abramowitz, H Johnson, et al. 2015The Plumbing of Land Surface Models: Benchmarking Model Performance. Journal of Hydrometeorology 16(3): 1425–1442. http://journals.ametsoc.org/doi/abs/10.1175/JHM-D-14-0158.1.

Pachepsky, Y. A., G. Martinez, F. Pan, T. Wagener, and T. Nicholson. "Evaluating Hydrological Model Performance Using Information Theory-Based Metrics." Hydrol. Earth Syst. Sci. Discuss. 2016 (February 15, 2016): 1–24. doi:10.5194/hess-2016-46.

---

## Referee Comment (RC2) · Anonymous Referee #2 · 18 Jun 2016

Summary

Wang et al. presented a pan evaporation modeling study using various statistical approaches. The overarching goal is to find "best" model for evaporation prediction. I believe that the results delivered in this study are significant, and may have broad impacts on e.g., reservoir water management, agriculture irrigation practice.

Strengths

1. This study comprehensively compared six soft computing methods and two regression methods.

2. Statistical models are applied at multiple sites across different climate regimes.

Weakness

1. Language: needs substantial editing efforts to ensure consistency and readability via improving e.g., wording, sentence structure, paragraph connection and cohesion.

2. Methodology: Training dataset is select randomly, however without ensemble the randomness of the data selection is still weak. Cross-site validation is also necessary before concluding which model is the "best" one.

Major comments

1. Models used 50% of data for training and the rest for testing (randomly chosen) at each study site. However, random sampling is not repeated to ensure generality of the model testing results. With only one random sampling of training dataset, the trained model is potentially biased to that particular case. What will happen, if use the testing data for training and training data for testing? Will the model predictability retain? In order to show the generality of the model performance, one has to do multiple ensembles of training data sampling and present the "mean model".

2. This study made lots of efforts on comparing different models and trying to find out the "best" one. The identification of the "best" model is based on within-site evaluation. In that sense, the significance of this study is highly limited. What happen if one would like to apply the "best" model to other sites with different climate, or do a regional modeling? This type of question could be answered by doing cross-site evaluation. For example, apply the trained model to other 7 sites and show the overall performance.

3. Results interpretation: Table, figures, and main text are heavily redundant, e.g., no need to repeat all the numbers (e.g., R2) in main text that have been included in table.

Specific comments

L24. The first sentence does not make too much sense, because this study focused on pan evaporation, which best inform water management such as agriculture. But for terrestrial ecological processes and regional climate change, evaporation is not as significant as it in agriculture. I would suggest the abstract starting with "Pan evaporation

plays . . . in informing . . . ". And followed with another sentence highlighting the fact that one of the basic challenges is modeling . . ...

L30. First time use Ep, better to define it earlier.

L31. No need to list all the climate variables. Maybe: "We develop, train, and validate the eight models at various sites crossing a wide range of climate".

L33. The first part of applications focused . . .. Remove this sentence, because the next sentence is actually presenting the accuracy comparison.

L38. Generalized models were also developed and tested. . ... Remove this sentence.

L42. BJ, CQ and HK station. Define the sites before use them.

L42. Recommendation or major implication based on this study is needed to end the abstract.

L48 "and air" -> "and air temperature"?

L50, roles in . . . -> roles in informing water resources redistribution and irrigation system design.

L55 is -> are

L56 one of the, remove; aspects in the hydrological cycle, remove

L57 to integrated -> for integrating

L59 some, remove

L63 for estimating Ep as a function of meteorological data -> to linking Ep to various meteorological drivers.

L64. But some of . . . -> but the applications of these techniques are often limited by data availability and completeness.

L68. What are conventional techniques, list a few.

L75 at a site in hot and dry climate -> at a hot and dry site

L76 is -> was

L70 ,for example -> .For example, . . . and from L70 – L77 replace ";" with "."

L110 provided an impetus -> impede

L111. Which provided an impetus for . . ., remove.

L112 "considering the importance of . . .or hydrological modeling", remove.

L116 in modeling Ep . . . -> in Ep modeling with different combinations of climate inputs.

L119 "using generalized . . . models" -> using eight different models.

L129. MLPs are organized as hierarchical networks with several layers

L133. its input -> input

L142. but they do not use-> without using

L143 The structure of, remove

L148 Two types of neurons (S-summation and D-summation) are connected to patter layer unit

L179 which is -> that is

L183 which can be used for optimization problems, remove

L193 which projects -> that projects

L196 efficient is enough, redundant to say quick, converging to global optimum.

L199 more simpler and more efficient

L200 This issue is caused by-> , due to the reason that LSSVM solves linear equations instead of a quadratic programming problem in SVM.

L204. This subject -> these models

L212 consists -> conssiting

L236 Why use monthly data? e.g., Goyal 2014 investigated various techniques to improve daily Ep in India. Is it possible to do daily at the eight selected sites as well?

L238 – 242. Site information are redundant, with figure 3. Or could be just put in a table.

L242-258. Put the site description (lon, lat, alt, mean annual temperature, mean annual precipitation) in a table. In the text, highlight the most important fact, do not literally reiterate the site information.

L280-286. Again, too many numbers in this section (which have already been shown in Table 1). Just illustrate the most important fact, e.g., which site has the highest Ep, why and how?

L305. ANFIS, GP, . . ., and SS. -> six soft computing and two regression models

L314-317 This study . . . in the application. Remove. Just present results and discussion, no need to reiterate what have been done.

L329. Models with full weather data have the best accuracy. This seems obvious but indeed has significant implications. As using more and more predictor variables, the response variables could be commonly better predicted. However, the issue is the expenditure. In this case, is the data availability. Variables like air temperature is relatively easy to measure and important for evaporation, we definitely want to include it in ep modeling. However, is there a predictor variable that is relatively hard to measure (unavailable) but is "must be include" predictor variable? Is it necessary to use the full model for large-scale (regional) prediction? If so, is this conclusion also valid at other study sites?

L367. The best accuracy were generally obtained form five-input models and GNRR

model perform better. It is excited to see that a certain model stand out. But it would be helpful, it one can go one step further and try to figure out the underlying reason why that model is "best". What kind of feature of that model could possibly lead to the success?

L414. It is obvious that ..... Throughout the paper, many places used this sentence structure "it is obvious that". Try to avoid if necessary, because it imply that if one can not immediately understand the results then he is stupid. Furthermore, sometime the results are not that obvious and it is always authors' responsibility to help the readers understand those results.

---

## Referee Comment (RC3) · C. Jimenez (Referee) · 12 Jul 2016

C. Jimenez (Referee)

carlos.jimenez@obspm.fr

The paper estimates pan evaporation by statistical models calibrated at 8 stations in China and briefly discusses the choice of climatic variables to drive the models. The paper can potentially be an interesting contribution to the field, but in my opinion it requires a revision to make it more attractive to the readers. I concur with most of the comments expressed by the other two reviewers, so I will not list any specific issues. A few things worth stressing on my view are:

(1) Statistical models. MLPs are in many cases used by default as the "soft computing" technique to statistically approximate geophysical relationships, so it is not a surprise to me that they came as the winner. Other methods may compete with the MLPs, but in my experience they are one of the best compromises between implementation

complexity and capacity to approximate mappings. The problem is that the MLPs are also well known for over-fitting issues, and as there is no cross-correlation tests in the paper (i.e., calibration and error performance on different stations), we may wonder if over-fitting may be playing a role here (other methods may be more robust in this sense, but will be penalized by showing a worst performance on the validation dataset). This needs to be discussed.

(2) Climate drivers. It may have been better to start by looking at the linear correlation between the climate drivers and the Ep at the different stations, to rank the relative importance of the drivers in a simple way. That could have been used to justify why RH and WS are only tested as part of the final combination of drivers to the models, and perhaps be used to reduce the number of driver-combinations to be tested. In my experience, the mapping between drivers and geophysical parameter has to be very non-linear for the linear correlations to differ significantly from a "correlation" inferred by applying first a non-linear estimator.

(3) Applications. My reading of the paper is that I better use MLPs to statistically mode Ep, but perhaps not for all climates. OK, but not sure whether that is a clear message to pass. I any case, I would suggest to go a bit further and use the constructed database to provide a more general model that it is not restricted to a specific climate type. This could have been tested by investigating cross-correlations (i.e., how a model trained in a station performs at a different station), and/or by calibrating the model with a database containing data from all stations. It is quite likely that the more general model cannot outperform the individual-station best model, but if the differences are reasonable that single-model could potentially be used to generate Ep over most China when driven with remote sensing data. To me, that would be an excellent outcome of the paper and of more utility than just showing that at a specific station one statistical model performs better than the others.

(4) Format. The paper, even with its current contents, needs a better way to present the results. There are currently more than 3000 numbers scattered around 12 tables

and around 70 scatter plots, with only one table ranking the models summarizing the main results. That material could be part of an appendix, but figures and/or tables synthesizing the main results are needed.

---

## Author Comment (AC1) · 8 Aug 2016

**Major comments There is no mention of local optima problems associated with many soft computing methods. This should at least be acknowledged, any methods for avoiding these problems should be explained in the methods sections for each of the models.**

Reply: as suggested, in the methods section, some statements were added for local optima problems of the applied models. We further evaluated the applied models with full weather inputs by changing training and testing period. Please see Table 14 for the results.

The results section is long and repetitive. It would be good to try to summarize the data as much as possible, and draw a bit of a narrative through the results. What is the key

message you're trying to communicate here? Some specific suggestions are given below in the Tables and Figures sections.

Reply: results section was revised and a discussion section was added.

There is no discussion section. It would be good to have some general discussion of the generalizability of these results, and the implications for others working in the field.

Reply: a discussion section was added as suggested (discussion of the generalizability of these results, and the implications for others working in the field).

**Minor comments l112-5: There is no justification given for the choice of these 8 models. Many more models are mentioned in the introduction. Why choose these 8 specifically?**

Reply: the specific reasons for selecting each soft computing model was provided in methods section as suggested.

l226: This sentence mentions a dataset, but this isn't actually described above.

Reply: In this study, MLR models are developed using the same dataset at eight stations in different climates (described in section 2.2) which was used to train and test the above soft computing models.)

l280-303: The important thing is that the sites are diverse. You are not trying to describe the sites, just use them to evaluate modes, yet lot of these statistics are just descriptive stats repeated from the table. Better would be to quote ranges (of means, variance, and extremes), and maybe try to relate those to global- or china-wide ranges, to show that the sites are representative.

Reply: we have added "For example, the monthly mean Ep ranged from 2.86 (CQ) to 7.26 mm (MQ) with associated Cv values changing from 0.32 to 0.75 in different climates; the minimum monthly Ep was about 0.15 mm while the maximum monthly Ep

reached 15.89 mm. Accordingly, the monthly meteorological variables varied greatly for each station in different climates, for example, the monthly mean Rg at CQ was about 8.8 MJ m-2, while the Rg values increased to 20.41 MJ m-2at LSA station; the monthly mean Ta also changed from 4.17oC to 24.08oC. The diverse climatic characteristics can also been seen from the statistics of Cv, Cx, xmin and xmax for each parameters at above stations. The monthly variations of Ep and associated climatic parameters in each station were further illustrated in Fig.5, for example, the monthly Ep, Ta, Hs and Rg are generally higher in summer and lower in winter months and there were also differences for each parameter in different stations, which indicated that above stations are representative for studying Ep in different climates.)" into the Case Study and Data section

Section 2.3: No rationale is given for the choice of metrics. All three metrics are highly correlated (all r⩾0.9), from what I can see, and therefore two metrics don't provide much more information after the ﬡrst. Consider using alternate metrics, such as the Nash-Sutcliffe model efﬡciency coefﬡcient, normalised mean error, correlation, or some of the metrics mentioned in Pachepsky et al. (2016).

Reply: as suggested Nash-Sutcliffe criterion was also included in the revised paper

l314-323: This section should be split up and moved into the relevant Methods subsections.

Reply: It was done as suggested.

l331 (and below): How are these "accuracy ranks" calculated? They are not mentioned in the methods section at all. Perhaps they should be included in the tables?

Reply: this was mentioned in the Results section.

l333-4: It is not clear that Ta and Rg are better at modelling Ep than RH or Ws, because RH and Ws are only included as fourth and ﬡfth variables. If the inputs are highly correlated, then RH and Ws may also perform reasonably by themselves.

Reply: we have also added two input combinations as iv) RH and v) Ws. Now, the effect of each variable on Ep can be clearly seen.

l413: the RĔȨ2 is not the same things as Pearson's Correlation Coefficient, except in the simple case of univariate linear regression. Also, the some of the correlations between Ws and Ep (I assume the R column in Table 1) are reasonably high, so it would be reasonable to assume some predictive power.

Reply: we agree with you that the R indicates that the Ws have reasonable and predictive power. Therefore, we have added "In overall, soft computing models with full weather data (Rg, Ta, Hs, RH and Ws) generally had the best accuracy. This indicates that all these variables are required for better Ep estimation. It can be seen from the applications that adding RH or Ws inputs into the applied models generally increase their accuracies in predicting Ep in all stations even though these parameters have the lowest correlation with Ep (see Table 1)." into the discussion section.

l445: The description of the generalized model should be moved to the methods section and expanded.

Reply: It was done as suggested

l477-480: Performance is not additive, especially when the predictor variables have significant covariance, so it is almost inevitable that RH and Ws will appear to be worse predictors relative to Rg and Ta, when they have only been included in models with multiple other variables.

Reply: we have added two input combinations as iv) RH and v) Ws.

l479: Ws doesn't decrease *all* simulation metric results, and again, it is not clear how this variable would perform as a predictor in the absence of other inputs, which are likely correlated.

Reply: we have added two input combinations as iv) RH and v) Ws.

**Tables Tables 1, 3-10, and 12: There is a *LOT* of data in all of the tables. It is very difficult to read information laid out like this. Consider using summary plots (possibly small multiples of parallel coordinate plots) instead and moving the tables to supplemental material, or colouring the table cells to give a clearer indication of performance (normalize colours per column).**

Reply: we have showed the best models by bold numbers.

Tables 3-10: There is no explanation given anywhere as to why there are two columns in tables 3-10 for each of the three metrics. Explain in-text, and in the table captions.

Reply: we have explained these as Training and Testing period

**Figures Figure 3: Colour the stars in the same colours as in Figure 4.**

Reply: It has been corrected.

Figure 4: Put the legend outside above the graphs, make it larger.

Reply: It has been corrected.

Figures 6-13: - The paper is about comparison between models, not sites. But I have to scroll between 8 pages to compare all of the models. It would be better to have a grid for each site that included all 8 models.

Reply: It has been corrected.

- If you remove the x- and y-axis tags from all but the first row and column, you can save significant space, and probably fit all models on 2x4 grid, allowing more plots per page.

Reply: It has been corrected.

- The bubble effect only adds unnecessary visual detail. Remove the 3d effect, and use smaller circles, so the detail in the scatter plot can be seen properly.

Reply: It has been corrected.

- I guess that the scatter plots include the seasonal cycle. It may be useful to have corresponding residuals plots, to show under which conditions the modes are performing poorly. - Units should be mm/day, I think.

Reply: It has been corrected.

**Technical notes l48: remove "and air". l49-50: Pan evaporation is a measurement, it doesn't play a role in the ecosystem. Remove clause, or move to previous sentence. l56: "..less _well_ understood.." l65: remove "the" l70: Full stop before "For example.." l95: "..in case of without local inputs and outputs" doesn't make sense. Reword. l97: remove first "the" l98: "On the contrary" probably should be "In contrast" l102: "at a few number of stations" makes no sense, re-write sentence, split at "forexample". l128: "_The_ MLP is _a_ well-known ..." l129-30: "_hierarchical_ networks _consisting of_ several layers.." l132: The neurons are the nodes. The connections are the synapses. Re-word sentence. l149: "two types of neurons, S-summation and D-summation, which...". l188: "MF" - abbreviation undefined. l190: "RMSE" - abbreviation undefined. l220: "variables are" l225: "analyse" l280: remove "It is clear that" l296: "has lower skewness", I think. l397: "indicate" l475: "..MLP _performance was_ superior to.." l476: Full stop after "stations". l479: "_Decreased_" (past tense) Table 1: Headers misaligned. R metric needs to be explained in footer. Also, it isprobably better to sort by variable first, and then by station, so that stations can becompared. If you do this, add minor grid lines between variables.**

Reply: All these have been considered in the revised paper.

Please also note the supplement to this comment:
http://www.hydrol-earth-syst-sci-discuss.net/hess-2016-247/hess-2016-247-AC1-supplement.pdf

———————————————

[Figure]

Fig. 1. Schematic architecture of: a) MLP neural network; b) GRNN.

[Figure]

[Figure]

**Fig. 3.**

[Figure]

**Fig. 4.**

[Figure]

**Fig. 5.**

[Figure]

**Fig. 6.**

[Figure]

**Fig. 7.**

[Figure]

**Fig. 8.**

[Figure]

**Fig. 9.**

[Figure]

**Fig. 10.**

[Figure]

**Fig. 11.**

[Figure]

**Fig. 12.**

[Figure]

**Fig. 13.**

**Supplement:**

Table 1.The geographical locations and associated annual climatic parameters

| Station | Longitude | Latitude | Altitude (m) | Temperature (°C) | Precipitation (mm) |
|---|---|---|---|---|---|
| HEB | 126°46′E | 45°45′N | 142.3 | 4.17 | 524.3 |
| ALT | 108°05′E | 47°44′N | 735.3 | 4.54 | 191.3 |
| MQ | 103°05′E | 38°38′N | 1367 | 8.33 | 113 |
| BJ | 116°28′E | 39°48′N | 31.3 | 12.20 | 571.9 |
| LSA | 91°08′E | 29°40′N | 3648.7 | 7.82 | 426.4 |
| CQ | 106°28′E | 29°35′N | 259.1 | 18.04 | 1104.5 |
| HZ | 120°10′E | 30°14′N | 41.7 | 16.45 | 1454.6 |
| HK | 110°21′E | 20°02′N | 13.9 | 24.08 | 1651.9 |

Table 2.Monthly statistical parameters of each data set for each station

| Station | Dataset | $x_{mean}$ | $S_x$ | $C_v$ | $C_x$ | $x_{min}$ | $x_{max}$ | $R$ |
|---|---|---|---|---|---|---|---|---|
| HEB | Rg | 12.98 | 5.35 | 0.41 | 0.00 | 3.68 | 28.71 | 0.89 |
| | Ta | 4.17 | 14.52 | 3.48 | -0.25 | -24.71 | 25.25 | 0.86 |
| | Hs | 7.02 | 1.59 | 0.23 | -0.25 | 2.82 | 10.89 | 0.79 |
| | RH | 65.44 | 11.01 | 0.17 | -0.44 | 36.23 | 85.06 | -0.36 |
| | Ws | 3.69 | 0.97 | 0.26 | 0.61 | 1.88 | 6.69 | 0.26 |
| | $E_P$ | 4.35 | 3.27 | 0.75 | 0.44 | 0.16 | 12.96 | 1 |
| ALT | Rg | 15.13 | 7.21 | 0.48 | -0.06 | 2.34 | 27.69 | 0.92 |
| | Ta | 4.54 | 13.95 | 3.07 | -0.25 | -25.08 | 24.87 | 0.93 |
| | Hs | 8.2 | 2.52 | 0.31 | -0.25 | 1.92 | 12.66 | 0.90 |
| | RH | 57.99 | 13.41 | 0.23 | 0 | 30.1 | 86.77 | -0.89 |
| | Ws | 2.40 | 0.99 | 0.41 | 0.05 | 0.31 | 5.46 | 0.69 |
| | $E_P$ | 4.72 | 3.84 | 0.81 | 0.33 | 0.15 | 13.79 | 1 |
| MQ | Rg | 16.41 | 4.98 | 0.30 | 0.07 | 7.21 | 26.9 | 0.92 |
| | Ta | 8.33 | 11.32 | 1.36 | -0.19 | -15.46 | 25.72 | 0.93 |
| | Hs | 8.37 | 1.12 | 0.13 | 0.30 | 5.47 | 11.38 | 0.72 |
| | RH | 44.82 | 9.06 | 0.2 | 0.12 | 24.3 | 74.58 | -0.29 |
| | Ws | 2.68 | 0.55 | 0.20 | 0.08 | 1.23 | 4.32 | 0.55 |
| | $E_P$ | 7.26 | 4.45 | 0.61 | 0.10 | 0.42 | 15.89 | 1 |
| BJ | Rg | 14.61 | 4.94 | 0.34 | 0.05 | 5.14 | 25.59 | 0.91 |
| | Ta | 12.20 | 10.74 | 0.88 | -0.17 | -7.6 | 29.56 | 0.75 |
| | Hs | 7.41 | 1.42 | 0.19 | 0.06 | 3.79 | 11.21 | 0.76 |
| | RH | 57.29 | 13.70 | 0.24 | 0.02 | 21.86 | 85.52 | 0.09 |
| | Ws | 2.50 | 0.67 | 0.27 | 0.49 | 1.07 | 4.65 | 0.14 |
| | $E_P$ | 5.09 | 2.83 | 0.56 | 0.70 | 0.85 | 15.63 | 1 |
| LSA | Rg | 20.41 | 4.20 | 0.21 | 0.11 | 10.39 | 30.69 | 0.68 |
| | Ta | 7.82 | 6.37 | 0.81 | -0.21 | -5.16 | 18.19 | 0.75 |
| | Hs | 8.19 | 0.96 | 0.12 | -0.59 | 4.66 | 10.55 | 0.18 |
| | RH | 44.39 | 15.10 | 0.34 | 0.30 | 15.36 | 76.61 | 0.19 |
| | Ws | 1.90 | 0.46 | 0.24 | 0.30 | 0.92 | 3.41 | 0.34 |
| | $E_P$ | 6.35 | 2.23 | 0.35 | 0.36 | 2.15 | 13.28 | 1 |
| CQ | Rg | 8.80 | 4.69 | 0.53 | 0.43 | 0 | 21.32 | 0.92 |
| | Ta | 17.93 | 7.46 | 0.42 | -0.10 | 0.64 | 30.90 | 0.85 |
| | Hs | 2.83 | 2.02 | 0.71 | 0.91 | 0 | 9.19 | 0.94 |
| | RH | 79.15 | 8.55 | 0.11 | -4.66 | 6.97 | 90.30 | -0.40 |
| | Ws | 1.36 | 0.34 | 0.25 | -0.12 | 0.64 | 2.13 | 0.58 |
| | $E_P$ | 2.86 | 1.94 | 0.68 | 0.87 | 0.54 | 9.32 | 1 |
| HZ | Rg | 11.63 | 4.20 | 0.36 | 0.54 | 3.93 | 24.83 | 0.94 |
| | Ta | 16.45 | 8.46 | 0.51 | -0.06 | -0.01 | 31.03 | 0.88 |
| | Hs | 4.99 | 1.74 | 0.35 | 0.63 | 1.19 | 11.25 | 0.80 |
| | RH | 78.04 | 5.63 | 0.07 | -0.80 | 53.74 | 90.42 | -0.04 |
| | Ws | 2.24 | 0.43 | 0.19 | 0.05 | 1.01 | 3.58 | 0.13 |
| | $E_P$ | 3.65 | 1.94 | 0.53 | 0.84 | 0.74 | 11.33 | 1 |
| HK | Rg | 13.86 | 4.33 | 0.31 | -0.05 | 4.06 | 24.34 | 0.90 |

| | | | | | | |
|---|---|---|---|---|---|---|
| ***Ta*** | 24.08 | 4.07 | 0.17 | -0.55 | 13.21 | 29.83 | 0.81 |
| ***Hs*** | 5.83 | 1.96 | 0.34 | -0.26 | 0.47 | 9.94 | 0.89 |
| ***RH*** | 84.14 | 3.61 | 0.04 | -0.52 | 71.39 | 94.46 | -0.41 |
| ***Ws*** | 2.65 | 0.66 | 0.25 | 0.61 | 1.33 | 4.98 | 0.04 |
| ***$E_P$*** | 5.00 | 1.59 | 0.32 | 0.08 | 1.37 | 9.97 | 1 |

The unit of *Rg, Ta, Pa, Ws* and *$E_P$* are MJ m$^{-2}$, ℃,hPa,ms$^{-1}$ and mm/day, respectively; $x_{mean}$, $S_x$, $C_v$, $C_x$, $x_{min}$ and $x_{max}$ denote the mean, standard deviation, variation coefficient, skewness, minimum and maximum values, respectively.

Table 3.The input combinations for different artificial intelligence techniques.

| Models | | | | | | Input combinations |
|---|---|---|---|---|---|---|
| ANFIS-GP | FG | GRNN | LSSVM | MARS | MLP | |
| ANFIS-GP1 | FG1 | GRNN1 | LSSVM1 | MARS1 | MLP1 | ***Rg*** |
| ANFIS-GP2 | FG2 | GRNN2 | LSSVM2 | MARS2 | MLP2 | ***Ta*** |
| ANFIS-GP3 | FG3 | GRNN3 | LSSVM3 | MARS3 | MLP3 | ***Hs*** |
| ANFIS-GP4 | FG4 | GRNN4 | LSSVM4 | MARS4 | MLP4 | ***RH*** |
| ANFIS-GP5 | FG5 | GRNN5 | LSSVM5 | MARS5 | MLP5 | ***Ws*** |
| ANFIS-GP6 | FG6 | GRNN6 | LSSVM6 | MARS6 | MLP6 | ***Rg, Ta*** |
| ANFIS-GP7 | FG7 | GRNN7 | LSSVM7 | MARS7 | MLP7 | ***Rg, Ta, Hs*** |
| ANFIS-GP8 | FG8 | GRNN8 | LSSVM8 | MARS8 | MLP8 | ***Rg, Ta, Hs, RH*** |
| ANFIS-GP9 | FG9 | GRNN9 | LSSVM9 | MARS9 | MLP9 | ***Rg, Ta, Hs, RH, Ws*** |

Table 4. Comparisons of different models for predicting *Ep* at HEB station.

| HEB | Training | | | | Testing | | | |
|---|---|---|---|---|---|---|---|---|
| | MAE | RMSE | $R^2$ | *E* | MAE | RMSE | $R^2$ | *E* |
| ANFIS-GP1 | 1.062 | 1.411 | 0.815 | 0.815 | 1.044 | 1.431 | 0.819 | 0.805 |
| ANFIS-GP2 | 1.226 | 1.68 | 0.737 | 0.737 | 1.082 | 1.471 | 0.797 | 0.794 |
| ANFIS-GP3 | 1.589 | 2.05 | 0.609 | 0.609 | 1.496 | 1.834 | 0.726 | 0.68 |
| ANFIS-GP4 | 2.681 | 2.972 | 0.178 | 0.178 | 2.862 | 3.171 | 0.071 | 0.044 |
| ANFIS-GP5 | 2.754 | 3.137 | 0.085 | 0.085 | 2.809 | 3.340 | 0.089 | -0.061 |
| ANFIS-GP6 | 0.865 | 1.225 | 0.86 | 0.86 | 0.781 | 1.089 | 0.894 | 0.887 |
| ANFIS-GP7 | 0.785 | 1.167 | 0.873 | 0.873 | 0.645 | 0.907 | 0.923 | 0.922 |
| ANFIS-GP8 | 0.429 | 0.601 | 0.966 | 0.966 | 0.517 | 0.751 | 0.956 | 0.946 |
| **ANFIS-GP9** | **0.378** | **0.521** | **0.975** | **0.975** | **0.431** | **0.600** | **0.967** | **0.966** |
| FG1 | 1.031 | 1.371 | 0.825 | 0.825 | 1.031 | 1.507 | 0.816 | 0.765 |
| FG2 | 1.151 | 1.632 | 0.752 | 0.752 | 1.077 | 1.502 | 0.786 | 0.786 |
| FG3 | 1.528 | 2.008 | 0.625 | 0.625 | 1.354 | 1.798 | 0.74 | 0.696 |
| FG4 | 2.487 | 2.877 | 0.23 | 0.23 | 2.677 | 3.083 | 0.118 | 0.096 |
| FG5 | 2.708 | 3.103 | 0.104 | 0.104 | 2.806 | 3.304 | 0.091 | -0.039 |
| FG6 | 0.719 | 1.071 | 0.893 | 0.893 | 0.688 | 1.178 | 0.891 | 0.870 |
| FG7 | 0.67 | 1.002 | 0.907 | 0.907 | 0.673 | 1.059 | 0.897 | 0.824 |
| FG8 | 0.39 | 0.563 | 0.971 | 0.97 | 0.45 | 0.638 | 0.969 | 0.961 |
| **FG9** | **0.325** | **0.451** | **0.981** | **0.981** | **0.421** | **0.554** | **0.971** | **0.971** |
| GRNN1 | 1.026 | 1.364 | 0.827 | 0.827 | 1.031 | 1.479 | 0.814 | 0.792 |
| GRNN2 | 1.138 | 1.617 | 0.757 | 0.757 | 1.054 | 1.472 | 0.796 | 0.794 |
| GRNN3 | 1.519 | 2 | 0.628 | 0.627 | 1.379 | 1.816 | 0.738 | 0.686 |
| GRNN4 | 2.451 | 2.835 | 0.253 | 0252 | 2.685 | 3.087 | 0.113 | 0.094 |
| GRNN5 | 2.72 | 3.097 | 0.122 | 0.108 | 2.795 | 3.272 | 0.085 | -0.018 |
| GRNN6 | 0.549 | 0.897 | 0.926 | 0.925 | 0.734 | 1.218 | 0.878 | 0.859 |
| GRNN7 | 0.453 | 0.71 | 0.954 | 0.953 | 0.696 | 1.124 | 0.887 | 0.88 |
| GRNN8 | 0.155 | 0.246 | 0.994 | 0.994 | 0.543 | 0.962 | 0.922 | 0.912 |
| **GRNN9** | **0.047** | **0.09** | **0.999** | **0.999** | **0.529** | **0.856** | **0.932** | **0.930** |
| LSSVM1 | 1.027 | 1.371 | 0.825 | 0.825 | 1.02 | 1.461 | 0.819 | 0.797 |
| LSSVM2 | 1.131 | 1.619 | 0.756 | 0.756 | 1.059 | 1.487 | 0.791 | 0.79 |
| LSSVM3 | 1.684 | 2.133 | 0.604 | 0.577 | 1.556 | 1.949 | 0.712 | 0.639 |
| LSSVM4 | 2.493 | 2.876 | 0.231 | 0.231 | 2.685 | 3.094 | 0.113 | 0.09 |
| LSSVM5 | 2.736 | 3.117 | 0.097 | 0.096 | 2.806 | 3.320 | 0.091 | -0.048 |
| LSSVM6 | 0.838 | 1.205 | 0.866 | 0.865 | 0.79 | 1.169 | 0.879 | 0.87 |
| LSSVM7 | 0.901 | 1.267 | 0.853 | 0.851 | 0.761 | 1.031 | 0.9 | 0.899 |
| LSSVM8 | 0.813 | 1.128 | 0.893 | 0.882 | 0.826 | 1.090 | 0.893 | 0.887 |
| **LSSVM9** | **0.483** | **0.667** | **0.960** | **0.959** | **0.589** | **0.766** | **0.959** | **0.944** |
| MARS1 | 1.038 | 1.371 | 0.825 | 0.825 | 1.064 | 1.581 | 0.805 | 0.762 |
| MARS2 | 1.088 | 1.543 | 0.779 | 0.779 | 1.093 | 1.563 | 0.771 | 0.768 |
| MARS3 | 1.537 | 2.01 | 0.624 | 0.624 | 1.369 | 1.795 | 0.744 | 0.694 |
| MARS4 | 2.457 | 2.852 | 0.243 | 0.243 | 2.731 | 3.133 | 0.103 | 0.066 |
| MARS5 | 2.695 | 3.079 | 0.118 | 0.118 | 2.795 | 3.303 | 0.097 | -0.037 |
| MARS6 | 0.659 | 0.972 | 0.912 | 0.912 | 0.806 | 1.390 | 0.861 | 0.816 |
| MARS7 | 0.659 | 0.972 | 0.912 | 0.912 | 0.806 | 1.390 | 0.861 | 0.816 |
| MARS8 | 0.543 | 0.708 | 0.953 | 0.953 | 0.597 | 0.933 | 0.935 | 0.917 |

| | MAE | RMSE | R² | E | MAE | RMSE | R² | E |
|---|---|---|---|---|---|---|---|---|
| **MARS9** | **0.50** | **0.635** | **0.962** | **0.962** | **0.570** | **0.749** | **0.950** | **0.947** |
| MLP1 | 1.044 | 1.374 | 0.824 | 0.824 | 1.03 | 1.483 | 0.818 | 0.794 |
| MLP2 | 1.082 | 1.567 | 0.771 | 0.771 | 1.03 | 1.490 | 0.792 | 0.791 |
| MLP3 | 1.135 | 1.618 | 0.757 | 0.757 | 1.04 | 1.460 | 0.798 | 0.797 |
| MLP4 | 2.539 | 2.893 | 0.221 | 0.221 | 2.729 | 3.107 | 0.108 | 0.082 |
| MLP5 | 2.711 | 3.107 | 0.102 | 0.102 | 2.807 | 3.304 | 0.090 | -0.038 |
| MLP6 | 0.655 | 0.963 | 0.914 | 0.914 | 0.716 | 1.148 | 0.892 | 0.891 |
| MLP7 | 0.608 | 0.908 | 0.923 | 0.923 | 0.584 | 0.879 | 0.928 | 0.923 |
| MLP8 | 0.314 | 0.458 | 0.98 | 0.98 | 0.409 | 0.607 | 0.970 | 0.966 |
| **MLP9** | **0.279** | **0.398** | **0.985** | **0.985** | **0.314** | **0.405** | **0.988** | **0.984** |
| SS | 0.954 | 1.327 | 0.838 | 0.838 | 0.822 | 1.152 | 0.886 | 0.885 |
| MLR | 0.825 | 1.05 | 0.897 | 0.897 | 0.874 | 1.160 | 0.875 | 0.875 |

Table 5. Comparisons of different models for predicting *Ep* at ALT station.

| ALT | Training | | | | Testing | | | |
|---|---|---|---|---|---|---|---|---|
| | MAE | RMSE | $R^2$ | E | MAE | RMSE | $R^2$ | E |
| ANFIS-GP1 | 1.19 | 1.597 | 0.841 | 0.841 | 1.003 | 1.268 | 0.896 | 0.848 |
| ANFIS-GP2 | 1.19 | 1.506 | 0.859 | 0.859 | 1.110 | 1.435 | 0.884 | 0.806 |
| ANFIS-GP3 | 1.345 | 1.763 | 0.807 | 0.807 | 1.214 | 1.601 | 0.844 | 0.758 |
| ANFIS-GP4 | 1.364 | 1.732 | 0.813 | 0.813 | 1.269 | 1.632 | 0.769 | 0.749 |
| ANFIS-GP5 | 2.354 | 2.848 | 0.495 | 0.495 | 2.161 | 2.578 | 0.379 | 0.373 |
| ANFIS-GP6 | 0.535 | 0.786 | 0.962 | 0.962 | 0.707 | 1.013 | 0.973 | 0.903 |
| ANFIS-GP7 | 0.494 | 0.737 | 0.966 | 0.966 | 0.691 | 1.012 | 0.977 | 0.903 |
| ANFIS-GP8 | 0.286 | 0.411 | 0.99 | 0.99 | 0.398 | 0.586 | 0.984 | 0.968 |
| **ANFIS-GP9** | **0.241** | **0.351** | **0.992** | **0.992** | **0.371** | **0.545** | **0.987** | **0.972** |
| FG1 | 1.079 | 1.398 | 0.878 | 0.878 | 0.994 | 1.300 | 0.891 | 0.838 |
| FG2 | 0.78 | 1.065 | 0.929 | 0.929 | 0.953 | 1.29 | 0.928 | 0.846 |
| FG3 | 1.052 | 1.375 | 0.882 | 0.882 | 1.002 | 1.328 | 0.92 | 0.835 |
| FG4 | 1.251 | 1.682 | 0.824 | 0.824 | 1.183 | 1.714 | 0.745 | 0.724 |
| FG5 | 2.237 | 2.79 | 0.515 | 0.515 | 2.099 | 2.554 | 0.404 | 0.386 |
| FG6 | 0.45 | 0.703 | 0.969 | 0.969 | 0.670 | 1.027 | 0.971 | 0.899 |
| FG7 | 0.49 | 0.717 | 0.968 | 0.968 | 0.697 | 1.043 | 0.971 | 0.898 |
| FG8 | 0.266 | 0.38 | 0.991 | 0.991 | 0.391 | 0.575 | 0.987 | 0.967 |
| **FG9** | **0.253** | **0.343** | **0.993** | **0.993** | **0.394** | **0.570** | **0.988** | **0.969** |
| GRNN1 | 1.052 | 1.38 | 0.882 | 0.881 | 0.951 | 1.229 | 0.90 | 0.858 |
| GRNN2 | 0.784 | 1.08 | 0.928 | 0.927 | 0.931 | 1.249 | 0.931 | 0.853 |
| GRNN3 | 1.041 | 1.368 | 0.884 | 0.883 | 0.977 | 1.290 | 0.92 | 0.843 |
| GRNN4 | 1.279 | 1.688 | 0.823 | 0.823 | 1.152 | 1.631 | 0.763 | 0.749 |
| GRNN5 | 2.439 | 2.859 | 0.512 | 0.491 | 2.239 | 2.560 | 0.397 | 0.382 |
| GRNN6 | 0.431 | 0.683 | 0.971 | 0.971 | 0.657 | 0.987 | 0.975 | 0.908 |
| GRNN7 | 0.384 | 0.629 | 0.975 | 0.975 | 0.705 | 1.015 | 0.978 | 0.903 |
| **GRNN8** | **0.173** | **0.278** | **0.995** | **0.995** | **0.468** | **0.658** | **0.988** | **0.959** |
| GRNN9 | 0.095 | 0.165 | 0.998 | 0.998 | 0.48 | 0.683 | 0.984 | 0.956 |
| LSSVM1 | 1.075 | 1.392 | 0.879 | 0.879 | 0.993 | 1.285 | 0.894 | 0.844 |
| LSSVM2 | 0.765 | 1.047 | 0.932 | 0.932 | 0.96 | 1.299 | 0.926 | 0.84 |

| | | | | | | | | |
|---|---|---|---|---|---|---|---|---|
| LSSVM3 | 1.447 | 1.839 | 0.827 | 0.789 | 1.034 | 1.391 | 0.863 | 0.817 |
| LSSVM4 | 1.278 | 1.692 | 0.822 | 0.822 | 1.192 | 1.668 | 0.753 | 0.738 |
| LSSVM5 | 2.335 | 2.814 | 0.508 | 0.507 | 2.189 | 2.565 | 0.391 | 0.379 |
| LSSVM6 | 0.520 | 0.769 | 0.964 | 0.963 | 0.673 | 0.954 | 0.973 | 0.914 |
| LSSVM7 | 0.617 | 0.868 | 0.955 | 0.953 | 0.749 | 0.945 | 0.973 | 0.916 |
| LSSVM8 | 0.709 | 0.927 | 0.953 | 0.946 | 0.63 | 0.770 | 0.968 | 0.944 |
| **LSSVM9** | **0.356** | **0.53** | **0.983** | **0.982** | **0.48** | **0.650** | **0.986** | **0.960** |
| MARS1 | 1.032 | 1.371 | 0.883 | 0.883 | 0.956 | 1.269 | 0.899 | 0.848 |
| MARS2 | 0.748 | 1.029 | 0.934 | 0.934 | 0.927 | 1.280 | 0.929 | 0.832 |
| MARS3 | 1.043 | 1.356 | 0.886 | 0.886 | 1.043 | 1.367 | 0.916 | 0.824 |
| MARS4 | 1.216 | 1.645 | 0.832 | 0.832 | 1.159 | 1.715 | 0.743 | 0.723 |
| MARS5 | 2.161 | 2.714 | 0.542 | 0.542 | 2.328 | 2.858 | 0.292 | 0.229 |
| MARS6 | 0.437 | 0.657 | 0.973 | 0.973 | 0.641 | 0.996 | 0.975 | 0.906 |
| MARS7 | 0.438 | 0.658 | 0.973 | 0.973 | 0.644 | 1.000 | 0.975 | 0.906 |
| MARS8 | 0.29 | 0.411 | 0.989 | 0.989 | 0.428 | 0.655 | 0.985 | 0.96 |
| **MARS9** | **0.276** | **0.382** | **0.991** | **0.991** | **0.403** | **0.622** | **0.987** | **0.964** |
| MLP1 | 1.03 | 1.363 | 0.884 | 0.884 | 0.951 | 1.268 | 0.897 | 0.849 |
| MLP2 | 0.787 | 1.07 | 0.929 | 0.929 | 0.951 | 1.282 | 0.93 | 0.845 |
| MLP3 | 0.752 | 1.039 | 0.933 | 0.933 | 0.93 | 1.280 | 0.929 | 0.846 |
| MLP4 | 1.25 | 1.683 | 0.824 | 0.824 | 1.183 | 1.699 | 0.751 | 0.727 |
| MLP5 | 2.266 | 2.805 | 0.51 | 0.51 | 2.089 | 2.542 | 0.405 | 0.391 |
| MLP6 | 0.445 | 0.689 | 0.97 | 0.97 | 0.667 | 1.033 | 0.971 | 0.900 |
| MLP7 | 0.521 | 0.769 | 0.963 | 0.963 | 0.659 | 1.017 | 0.974 | 0.902 |
| MLP8 | 0.234 | 0.34 | 0.993 | 0.993 | 0.348 | 0.523 | 0.989 | 0.974 |
| **MLP9** | **0.161** | **0.211** | **0.989** | **0.989** | **0.190** | **0.265** | **0.989** | **0.978** |
| SS | 0.539 | 0.761 | 0.964 | 0.964 | 0.681 | 1.053 | 0.963 | 0.963 |
| MLR | 0.712 | 0.89 | 0.951 | 0.951 | 0.740 | 0.861 | 0.969 | 0.968 |

Table 6. Comparisons of different models for predicting *Ep* at MQ station.

| MQ | Training | | | | Testing | | | |
|---|---|---|---|---|---|---|---|---|
| | MAE | RMSE | $R^2$ | E | MAE | RMSE | $R^2$ | E |
| ANFIS-GP1 | 1.337 | 1.76 | 0.85 | 0.85 | 1.133 | 1.396 | 0.941 | 0.889 |
| ANFIS-GP2 | 1.33 | 1.698 | 0.86 | 0.86 | 1.203 | 1.587 | 0.863 | 0.856 |
| ANFIS-GP3 | 2.467 | 3.11 | 0.53 | 0.53 | 2.453 | 3.045 | 0.55 | 0.47 |
| ANFIS-GP4 | 3.895 | 4.324 | 0.092 | 0.092 | 3.758 | 4.146 | 0.035 | 0.018 |
| ANFIS-GP5 | 3.256 | 3.807 | 0.296 | 0.296 | 2.879 | 3.385 | 0.353 | 0.345 |
| ANFIS-GP6 | 0.83 | 1.178 | 0.933 | 0.933 | 0.868 | 1.220 | 0.952 | 0.915 |
| ANFIS-GP7 | 0.828 | 1.165 | 0.934 | 0.934 | 0.882 | 1.229 | 0.951 | 0.914 |
| ANFIS-GP8 | 0.648 | 0.886 | 0.962 | 0.962 | 0.608 | 0.810 | 0.981 | 0.963 |
| **ANFIS-GP9** | **0.474** | **0.66** | **0.979** | **0.979** | **0.512** | **0.646** | **0.987** | **0.976** |
| FG1 | 1.297 | 1.735 | 0.854 | 0.854 | 1.112 | 1.412 | 0.926 | 0.886 |
| FG2 | 1.263 | 1.638 | 0.87 | 0.87 | 1.198 | 1.555 | 0.87 | 0.862 |
| FG3 | 2.447 | 3.057 | 0.546 | 0.546 | 2.373 | 2.953 | 0.58 | 0.504 |
| FG4 | 3.871 | 4.307 | 0.10 | 0.1 | 3.746 | 4.130 | 0.04 | 0.026 |
| FG5 | 3.215 | 3.782 | 0.306 | 0.306 | 2.83 | 3.344 | 0.366 | 0.361 |

| | | | | | | | |
|---|---|---|---|---|---|---|---|
| FG6 | 0.828 | 1.178 | 0.933 | 0.933 | 0.854 | 1.196 | 0.952 | 0.917 |
| FG7 | 0.795 | 1.13 | 0.938 | 0.938 | 0.923 | 1.335 | 0.942 | 0.914 |
| FG8 | 0.608 | 0.81 | 0.968 | 0.968 | 0.636 | 0.805 | 0.978 | 0.968 |
| **FG9** | **0.456** | **0.614** | **0.983** | **0.973** | **0.435** | **0.574** | **0.99** | **0.974** |
| GRNN1 | 1.289 | 1.725 | 0.856 | 0.856 | 1.076 | 1.386 | 0.927 | 0.890 |
| GRNN2 | 1.225 | 1.593 | 0.877 | 0.877 | 1.148 | 1.504 | 0.876 | 0.871 |
| GRNN3 | 2.441 | 3.04 | 0.552 | 0.551 | 2.35 | 2.909 | 0.585 | 0.516 |
| GRNN4 | 3.845 | 4.281 | 0.112 | 0.111 | 3.701 | 4.091 | 0.051 | 0.044 |
| GRNN5 | 3.379 | 3.85 | 0.304 | 0.281 | 3.026 | 3.429 | 0.363 | 0.328 |
| GRNN6 | 0.688 | 1.002 | 0.951 | 0.951 | 0.777 | 1.099 | 0.954 | 0.931 |
| GRNN7 | 0.508 | 0.761 | 0.972 | 0.972 | 0.849 | 1.210 | 0.948 | 0.916 |
| GRNN8 | 0.178 | 0.291 | 0.996 | 0.996 | 0.66 | 0.947 | 0.966 | 0.949 |
| **GRNN9** | **0.055** | **0.126** | **0.999** | **0.999** | **0.599** | **0.832** | **0.973** | **0.960** |
| LSSVM1 | 1.295 | 1.732 | 0.854 | 0.854 | 1.107 | 1.411 | 0.927 | 0.886 |
| LSSVM2 | 1.259 | 1.634 | 0.87 | 0.87 | 1.202 | 1.561 | 0.869 | 0.861 |
| LSSVM3 | 2.713 | 3.216 | 0.522 | 0.498 | 2.523 | 2.983 | 0.558 | 0.492 |
| LSSVM4 | 3.861 | 4.296 | 0.105 | 0.104 | 3.711 | 4.094 | 0.052 | 0.043 |
| LSSVM5 | 3.242 | 3.789 | 0.304 | 0.303 | 2.872 | 3.369 | 0.358 | 0.352 |
| LSSVM6 | 0.841 | 1.182 | 0.933 | 0.932 | 0.858 | 1.166 | 0.951 | 0.922 |
| LSSVM7 | 0.911 | 1.225 | 0.929 | 0.927 | 0.933 | 1.200 | 0.95 | 0.918 |
| LSSVM8 | 0.982 | 1.243 | 0.937 | 0.925 | 0.919 | 1.118 | 0.960 | 0.929 |
| **LSSVM9** | **0.549** | **0.747** | **0.974** | **0.973** | **0.544** | **0.711** | **0.982** | **0.971** |
| MARS1 | 1.352 | 1.76 | 0.85 | 0.85 | 1.133 | 1.403 | 0.936 | 0.888 |
| MARS2 | 1.076 | 1.46 | 0.897 | 0.897 | 1.08 | 1.472 | 0.888 | 0.876 |
| MARS3 | 2.419 | 3.039 | 0.552 | 0.552 | 2.436 | 2.996 | 0.564 | 0.487 |
| MARS4 | 3.829 | 4.282 | 0.11 | 0.11 | 3.751 | 4.125 | 0.046 | 0.028 |
| MARS5 | 3.225 | 3.813 | 0.294 | 0.294 | 2.878 | 3.391 | 0.35 | 0.343 |
| MARS6 | 0.804 | 1.127 | 0.938 | 0.938 | 0.921 | 1.247 | 0.948 | 0.911 |
| MARS7 | 0.807 | 1.126 | 0.938 | 0.938 | 0.97 | 1.290 | 0.95 | 0.905 |
| MARS8 | 0.668 | 0.87 | 0.963 | 0.963 | 0.735 | 0.929 | 0.973 | 0.951 |
| **MARS9** | **0.546** | **0.72** | **0.975** | **0.975** | **0.627** | **0.826** | **0.977** | **0.961** |
| MLP1 | 1.297 | 1.735 | 0.854 | 0.854 | 1.107 | 1.408 | 0.928 | 0.887 |
| MLP2 | 1.057 | 1.458 | 0.897 | 0.897 | 1.113 | 1.492 | 0.888 | 0.873 |
| MLP3 | 1.139 | 1.524 | 0.887 | 0.887 | 1.108 | 1.488 | 0.884 | 0.872 |
| MLP4 | 3.833 | 4.289 | 0.107 | 0.107 | 3.676 | 4.069 | 0.063 | 0.054 |
| MLP5 | 3.179 | 3.761 | 0.313 | 0.313 | 2.81 | 3.329 | 0.371 | 0.367 |
| MLP6 | 0.724 | 1.026 | 0.949 | 0.949 | 0.797 | 1.074 | 0.96 | 0.935 |
| MLP7 | 0.742 | 1.064 | 0.945 | 0.945 | 0.821 | 1.113 | 0.959 | 0.929 |
| MLP8 | 0.538 | 0.738 | 0.974 | 0.974 | 0.538 | 0.716 | 0.981 | 0.97 |
| **MLP9** | **0.384** | **0.532** | **0.986** | **0.986** | **0.358** | **0.489** | **0.99** | **0.986** |
| SS | 0.922 | 1.281 | 0.92 | 0.92 | 1.039 | 1.389 | 0.944 | 0.942 |
| MLR | 0.77 | 0.967 | 0.955 | 0.955 | 0.784 | 0.921 | 0.972 | 0.971 |

Table 7. Comparisons of different models for predicting *Ep* at BJ station.

| BJ | Training | | | | Testing | | | |
|---|---|---|---|---|---|---|---|---|
| | MAE | RMSE | $R^2$ | E | MAE | RMSE | $R^2$ | E |
| ANFIS-GP1 | 0.872 | 1.205 | 0.826 | 0.826 | 0.749 | 0.956 | 0.922 | 0.868 |
| ANFIS-GP2 | 1.439 | 1.907 | 0.564 | 0.564 | 1.294 | 1.554 | 0.662 | 0.650 |
| ANFIS-GP3 | 1.431 | 1.818 | 0.603 | 0.603 | 1.482 | 1.880 | 0.561 | 0.488 |
| ANFIS-GP4 | 2.306 | 2.881 | 0.005 | 0.005 | 2.223 | 2.608 | 0.019 | 0.014 |
| ANFIS-GP5 | 2.345 | 2.865 | 0.015 | 0.015 | 2.22 | 2.577 | 0.07 | 0.038 |
| ANFIS-GP6 | 0.846 | 1.189 | 0.831 | 0.831 | 0.717 | 0.923 | 0.921 | 0.877 |
| ANFIS-GP7 | 0.742 | 1.071 | 0.862 | 0.862 | 0.688 | 0.972 | 0.909 | 0.863 |
| ANFIS-GP8 | 0.464 | 0.735 | 0.935 | 0.935 | 0.384 | 0.510 | 0.965 | 0.962 |
| **ANFIS-GP9** | **0.424** | **0.657** | **0.948** | **0.948** | **0.361** | **0.48** | **0.971** | **0.967** |
| FG1 | 0.835 | 1.127 | 0.848 | 0.848 | 0.823 | 1.075 | 0.914 | 0.828 |
| FG2 | 1.416 | 1.891 | 0.571 | 0.571 | 1.256 | 1.544 | 0.665 | 0.653 |
| FG3 | 1.387 | 1.733 | 0.64 | 0.64 | 1.483 | 1.846 | 0.561 | 0.504 |
| FG4 | 2.244 | 2.839 | 0.033 | 0.033 | 2.23 | 2.653 | 0.001 | -0.051 |
| FG5 | 2.288 | 2.822 | 0.045 | 0.045 | 2.234 | 2.608 | 0.02 | 0.015 |
| FG6 | 0.742 | 1.063 | 0.864 | 0.864 | 0.688 | 0.997 | 0.922 | 0.855 |
| FG7 | 0.721 | 1.052 | 0.867 | 0.867 | 0.679 | 0.959 | 0.926 | 0.869 |
| FG8 | 0.451 | 0.721 | 0.938 | 0.938 | 0.394 | 0.484 | 0.971 | 0.965 |
| **FG9** | **0.438** | **0.662** | **0.947** | **0.947** | **0.355** | **0.443** | **0.977** | **0.972** |
| GRNN1 | 0.819 | 1.114 | 0.852 | 0.851 | 0.811 | 1.062 | 0.916 | 0.837 |
| GRNN2 | 1.379 | 1.852 | 0.589 | 0.588 | 1.23 | 1.520 | 0.678 | 0.665 |
| GRNN3 | 1.347 | 1.702 | 0.654 | 0.653 | 1.487 | 1.850 | 0.559 | 0.504 |
| GRNN4 | 2.262 | 2.827 | 0.044 | 0.041 | 2.253 | 2.667 | 0.001 | -0.03 |
| GRNN5 | 2.32 | 2.856 | 0.023 | 0.021 | 2.224 | 2.591 | 0.101 | 0.027 |
| GRNN6 | 0.626 | 0.924 | 0.898 | 0.898 | 0.657 | 0.939 | 0.904 | 0.872 |
| GRNN7 | 0.473 | 0.754 | 0.932 | 0.932 | 0.68 | 0.967 | 0.885 | 0.865 |
| GRNN8 | 0.185 | 0.348 | 0.986 | 0.985 | 0.403 | 0.541 | 0.958 | 0.958 |
| **GRNN9** | **0.166** | **0.301** | **0.989** | **0.989** | **0.356** | **0.473** | **0.968** | **0.967** |
| LSSVM1 | 0.831 | 1.121 | 0.849 | 0.849 | 0.823 | 1.068 | 0.916 | 0.835 |
| LSSVM2 | 1.409 | 1.883 | 0.575 | 0.575 | 1.238 | 1.525 | 0.677 | 0.663 |
| LSSVM3 | 1.502 | 1.877 | 0.606 | 0.577 | 1.522 | 1.865 | 0.547 | 0.496 |
| LSSVM4 | 2.281 | 2.851 | 0.025 | 0.025 | 2.236 | 2.630 | 0.008 | -0.18 |
| LSSVM5 | 2.332 | 2.858 | 0.02 | 0.02 | 2.216 | 2.579 | 0.093 | 0.036 |
| LSSVM6 | 0.775 | 1.117 | 0.852 | 0.85 | 0.647 | 0.927 | 0.926 | 0.875 |
| LSSVM7 | 0.785 | 1.129 | 0.849 | 0.847 | 0.66 | 0.932 | 0.92 | 0.874 |
| LSSVM8 | 0.733 | 1.074 | 0.876 | 0.862 | 0.619 | 0.823 | 0.926 | 0.902 |
| **LSSVM9** | **0.481** | **0.753** | **0.934** | **0.932** | **0.335** | **0.437** | **0.974** | **0.972** |
| MARS1 | 0.835 | 1.129 | 0.847 | 0.847 | 0.857 | 1.080 | 0.914 | 0.831 |
| MARS2 | 1.364 | 1.832 | 0.597 | 0.597 | 1.282 | 1.607 | 0.659 | 0.626 |
| MARS3 | 1.359 | 1.717 | 0.646 | 0.646 | 1.48 | 1.844 | 0.558 | 0.507 |
| MARS4 | 2.183 | 2.728 | 0.107 | 0.107 | 2.392 | 2.962 | 0.001 | -0.271 |
| MARS5 | 2.323 | 2.86 | 0.019 | 0.019 | 2.245 | 2.616 | 0.01 | 0.008 |
| MARS6 | 0.694 | 0.974 | 0.886 | 0.886 | 0.685 | 0.986 | 0.904 | 0.859 |
| MARS7 | 0.691 | 0.977 | 0.886 | 0.886 | 0.671 | 0.967 | 0.911 | 0.865 |
| MARS8 | 0.52 | 0.767 | 0.929 | 0.929 | 0.5 | 0.603 | 0.963 | 0.947 |

| | MAE | RMSE | R² | E | MAE | RMSE | R² | E |
|---|---|---|---|---|---|---|---|---|
| **MARS9** | **0.478** | **0.717** | **0.938** | **0.938** | **0.427** | **0.527** | **0.971** | **0.960** |
| MLP1 | 0.784 | 1.075 | 0.861 | 0.861 | 0.813 | 1.045 | 0.914 | 0.842 |
| MLP2 | 1.325 | 1.803 | 0.61 | 0.61 | 1.249 | 1.595 | 0.677 | 0.631 |
| MLP3 | 1.401 | 1.875 | 0.578 | 0.578 | 1.236 | 1.523 | 0.678 | 0.664 |
| MLP4 | 2.268 | 2.857 | 0.021 | 0.021 | 2.214 | 2.620 | 0.013 | 0.005 |
| MLP5 | 2.298 | 2.823 | 0.045 | 0.044 | 2.204 | 2.581 | 0.042 | 0.035 |
| MLP6 | 0.675 | 0.968 | 0.888 | 0.888 | 0.66 | 0.974 | 0.911 | 0.867 |
| MLP7 | 0.653 | 0.962 | 0.889 | 0.889 | 0.62 | 0.907 | 0.904 | 0.88 |
| **MLP8** | **0.417** | **0.692** | **0.943** | **0.943** | **0.312** | **0.394** | **0.982** | **0.977** |
| MLP9 | 0.337 | 0.506 | 0.969 | 0.969 | 0.314 | 0.428 | 0.979 | 0.972 |
| SS | 0.89 | 1.263 | 0.816 | 0.816 | 0.647 | 0.921 | 0.897 | 0.895 |
| MLR | 0.614 | 0.879 | 0.907 | 0.907 | 0.514 | 0.648 | 0.946 | 0.945 |

Table 8. Comparisons of different models for predicting $Ep$ at LSA station.

| LSA | Training | | | | Testing | | | |
|---|---|---|---|---|---|---|---|---|
| | MAE | RMSE | R² | E | MAE | RMSE | R² | E |
| ANFIS-GP1 | 1.327 | 1.718 | 0.411 | 0.411 | 1.072 | 1.424 | 0.594 | 0.581 |
| ANFIS-GP2 | 1.245 | 1.523 | 0.536 | 0.536 | 1.192 | 1.417 | 0.601 | 0.585 |
| ANFIS-GP3 | 1.821 | 2.218 | 0.017 | 0.017 | 1.796 | 2.148 | 0.055 | 0.046 |
| ANFIS-GP4 | 1.772 | 2.196 | 0.037 | 0.037 | 1.794 | 2.127 | 0.072 | 0.065 |
| ANFIS-GP5 | 1.721 | 2.11 | 0.111 | 0.111 | 1.695 | 2.033 | 0.178 | 0.146 |
| ANFIS-GP6 | 1.149 | 1.471 | 0.568 | 0.568 | 1.046 | 1.304 | 0.651 | 0.648 |
| ANFIS-GP7 | 0.966 | 1.223 | 0.701 | 0.701 | 0.875 | 1.082 | 0.761 | 0.758 |
| **ANFIS-GP8** | **0.529** | **0.675** | **0.909** | **0.909** | **0.73** | **0.907** | **0.896** | **0.830** |
| ANFIS-GP9 | 0.478 | 0.61 | 0.926 | 0.926 | 0.816 | 1.038 | 0.875 | 0.777 |
| FG1 | 1.324 | 1.715 | 0.413 | 0.413 | 1.073 | 1.415 | 0.600 | 0.584 |
| FG2 | 1.151 | 1.465 | 0.571 | 0.571 | 1.159 | 1.392 | 0.621 | 0.600 |
| FG3 | 1.803 | 2.169 | 0.06 | 0.06 | 1.771 | 2.093 | 0.118 | 0.115 |
| FG4 | 1.627 | 2.072 | 0.143 | 0.143 | 1.64 | 2.036 | 0.145 | 0.141 |
| FG5 | 1.683 | 2.092 | 0.125 | 0.125 | 1.704 | 2.022 | 0.163 | 0.152 |
| FG6 | 1.044 | 1.381 | 0.619 | 0.619 | 0.987 | 1.201 | 0.725 | 0.705 |
| FG7 | 0.968 | 1.215 | 0.705 | 0.705 | 0.896 | 1.099 | 0.757 | 0.737 |
| FG8 | 0.499 | 0.631 | 0.921 | 0.921 | 0.767 | 0.925 | 0.903 | 0.823 |
| **FG9** | **0.491** | **0.61** | **0.926** | **0.926** | **0.729** | **0.886** | **0.914** | **0.86** |
| GRNN1 | 1.296 | 1.692 | 0.429 | 0.428 | 1.094 | 1.436 | 0.587 | 0.574 |
| GRNN2 | 1.025 | 1.336 | 0.647 | 0.643 | 1.072 | 1.288 | 0.679 | 0.657 |
| GRNN3 | 1.783 | 2.152 | 0.077 | 0.075 | 1.762 | 2.080 | 0.134 | 0.105 |
| GRNN4 | 1.618 | 2.079 | 0.138 | 0.136 | 1.586 | 1.971 | 0.209 | 0.196 |
| GRNN5 | 1.712 | 2.108 | 0.119 | 0.112 | 1.705 | 2.052 | 0.176 | 0.13 |
| GRNN6 | 0.791 | 1.067 | 0.778 | 0.773 | 0.83 | 1.044 | 0.786 | 0.775 |
| GRNN7 | 0.425 | 0.598 | 0.93 | 0.928 | 0.789 | 1.025 | 0.789 | 0.783 |
| GRNN8 | 0.137 | 0.202 | 0.992 | 0.992 | 0.566 | 0.711 | 0.914 | 0.895 |
| **GRNN9** | **0.056** | **0.103** | **0.998** | **0.998** | **0.459** | **0.592** | **0.933** | **0.932** |
| LSSVM1 | 1.307 | 1.706 | 0.419 | 0.419 | 1.083 | 1.420 | 0.594 | 0.583 |
| LSSVM2 | 1.008 | 1.32 | 0.652 | 0.652 | 1.109 | 1.317 | 0.67 | 0.641 |

| | | | | | | | | |
|---|---|---|---|---|---|---|---|---|
| LSSVM3 | 1.811 | 2.209 | 0.027 | 0.025 | 1.791 | 2.144 | 0.07 | 0.05 |
| LSSVM4 | 1.663 | 2.113 | 0.109 | 0.107 | 1.638 | 2.010 | 0.187 | 0.165 |
| LSSVM5 | 1.721 | 2.109 | 0.112 | 0.111 | 1.692 | 2.030 | 0.182 | 0.148 |
| LSSVM6 | 1.107 | 1.431 | 0.592 | 0.591 | 1.033 | 1.268 | 0.681 | 0.668 |
| LSSVM7 | 1.073 | 1.336 | 0.65 | 0.644 | 0.915 | 1.119 | 0.759 | 0.741 |
| LSSVM8 | 0.957 | 1.199 | 0.766 | 0.713 | 0.869 | 1.043 | 0.833 | 0.775 |
| LSSVM9 | 0.57 | 0.753 | 0.894 | 0.887 | 0.582 | 0.702 | 0.926 | 0.898 |
| MARS1 | 1.311 | 1.72 | 0.409 | 0.409 | 1.091 | 1.440 | 0.585 | 0.571 |
| MARS2 | 1.012 | 1.318 | 0.653 | 0.653 | 1.098 | 1.299 | 0.683 | 0.651 |
| MARS3 | 1.82 | 2.182 | 0.049 | 0.049 | 1.766 | 2.089 | 0.12 | 0.097 |
| MARS4 | 1.663 | 2.098 | 0.121 | 0.121 | 1.639 | 2.020 | 0.159 | 0.157 |
| MARS5 | 1.694 | 2.096 | 0.122 | 0.122 | 1.705 | 2.037 | 0.161 | 0.143 |
| MARS6 | 0.917 | 1.23 | 0.698 | 0.698 | 0.947 | 1.176 | 0.735 | 0.714 |
| MARS7 | 0.94 | 1.227 | 0.699 | 0.699 | 0.913 | 1.135 | 0.746 | 0.734 |
| MARS8 | 0.501 | 0.641 | 0.918 | 0.918 | 0.762 | 0.929 | 0.91 | 0.822 |
| MARS9 | 0.516 | 0.662 | 0.912 | 0.912 | 0.668 | 0.818 | 0.924 | 0.862 |
| MLP1 | 1.308 | 1.707 | 0.418 | 0.418 | 1.073 | 1.413 | 0.596 | 0.586 |
| MLP2 | 0.992 | 1.307 | 0.659 | 0.658 | 1.111 | 1.313 | 0.675 | 0.647 |
| MLP3 | 0.994 | 1.316 | 0.654 | 0.654 | 1.108 | 1.312 | 0.675 | 0.647 |
| MLP4 | 1.601 | 2.075 | 0.14 | 0.14 | 1.561 | 1.956 | 0.216 | 0.207 |
| MLP5 | 1.686 | 2.095 | 0.124 | 0.124 | 1.703 | 2.022 | 0.162 | 0.155 |
| MLP6 | 0.883 | 1.187 | 0.719 | 0.719 | 0.918 | 1.123 | 0.754 | 0.746 |
| MLP7 | 0.686 | 0.91 | 0.835 | 0.835 | 0.728 | 0.958 | 0.825 | 0.803 |
| MLP8 | 0.397 | 0.503 | 0.949 | 0.949 | 0.629 | 0.771 | 0.928 | 0.877 |
| MLP9 | 0.522 | 0.681 | 0.907 | 0.907 | 0.53 | 0.638 | 0.936 | 0.916 |
| SS | 1.198 | 1.577 | 0.515 | 0.515 | 0.969 | 1.307 | 0.652 | 0.652 |
| MLR | 0.628 | 0.795 | 0.874 | 0.874 | 0.656 | 0.789 | 0.906 | 0.905 |

Table 9. Comparisons of different models for predicting $Ep$ at CQ station.

| | Training | | | | Testing | | | |
|---|---|---|---|---|---|---|---|---|
| CQ | MAE | RMSE | $R^2$ | E | MAE | RMSE | $R^2$ | E |
| ANFIS-GP1 | 0.466 | 0.859 | 0.815 | 0.815 | 0.280 | 0.397 | 0.958 | 0.953 |
| ANFIS-GP2 | 0.82 | 1.189 | 0.645 | 0.645 | 0.693 | 0.959 | 0.748 | 0.726 |
| ANFIS-GP3 | 0.539 | 0.722 | 0.869 | 0.869 | 0.537 | 0.679 | 0.876 | 0.863 |
| ANFIS-GP4 | 1.631 | 1.958 | 0.036 | 0.036 | 1.486 | 1.743 | 0.553 | 0.096 |
| ANFIS-GP5 | 1.298 | 1.619 | 0.341 | 0.341 | 1.234 | 1.557 | 0.406 | 0.279 |
| ANFIS-GP6 | 0.416 | 0.786 | 0.845 | 0.845 | 0.316 | 0.398 | 0.959 | 0.953 |
| ANFIS-GP7 | 0.369 | 0.492 | 0.939 | 0.939 | 0.242 | 0.329 | 0.968 | 0.968 |
| ANFIS-GP8 | 0.225 | 0.29 | 0.979 | 0.979 | 0.224 | 0.312 | 0.976 | 0.971 |
| ANFIS-GP9 | 0.187 | 0.244 | 0.985 | 0.985 | 0.203 | 0.300 | 0.978 | 0.973 |
| FG1 | 0.467 | 0.805 | 0.837 | 0.837 | 0.294 | 0.375 | 0.963 | 0.959 |
| FG2 | 0.616 | 0.877 | 0.807 | 0.807 | 0.568 | 0.685 | 0.876 | 0.860 |
| FG3 | 0.474 | 0.672 | 0.887 | 0.887 | 0.479 | 0.607 | 0.905 | 0.890 |
| FG4 | 1.066 | 1.343 | 0.547 | 0.547 | 1.015 | 1.241 | 0.542 | 0.539 |
| FG5 | 1.291 | 1.601 | 0.356 | 0.356 | 1.308 | 1.659 | 0.324 | 0.183 |

| | | | | | | | | |
|---|---|---|---|---|---|---|---|---|
| FG6 | 0.385 | 0.704 | 0.876 | 0.876 | 0.303 | 0.384 | 0.96 | 0.957 |
| FG7 | 0.396 | 0.572 | 0.918 | 0.918 | 0.294 | 0.385 | 0.958 | 0.956 |
| FG8 | 0.287 | 0.38 | 0.964 | 0.964 | 0.229 | 0.299 | 0.974 | 0.973 |
| **FG9** | **0.195** | **0.25** | **0.984** | **0.984** | **0.182** | **0.280** | **0.981** | **0.979** |
| GRNN1 | 0.437 | 0.746 | 0.861 | 0.86 | 0.284 | 0.374 | 0.963 | 0.958 |
| GRNN2 | 0.574 | 0.845 | 0.823 | 0.821 | 0.507 | 0.651 | 0.883 | 0.874 |
| GRNN3 | 0.453 | 0.652 | 0.893 | 0.893 | 0.473 | 0.610 | 0.902 | 0.889 |
| GRNN4 | 1.212 | 1.475 | 0.562 | 0.453 | 1.158 | 1.330 | 0.557 | 0.474 |
| GRNN5 | 1.318 | 1.617 | 0.354 | 0.342 | 1.253 | 1.548 | 0.395 | 0.287 |
| GRNN6 | 0.306 | 0.598 | 0.911 | 0.91 | 0.279 | 0.384 | 0.959 | 0.956 |
| GRNN7 | 0.197 | 0.278 | 0.981 | 0.981 | 0.243 | 0.328 | 0.968 | 0.968 |
| **GRNN8** | **0.145** | **0.203** | **0.99** | **0.99** | **0.177** | **0.240** | **0.983** | **0.983** |
| GRNN9 | 0.227 | 0.308 | 0.977 | 0.977 | 0.234 | 0.297 | 0.975 | 0.974 |
| LSSVM1 | 0.449 | 0.758 | 0.856 | 0.856 | 0.282 | 0.377 | 0.961 | 0.958 |
| LSSVM2 | 0.552 | 0.825 | 0.829 | 0.829 | 0.503 | 0.650 | 0.888 | 0.874 |
| LSSVM3 | 0.687 | 0.862 | 0.887 | 0.813 | 0.625 | 0.765 | 0.906 | 0.826 |
| LSSVM4 | 1.07 | 1.345 | 0.548 | 0.545 | 1.003 | 1.222 | 0.556 | 0.555 |
| LSSVM5 | 1.305 | 1.626 | 0.336 | 0.335 | 1.233 | 1.543 | 0.406 | 0.292 |
| LSSVM6 | 0.399 | 0.741 | 0.864 | 0.862 | 0.322 | 0.399 | 0.960 | 0.953 |
| LSSVM7 | 0.391 | 0.586 | 0.918 | 0.914 | 0.266 | 0.355 | 0.966 | 0.962 |
| LSSVM8 | 0.407 | 0.634 | 0.916 | 0.899 | 0.284 | 0.392 | 0.968 | 0.954 |
| **LSSVM9** | **0.313** | **0.482** | **0.944** | **0.941** | **0.219** | **0.290** | **0.976** | **0.975** |
| MARS1 | 0.5 | 0.753 | 0.858 | 0.858 | 0.274 | 0.357 | 0.964 | 0.962 |
| MARS2 | 0.559 | 0.806 | 0.837 | 0.837 | 0.509 | 0.653 | 0.888 | 0.873 |
| MARS3 | 0.453 | 0.664 | 0.889 | 0.889 | 0.466 | 0.599 | 0.904 | 0.904 |
| MARS4 | 0.974 | 1.234 | 0.617 | 0.617 | 1.09 | 1.38 | 0.466 | 0.433 |
| MARS5 | 1.236 | 1.51 | 0.427 | 0.427 | 1.321 | 1.741 | 0.277 | 0.10 |
| MARS6 | 0.354 | 0.617 | 0.904 | 0.904 | 0.333 | 0.444 | 0.949 | 0.941 |
| MARS7 | 0.336 | 0.48 | 0.942 | 0.942 | 0.292 | 0.372 | 0.959 | 0.959 |
| **MARS8** | **0.273** | **0.426** | **0.954** | **0.954** | **0.221** | **0.300** | **0.973** | **0.973** |
| MARS9 | 0.267 | 0.417 | 0.956 | 0.956 | 0.25 | 0.323 | 0.970 | 0.969 |
| MLP1 | 0.419 | 0.733 | 0.865 | 0.865 | 0.27 | 0.371 | 0.96 | 0.959 |
| MLP2 | 0.55 | 0.81 | 0.835 | 0.835 | 0.509 | 0.658 | 0.887 | 0.872 |
| MLP3 | 0.568 | 0.845 | 0.82 | 0.82 | 0.502 | 0.637 | 0.893 | 0.877 |
| MLP4 | 1.052 | 1.338 | 0.55 | 0.55 | 0.999 | 1.231 | 0.55 | 0.549 |
| MLP5 | 1.299 | 1.61 | 0.348 | 0.348 | 1.263 | 1.591 | 0.381 | 0.247 |
| MLP6 | 0.334 | 0.65 | 0.894 | 0.894 | 0.266 | 0.355 | 0.966 | 0.947 |
| MLP7 | 0.252 | 0.348 | 0.97 | 0.969 | 0.218 | 0.296 | 0.975 | 0.971 |
| **MLP8** | **0.185** | **0.239** | **0.986** | **0.986** | **0.167** | **0.230** | **0.985** | **0.984** |
| MLP9 | 0.161 | 0.211 | 0.989 | 0.989 | 0.189 | 0.265 | 0.985 | 0.979 |
| SS | 0.379 | 0.786 | 0.847 | 0.847 | 0.226 | 0.307 | 0.973 | 0.971 |
| MLR | 0.389 | 0.534 | 0.928 | 0.928 | 0.317 | 0.398 | 0.955 | 0.955 |

Table 10. Comparisons of different models for predicting *Ep* at HZ station.

| HZ | Training | | | | Testing | | | |
|---|---|---|---|---|---|---|---|---|
| | MAE | RMSE | $R^2$ | E | MAE | RMSE | $R^2$ | E |
| ANFIS-GP1 | 0.532 | 0.698 | 0.87 | 0.87 | 0.451 | 0.605 | 0.903 | 0.902 |
| ANFIS-GP2 | 0.72 | 1.001 | 0.734 | 0.734 | 0.728 | 0.965 | 0.754 | 0.728 |
| ANFIS-GP3 | 0.937 | 1.164 | 0.64 | 0.64 | 0.991 | 1.178 | 0.694 | 0.631 |
| ANFIS-GP4 | 1.567 | 1.94 | 0.001 | 0.001 | 1.59 | 1.943 | 0.017 | -0.004 |
| ANFIS-GP5 | 1.569 | 1.931 | 0.009 | 0.009 | 1.557 | 1.910 | 0.084 | 0.029 |
| ANFIS-GP6 | 0.377 | 0.521 | 0.928 | 0.928 | 0.333 | 0.448 | 0.948 | 0.947 |
| ANFIS-GP7 | 0.357 | 0.482 | 0.938 | 0.938 | 0.311 | 0.397 | 0.961 | 0.958 |
| **ANFIS-GP8** | **0.272** | **0.356** | **0.966** | **0.966** | **0.329** | **0.427** | **0.965** | **0.951** |
| ANFIS-GP9 | 0.242 | 0.312 | 0.974 | 0.974 | 0.347 | 0.453 | 0.949 | 0.945 |
| FG1 | 0.519 | 0.686 | 0.875 | 0.875 | 0.438 | 0.590 | 0.908 | 0.907 |
| FG2 | 0.612 | 0.79 | 0.834 | 0.834 | 0.613 | 0.764 | 0.846 | 0.845 |
| FG3 | 0.943 | 1.151 | 0.648 | 0.648 | 1.012 | 1.188 | 0.701 | 0.624 |
| FG4 | 1.572 | 1.925 | 0.015 | 0.015 | 1.576 | 1.922 | 0.024 | 0.018 |
| FG5 | 1.519 | 1.862 | 0.08 | 0.08 | 1.67 | 2.014 | 0.02 | -0.081 |
| FG6 | 0.358 | 0.485 | 0.938 | 0.938 | 0.299 | 0.397 | 0.959 | 0.958 |
| FG7 | 0.344 | 0.462 | 0.943 | 0.943 | 0.29 | 0.373 | 0.965 | 0.963 |
| FG8 | 0.269 | 0.347 | 0.968 | 0.968 | 0.295 | 0.375 | 0.974 | 0.952 |
| **FG9** | **0.26** | **0.36** | **0.966** | **0.966** | **0.278** | **0.369** | **0.964** | **0.963** |
| GRNN1 | 0.519 | 0.68 | 0.878 | 0.877 | 0.457 | 0.607 | 0.904 | 0.902 |
| GRNN2 | 0.556 | 0.733 | 0.859 | 0.857 | 0.581 | 0.736 | 0.86 | 0.856 |
| GRNN3 | 0.926 | 1.127 | 0.664 | 0.662 | 1.02 | 1.197 | 0.705 | 0.619 |
| GRNN4 | 1.564 | 1.926 | 0.023 | 0.022 | 1.578 | 1.930 | 0.006 | 0.005 |
| GRNN5 | 1.526 | 1.882 | 0.09 | 0.059 | 1.621 | 1.971 | 0.011 | -0.033 |
| GRNN6 | 0.322 | 0.438 | 0.949 | 0.949 | 0.314 | 0.409 | 0.957 | 0.955 |
| GRNN7 | 0.238 | 0.327 | 0.972 | 0.972 | 0.295 | 0.404 | 0.961 | 0.957 |
| **GRNN8** | **0.119** | **0.17** | **0.992** | **0.992** | **0.308** | **0.400** | **0.960** | **0.956** |
| GRNN9 | 0.047 | 0.084 | 0.998 | 0.998 | 0.367 | 0.524 | 0.929 | 0.927 |
| LSSVM1 | 0.517 | 0.679 | 0.878 | 0.878 | 0.442 | 0.596 | 0.906 | 0.905 |
| LSSVM2 | 0.55 | 0.713 | 0.865 | 0.865 | 0.546 | 0.695 | 0.873 | 0.872 |
| LSSVM3 | 0.996 | 1.214 | 0.638 | 0.608 | 1.074 | 1.267 | 0.678 | 0.573 |
| LSSVM4 | 1.554 | 1.918 | 0.022 | 0.022 | 1.568 | 1.937 | 0.005 | 0.002 |
| LSSVM5 | 1.527 | 1.865 | 0.078 | 0.075 | 1.654 | 1.996 | 0.016 | -0.06 |
| LSSVM6 | 0.367 | 0.504 | 0.933 | 0.932 | 0.325 | 0.437 | 0.951 | 0.949 |
| LSSVM7 | 0.364 | 0.496 | 0.937 | 0.935 | 0.328 | 0.427 | 0.961 | 0.951 |
| **LSSVM8** | **0.385** | **0.538** | **0.935** | **0.923** | **0.379** | **0.493** | **0.953** | **0.935** |
| LSSVM9 | 0.278 | 0.382 | 0.963 | 0.961 | 0.296 | 0.397 | 0.961 | 0.958 |
| MARS1 | 0.52 | 0.69 | 0.874 | 0.874 | 0.443 | 0.601 | 0.904 | 0.904 |
| MARS2 | 0.534 | 0.686 | 0.875 | 0.875 | 0.524 | 0.673 | 0.881 | 0.879 |
| MARS3 | 0.915 | 1.132 | 0.660 | 0.66 | 1.032 | 1.226 | 0.675 | 0.60 |
| MARS4 | 1.571 | 1.94 | 0.0 | 0 | 1.591 | 1.939 | 0.00 | 0 |
| MARS5 | 1.486 | 1.833 | 0.107 | 0.107 | 1.712 | 2.069 | 0.024 | -0.139 |
| MARS6 | 0.339 | 0.449 | 0.946 | 0.946 | 0.273 | 0.362 | 0.966 | 0.965 |
| MARS7 | 0.335 | 0.437 | 0.949 | 0.949 | 0.282 | 0.358 | 0.966 | 0.966 |
| MARS8 | 0.287 | 0.374 | 0.963 | 0.963 | 0.314 | 0.386 | 0.976 | 0.96 |

| | MAE | RMSE | R² | E | MAE | RMSE | R² | E |
|---|---|---|---|---|---|---|---|---|
| **MARS9** | **0.27** | **0.358** | **0.966** | **0.966** | **0.276** | **0.361** | **0.967** | **0.965** |
| MLP1 | 0.529 | 0.691 | 0.873 | 0.873 | 0.449 | 0.598 | 0.906 | 0.905 |
| MLP2 | 0.523 | 0.68 | 0.877 | 0.877 | 0.523 | 0.674 | 0.881 | 0.879 |
| MLP3 | 0.908 | 1.124 | 0.664 | 0.664 | 0.992 | 1.181 | 0.698 | 0.626 |
| MLP4 | 1.554 | 1.919 | 0.022 | 0.022 | 1.564 | 1.931 | 0.01 | 0.008 |
| MLP5 | 1.541 | 1.883 | 0.058 | 0.058 | 1.578 | 1.929 | 0.056 | 0.01 |
| MLP6 | 0.334 | 0.65 | 0.894 | 0.894 | 0.266 | 0.355 | 0.966 | 0.962 |
| MLP7 | 0.333 | 0.446 | 0.947 | 0.947 | 0.279 | 0.348 | 0.968 | 0.966 |
| MLP8 | 0.247 | 0.326 | 0.972 | 0.972 | 0.318 | 0.405 | 0.978 | 0.952 |
| **MLP9** | **0.244** | **0.319** | **0.973** | **0.973** | **0.263** | **0.340** | **0.977** | **0.969** |
| SS | 0.35 | 0.487 | 0.938 | 0.938 | 0.291 | 0.388 | 0.96 | 0.96 |
| MLR | 0.32 | 0.427 | 0.952 | 0.952 | 0.395 | 0.486 | 0.942 | 0.94 |

Table 11. Comparisons of different models for predicting *Ep* at HK station.

| | Training | | | | Testing | | | |
|---|---|---|---|---|---|---|---|---|
| HK | MAE | RMSE | $R^2$ | E | MAE | RMSE | $R^2$ | E |
| ANFIS-GP1 | 0.528 | 0.688 | 0.814 | 0.814 | 0.669 | 0.800 | 0.854 | 0.727 |
| ANFIS-GP2 | 0.741 | 0.964 | 0.634 | 0.634 | 0.802 | 0.970 | 0.742 | 0.599 |
| ANFIS-GP3 | 0.619 | 0.798 | 0.749 | 0.749 | 0.482 | 0.610 | 0.851 | 0.841 |
| ANFIS-GP4 | 1.23 | 1.451 | 0.171 | 0.171 | 1.200 | 1.490 | 0.268 | 0.054 |
| ANFIS-GP5 | 1.305 | 1.594 | 0.01 | 0.01 | 1.333 | 1.585 | 0.030 | -0.072 |
| ANFIS-GP6 | 0.488 | 0.646 | 0.836 | 0.836 | 0.660 | 0.796 | 0.861 | 0.73 |
| ANFIS-GP7 | 0.46 | 0.597 | 0.86 | 0.86 | 0.494 | 0.609 | 0.891 | 0.842 |
| ANFIS-GP8 | 0.388 | 0.501 | 0.901 | 0.901 | 0.809 | 0.930 | 0.919 | 0.631 |
| **ANFIS-GP9** | **0.286** | **0.379** | **0.943** | **0.943** | **0.428** | **0.555** | **0.925** | **0.869** |
| FG1 | 0.506 | 0.661 | 0.828 | 0.828 | 0.662 | 0.792 | 0.858 | 0.732 |
| FG2 | 0.716 | 0.914 | 0.671 | 0.671 | 0.793 | 0.940 | 0.784 | 0.623 |
| FG3 | 0.612 | 0.768 | 0.768 | 0.768 | 0.503 | 0.630 | 0.850 | 0.831 |
| FG4 | 1.103 | 1.333 | 0.30 | 0.300 | 1.169 | 1.480 | 0.306 | 0.077 |
| FG5 | 1.234 | 1.531 | 0.08 | 0.080 | 1.430 | 1.739 | 0.030 | -0.251 |
| FG6 | 0.471 | 0.626 | 0.846 | 0.846 | 0.659 | 0.786 | 0.875 | 0.738 |
| FG7 | 0.451 | 0.591 | 0.863 | 0.863 | 0.485 | 0.596 | 0.895 | 0.843 |
| FG8 | 0.39 | 0.496 | 0.903 | 0.903 | 0.718 | 0.849 | 0.920 | 0.688 |
| **FG9** | **0.311** | **0.397** | **0.938** | **0.938** | **0.414** | **0.552** | **0.932** | **0.870** |
| GRNN1 | 0.505 | 0.666 | 0.829 | 0.829 | 0.673 | 0.810 | 0.854 | 0.719 |
| GRNN2 | 0.699 | 0.902 | 0.681 | 0.680 | 0.786 | 0.929 | 0.776 | 0.63 |
| GRNN3 | 0.6 | 0.759 | 0.775 | 0.773 | 0.511 | 0.642 | 0.845 | 0.825 |
| GRNN4 | 1.11 | 1.335 | 0.311 | 0.299 | 1.178 | 1.472 | 0.316 | 0.077 |
| GRNN5 | 1.295 | 1.581 | 0.025 | 0.016 | 1.342 | 1.600 | 0.008 | -0.09 |
| GRNN6 | 0.452 | 0.605 | 0.859 | 0.859 | 0.65 | 0.771 | 0.879 | 0.734 |
| GRNN7 | 0.405 | 0.535 | 0.889 | 0.889 | 0.484 | 0.589 | 0.892 | 0.842 |
| GRNN8 | 0.408 | 0.538 | 0.894 | 0.894 | 0.539 | 0.651 | 0.916 | 0.853 |
| **GRNN9** | **0.241** | **0.342** | **0.956** | **0.956** | **0.415** | **0.512** | **0.917** | **0.881** |
| LSSVM1 | 0.502 | 0.658 | 0.829 | 0.829 | 0.671 | 0.802 | 0.858 | 0.726 |
| LSSVM2 | 0.70 | 0.895 | 0.685 | 0.685 | 0.822 | 0.971 | 0.793 | 0.598 |

| | | | | | | | | |
|---|---|---|---|---|---|---|---|---|
| LSSVM3 | 0.649 | 0.828 | 0.766 | 0.73 | 0.578 | 0.714 | 0.852 | 0.782 |
| LSSVM4 | 1.10 | 1.333 | 0.301 | 0.301 | 1.151 | 1.469 | 0.311 | 0.08 |
| LSSVM5 | 1.296 | 1.588 | 0.009 | 0.007 | 1.349 | 1.611 | 0.007 | -0.107 |
| LSSVM6 | 0.481 | 0.64 | 0.841 | 0.839 | 0.658 | 0.787 | 0.868 | 0.736 |
| LSSVM7 | 0.463 | 0.609 | 0.856 | 0.854 | 0.491 | 0.603 | 0.891 | 0.845 |
| LSSVM8 | 0.441 | 0.571 | 0.880 | 0.872 | 0.552 | 0.664 | 0.917 | 0.812 |
| **LSSVM9** | **0.326** | **0.425** | **0.930** | **0.929** | **0.398** | **0.501** | **0.925** | **0.893** |
| MARS1 | 0.506 | 0.661 | 0.828 | 0.828 | 0.662 | 0.791 | 0.860 | 0.733 |
| MARS2 | 0.664 | 0.862 | 0.708 | 0.708 | 0.858 | 1.023 | 0.766 | 0.554 |
| MARS3 | 0.603 | 0.758 | 0.774 | 0.774 | 0.5 | 0.638 | 0.842 | 0.826 |
| MARS4 | 1.084 | 1.315 | 0.319 | 0.319 | 1.185 | 1.500 | 0.243 | 0.041 |
| MARS5 | 1.266 | 1.56 | 0.043 | 0.043 | 1.364 | 1.634 | 0.001 | -0.138 |
| MARS6 | 0.438 | 0.581 | 0.867 | 0.867 | 0.733 | 0.899 | 0.869 | 0.655 |
| MARS7 | 0.426 | 0.547 | 0.882 | 0.882 | 0.536 | 0.691 | 0.891 | 0.797 |
| MARS8 | 0.407 | 0.517 | 0.895 | 0.895 | 0.682 | 0.807 | 0.917 | 0.722 |
| **MARS9** | **0.322** | **0.414** | **0.932** | **0.932** | **0.397** | **0.515** | **0.927** | **0.887** |
| MLP1 | 0.512 | 0.671 | 0.823 | 0.823 | 0.657 | 0.793 | 0.855 | 0.732 |
| MLP2 | 0.686 | 0.878 | 0.697 | 0.697 | 0.822 | 0.979 | 0.792 | 0.591 |
| MLP3 | 0.707 | 0.903 | 0.679 | 0.679 | 0.821 | 0.973 | 0.79 | 0.626 |
| MLP4 | 1.073 | 1.309 | 0.325 | 0.325 | 1.137 | 1.459 | 0.293 | 0.092 |
| MLP5 | 1.306 | 1.591 | 0.005 | 0.005 | 1.329 | 1.576 | 0.028 | -0.058 |
| MLP6 | 0.47 | 0.623 | 0.847 | 0.847 | 0.657 | 0.779 | 0.878 | 0.741 |
| MLP7 | 0.421 | 0.542 | 0.884 | 0.884 | 0.485 | 0.594 | 0.897 | 0.847 |
| MLP8 | 0.431 | 0.554 | 0.88 | 0.879 | 0.671 | 0.786 | 0.916 | 0.736 |
| **MLP9** | **0.34** | **0.444** | **0.923** | **0.923** | **0.386** | **0.491** | **0.930** | **0.897** |
| SS | 0.523 | 0.683 | 0.827 | 0.827 | 0.64 | 0.773 | 0.823 | 0.822 |
| MLR | 0.328 | 0.431 | 0.927 | 0.927 | 0.396 | 0.505 | 0.927 | 0.925 |

Table 12. Accuracy ranks[*] of the soft computing models in estimating *Ep*.

| Stations | ANFIS-GP | FG | GRNN | LSSV | MARS | MLP | MLR |
|----------|----------|-----|------|------|------|-----|-----|
| HEB | 2 | 3 | 4 | 6 | 5 | 1 | 7 |
| ALT | 3 | 4 | 6 | 2 | 5 | 1 | 7 |
| MQ | 3 | 2 | 4 | 7 | 5 | 1 | 6 |
| BJ | 4 | 5 | 3 | 2 | 6 | 1 | 7 |
| LSA | 6 | 5 | 1 | 3 | 4 | 2 | 7 |
| CQ | 6 | 3 | 2 | 4 | 5 | 1 | 7 |
| HZ | 6 | 4 | 2 | 5 | 3 | 1 | 7 |
| HK | 6 | 7 | 3 | 5 | 4 | 1 | 2 |
| ALL | 4 | 5 | 2 | 3 | 6 | 1 | 7 |
| **Total** | **40** | **38** | **27** | **37** | **43** | **10** | **57** |

[*]Accuracy ranks were determined according to the RMSE, MAE, E and $R^2$ criteria. For the HEB, for example, MLP has the highest accuracy (1st model) while the MLR has the lowest accuracy (7th model).

Table 13. Comparisons of different models for predicting *Ep* at all stations.

| | Training | | | | Testing | | | |
|---|---|---|---|---|---|---|---|---|
| | MAE | RMSE | R$^2$ | *E* | MAE | RMSE | R$^2$ | *E* |
| ANFIS-GP1 | 1.204 | 1.681 | 0.739 | 0.739 | 1.022 | 1.378 | 0.804 | 0.803 |
| ANFIS-GP2 | 1.906 | 2.522 | 0.412 | 0.412 | 1.768 | 2.345 | 0.437 | 0.431 |
| ANFIS-GP3 | 1.913 | 2.377 | 0.478 | 0.478 | 1.877 | 2.262 | 0.475 | 0.471 |
| ANFIS-GP4 | 1.23 | 1.451 | 0.171 | 0.171 | 1.20 | 1.490 | 0.268 | 0.054 |
| ANFIS-GP5 | 1.305 | 1.594 | 0.001 | 0.001 | 1.333 | 1.585 | 0.029 | -0.072 |
| ANFIS-GP6 | 0.994 | 1.446 | 0.807 | 0.807 | 0.88 | 1.228 | 0.847 | 0.844 |
| ANFIS-GP7 | 0.917 | 1.341 | 0.834 | 0.834 | 0.782 | 1.113 | 0.872 | 0.872 |
| ANFIS-GP8 | 0.606 | 0.846 | 0.934 | 0.934 | 0.601 | 0.833 | 0.933 | 0.928 |
| ANFIS-GP9 | 0.517 | 0.738 | 0.95 | 0.95 | 0.486 | 0.666 | 0.957 | 0.957 |
| FG1 | 1.208 | 1.676 | 0.74 | 0.74 | 1.028 | 1.377 | 0.805 | 0.804 |
| FG2 | 1.883 | 2.511 | 0.417 | 0.417 | 1.741 | 2.332 | 0.443 | 0.438 |
| FG3 | 1.8 | 2.221 | 0.544 | 0.544 | 1.812 | 2.148 | 0.524 | 0.521 |
| FG4 | 1.106 | 1.336 | 0.298 | 0.298 | 1.151 | 1.469 | 0.318 | 0.075 |
| FG5 | 1.288 | 1.567 | 0.034 | 0.034 | 1.326 | 1.565 | 0.009 | -0.044 |
| FG6 | 0.936 | 1.378 | 0.824 | 0.824 | 0.821 | 1.154 | 0.865 | 0.862 |
| FG7 | 0.883 | 1.294 | 0.845 | 0.845 | 0.753 | 1.072 | 0.882 | 0.880 |
| FG8 | 0.589 | 0.834 | 0.936 | 0.936 | 0.607 | 0.842 | 0.931 | 0.929 |
| FG9 | 0.518 | 0.744 | 0.949 | 0.949 | 0.495 | 0.678 | 0.956 | 0.954 |
| GRNN1 | 1.193 | 1.669 | 0.743 | 0.743 | 1.013 | 1.373 | 0.806 | 0.805 |
| GRNN2 | 1.859 | 2.49 | 0.427 | 0.427 | 1.716 | 2.311 | 0.453 | 0.448 |
| GRNN3 | 1.772 | 2.216 | 0.549 | 0.546 | 1.773 | 2.127 | 0.532 | 0.532 |
| GRNN4 | 1.11 | 1.335 | 0.311 | 0.299 | 1.178 | 1.472 | 0.316 | 0.077 |
| GRNN5 | 1.295 | 1.581 | 0.025 | 0.016 | 1.342 | 1.600 | 0.02 | -0.091 |
| GRNN6 | 0.819 | 1.234 | 0.86 | 0.859 | 0.733 | 1.075 | 0.884 | 0.880 |
| GRNN7 | 0.724 | 1.114 | 0.886 | 0.885 | 0.642 | 0.963 | 0.905 | 0.904 |
| GRNN8 | 0.458 | 0.674 | 0.958 | 0.958 | 0.489 | 0.723 | 0.947 | 0.946 |
| GRNN9 | 0.265 | 0.425 | 0.984 | 0.983 | 0.364 | 0.573 | 0.967 | 0.966 |
| LSSVM1 | 1.198 | 1.667 | 0.743 | 0.743 | 1.017 | 1.371 | 0.807 | 0.806 |
| LSSVM2 | 1.85 | 2.495 | 0.425 | 0.425 | 1.703 | 2.312 | 0.453 | 0.447 |
| LSSVM3 | 1.854 | 2.314 | 0.506 | 0.505 | 1.858 | 2.215 | 0.493 | 0.492 |
| LSSVM4 | 1.10 | 1.333 | 0.301 | 0.301 | 1.151 | 1.469 | 0.311 | 0.08 |
| LSSVM5 | 1.296 | 1.588 | 0.009 | 0.007 | 1.349 | 1.611 | 0.007 | -0.107 |
| LSSVM6 | 0.935 | 1.386 | 0.823 | 0.822 | 0.806 | 1.149 | 0.866 | 0.864 |
| LSSVM7 | 0.933 | 1.369 | 0.827 | 0.827 | 0.8 | 1.134 | 0.867 | 0.867 |
| LSSVM8 | 0.824 | 1.148 | 0.879 | 0.878 | 0.774 | 1.023 | 0.893 | 0.892 |
| LSSVM9 | 0.494 | 0.719 | 0.952 | 0.952 | 0.476 | 0.657 | 0.958 | 0.955 |
| MARS1 | 1.198 | 1.666 | 0.744 | 0.744 | 1.021 | 1.373 | 0.806 | 0.805 |
| MARS2 | 1.793 | 2.428 | 0.455 | 0.455 | 1.676 | 2.268 | 0.476 | 0.468 |
| MARS3 | 1.772 | 2.206 | 0.55 | 0.55 | 1.77 | 2.125 | 0.534 | 0.533 |
| MARS4 | 1.084 | 1.315 | 0.319 | 0.319 | 1.185 | 1.500 | 0.243 | 0.040 |
| MARS5 | 1.268 | 1.561 | 0.04 | 0.04 | 1.386 | 1.677 | 0.012 | -0.199 |

| | MAE | RMSE | R$^2$ | E | MAE | RMSE | R$^2$ | E |
|---|---|---|---|---|---|---|---|---|
| MARS6 | 1.025 | 1.439 | 0.808 | 0.808 | 0.924 | 1.232 | 0.846 | 0.843 |
| MARS7 | 0.925 | 1.324 | 0.838 | 0.838 | 0.804 | 1.113 | 0.873 | 0.872 |
| MARS8 | 0.782 | 1.03 | 0.902 | 0.902 | 0.754 | 0.956 | 0.910 | 0.906 |
| **MARS9** | **0.692** | **0.933** | **0.920** | **0.920** | **0.654** | **0.829** | **0.932** | **0.929** |
| MLP1 | 1.196 | 1.663 | 0.744 | 0.744 | 1.020 | 1.373 | 0.806 | 0.805 |
| MLP2 | 1.835 | 2.485 | 0.429 | 0.429 | 1.689 | 2.304 | 0.457 | 0.451 |
| MLP3 | 1.842 | 2.491 | 0.426 | 0.426 | 1.695 | 2.302 | 0.458 | 0.452 |
| MLP4 | 1.073 | 1.309 | 0.325 | 0.325 | 1.137 | 1.459 | 0.293 | 0.090 |
| MLP5 | 1.306 | 1.591 | 0.005 | 0.005 | 1.329 | 1.576 | 0.03 | -0.058 |
| MLP6 | 0.836 | 1.256 | 0.854 | 0.854 | 0.740 | 1.086 | 0.882 | 0.878 |
| MLP7 | 0.774 | 1.181 | 0.871 | 0.871 | 0.649 | 0.980 | 0.902 | 0.901 |
| MLP8 | 0.529 | 0.758 | 0.947 | 0.947 | 0.531 | 0.770 | 0.943 | 0.939 |
| **MLP9** | **0.279** | **0.398** | **0.985** | **0.985** | **0.314** | **0.405** | **0.988** | **0.981** |
| SS | 1.107 | 1.544 | 0.785 | 0.785 | 1.007 | 1.336 | 0.823 | 0.822 |
| MLR | 0.905 | 1.235 | 0.859 | 0.859 | 0.860 | 1.091 | 0.880 | 0.880 |

Table 14. Evaluation of the optimal models by training with testing dataset and testing with training dataset

| | Training | | | | Testing | | | |
|---|---|---|---|---|---|---|---|---|
| | MAE | RMSE | R$^2$ | E | MAE | RMSE | R$^2$ | E |
| ANFIS-GP9 | 0.447 | 0.631 | 0.959 | 0.959 | 0.526 | 0.792 | 0.945 | 0.942 |
| FG9 | 0.476 | 0.686 | 0.951 | 0.951 | 0.591 | 0.864 | 0.934 | 0.931 |
| GRNN9 | 0.230 | 0.331 | 0.989 | 0.989 | 0.493 | 0.820 | 0.941 | 0.927 |
| LSSVM9 | 0.703 | 0.962 | 0.907 | 0.907 | 0.784 | 1.170 | 0.882 | 0.826 |
| MARS9 | 0.600 | 0.783 | 0.937 | 0.937 | 0.691 | 0.968 | 0.916 | 0.913 |
| **MLP9** | **0.383** | **0.537** | **0.970** | **0.970** | **0.481** | **0.735** | **0.953** | **0.950** |

Table 15. The MLP model performances tested at different stations with full weather inputs

| | Training | | | | Testing | | | |
|---|---|---|---|---|---|---|---|---|
| | MAE | RMSE | R$^2$ | E | MAE | RMSE | R$^2$ | E |
| HEB | 0.150 | 0.197 | 0.996 | 0.996 | 0.498 | 0.687 | 0.970 | 0.956 |
| ALT | 0.193 | 0.255 | 0.994 | 0.994 | 0.524 | 0.831 | 0.980 | 0.957 |
| MQ | 0.446 | 0.542 | 0.984 | 0.983 | 0.693 | 0.908 | 0.974 | 0.960 |
| BJ | 0.187 | 0.241 | 0.992 | 0.992 | 0.468 | 0.813 | 0.930 | 0.921 |
| LSA | 0.270 | 0.342 | 0.976 | 0.976 | 0.636 | 0.788 | 0.880 | 0.876 |
| CQ | 0.114 | 0.159 | 0.992 | 0.992 | 0.317 | 0.740 | 0.865 | 0.862 |
| HZ | 0.208 | 0.271 | 0.981 | 0.981 | 0.403 | 0.535 | 0.937 | 0.924 |
| HK | 0.317 | 0.393 | 0.934 | 0.934 | 0.414 | 0.548 | 0.922 | 0.882 |

---

## Author Comment (AC2) · 8 Aug 2016

Weakness

1. Language: needs substantial editing efforts to ensure consistency and readability via improving e.g., wording, sentence structure, paragraph connection and cohesion.

Reply: The language of the whole manuscript has been carefully revised and improved, thank you.

2. Methodology: Training dataset is select randomly, however without ensemble the randomness of the data selection is still weak. Cross-site validation is also necessary before concluding which model is the "best" one.

Reply: we further evaluated the applied models with full weather inputs by changing

training and testing period. Please see Table 14 for the results.

Major comments 1. Models used 50% of data for training and the rest for testing (randomly chosen) at each study site. However, random sampling is not repeated to ensure generality of the model testing results. With only one random sampling of training dataset, the trained model is potentially biased to that particular case. What will happen, if use the testing data for training and training data for testing? Will the model predictability retain? In order to show the generality of the model performance, one has to do multiple ensembles of training data sampling and present the "mean model".

Reply: as suggested models with full weather inputs were evaluated by changing training and testing period. Please see Table 14 for the results.

2. This study made lots of efforts on comparing different models and trying to find out the "best" one. The identification of the "best" model is based on within-site evaluation. In that sense, the significance of this study is highly limited. What happen if one would like to apply the "best" model to other sites with different climate, or do a regional modeling? This type of question could be answered by doing cross-site evaluation. For example, apply the trained model to other 7 sites and show the overall performance.

Reply: as suggested, the generalized MLP9 model was also tested at each site.

3. Results interpretation: Table, figures, and main text are heavily redundant, e.g., no need to repeat all the numbers (e.g., R2) in main text that have been included in table.

Reply: results section has been revised.

Specific comments

L24. The first sentence does not make too much sense, because this study focused on pan evaporation, which best inform water management such as agriculture. But for terrestrial ecological processes and regional climate change, evaporation is not as significant as it in agriculture. I would suggest the abstract starting with "Pan evaporation
Interactive
comment

plays . . . in informing . . . ". And followed with another sentence highlighting the fact that one of the basic challenges is modeling . . ... L30. First time use Ep, better to define it earlier. L31. No need to list all the climate variables. Maybe: "We develop, train, and validate the eight models at various sites crossing a wide range of climate". L33. The first part of applications focused . . .. Remove this sentence, because the next sentence is actually presenting the accuracy comparison. L38. Generalized models were also developed and tested. . ... Remove this sentence. L42. BJ, CQ and HK station. Define the sites before use them. L42. Recommendation or major implication based on this study is needed to end the abstract. L48 "and air" -> "and air temperature"? L50, roles in . . . -> roles in informing water resources redistribution and irrigation system design. L55 is -> are L56 one of the, remove; aspects in the hydrological cycle, remove L57 to integrated -> for integrating L59 some, remove L63 for estimating Ep as a function of meteorological data -> to linking Ep to various meteorological drivers. L64. But some of . . . -> but the applications of these techniques are often limited by data availability and completeness. L68. What are conventional techniques, list a few. L75 at a site in hot and dry climate -> at a hot and dry site L76 is -> was L70 for example -> .For example, . . .and from L70 – L77 replace ";" with "." L110 provided an impetus -> impede L111. Which provided an impetus for . . ., remove. L112 "considering the importance of . . .or hydrological modeling", remove. L116 in modeling Ep. . . -> in Ep modeling with different combinations of climate inputs. L119 "using generalized . . . models" -> using eight different models. L129. MLPs are organized as hierarchical networks with several layers L133. its input -> input L142. but they do not use-> without using L143 The structure of, remove L148 Two types of neurons (S-summation and D-summation) are connected to patter layer unit L179 which is -> that is L183 which can be used for optimization problems, remove L193 which projects -> that projects L196 efficient is enough, redundant to say quick, converging to global optimum. L199 more simpler and more efficient L200 This issue is caused by-> , due to the reason that LSSVM solves linear equations instead of a quadratic programming problem in SVM. L204. This subject -> these models L212 consists ->conssiting

Reply: All the corrections above have been made carefully.

L236 Why use monthly data? e.g., Goyal 2014 investigated various techniques to improve daily Ep in India. Is it possible to do daily at the eight selected sites as well?

Reply: we studied with monthly time scale following the related literature (Kisi 2015, Guven and Kisi 2013, Citakoglu et al. 2014, Tezel and Buyukyildiz 2016, Kisi 2009, Campos-Aranda 2004, Fennessey 2000, Savenije 1997, Francisco and Aranda 1997, Kim 2011, Alvarez et al. 2007, Almedeij 2012) which also used pan evaporation in monthly time scale.

References:

Kisi, O. (2015) Pan evaporation modeling using least square support vector machine, multivariate adaptive regression splines and M5 model tree. Journal of Hydrology 528, 312-320.

Guven, A. and Kisi, O. (2013) Monthly pan evaporation modeling using linear genetic programming. Journal of Hydrology 503, 178-185.

Citakoglu, H., Cobaner, M., Haktanir, T. and Kisi, O. (2014) Estimation of Monthly Mean Reference Evapotranspiration in Turkey. Water Resources Management 28(1), 99-113.

Tezel, G. and Buyukyildiz, M. (2016) Monthly evaporation forecasting using artificial neural networks and support vector machines. Theoretical and Applied Climatology 124(1-2), 69-80.

Kisi, O. (2009) Modeling monthly evaporation using two different neural computing techniques. Irrigation Science 27(5), 417-430.

Campos-Aranda, D.F. (2004) Estimation of monthly evaporation in a class-A pan in the Mexican Republic through temperature data. IngenieriaHidraulica En Mexico 19(4), 85-96.

Fennessey, N.M. (2000) Estimating average monthly lake evaporation in the northeast

United States. Journal of the American Water Resources Association 36(4), 759-769.

Savenije, H.H.G. (1997) Determination of evaporation from a catchment water balance at a monthly time scale. Hydrology and Earth System Sciences 1(1), 93-100.

Francisco, D. and Aranda, C. (1997) Estimate of the monthly evaporation from a simulation of operations of the El Cuchillo-Marte R. Gomez Dam system. IngenieriaHidraulica En Mexico 12(2), 49-56.

Kim, S. (2011) Nonlinear Hydrologic Modeling Using the Stochastic and Neural Networks Approach. Disaster Advances 4(1), 53-63.

Alvarez, V.M., Gonzalez-Real, M.M., Baille, A. and Martinez, J.M.M. (2007) A novel approach for estimating the pan coefficient of irrigation water reservoirs application to South Eastern Spain. Agricultural Water Management 92(1-2), 29-40.

Almedeij, J. (2012) Modeling Pan Evaporation for Kuwait by Multiple Linear Regression. Scientific World Journal.

L238 – 242. Site information are redundant, with figure 3. Or could be just put in a table.

Reply: It was corrected (see Table 1).

L242-258. Put the site description (lon, lat, alt, mean annual temperature, mean annual precipitation) in a table. In the text, highlight the most important fact, do not literally reiterate the site information.

Reply: It was corrected (see Table 1).

L280-286. Again, too many numbers in this section (which have already been shown in Table 1). Just illustrate the most important fact, e.g., which site has the highest Ep, why and how? L305. ANFIS, GP, . . ., and SS. -> six soft computing and two regression models L314-317 This study . . . in the application. Remove. Just present results and discussion, no need to reiterate what have been done. L329. Models with full

weather data have the best accuracy. This seems obvious but indeed has significant implications. As using more and more predictor variables, the response variables could be commonly better predicted. However, the issue is the expenditure. In this case, is the data availability. Variables like air temperature is relatively easy to measure and important for evaporation, we definitely want to include it in ep modeling. However, is there a predictor variable that is relatively hard to measure (unavailable) but is "must be include" predictor variable? Is it necessary to use the full model for large-scale (regional) prediction? If so, is this conclusion also valid at other study sites?

Reply: model results for the 1st, 6th, 7th and 8th input combinations indicates that the soft computing models can be successfully used with local calibration (see Tables 4-11). Table 13 shows that the generalization of the soft computing models are possible in case of limited inputs. However, we have not investigated the accuracy of the generalized models with limited data for each station, separately because of the length of the paper. This may be done in another study.

L367. The best accuracy were generally obtained from five-input models and GNRR model perform better. It is excited to see that a certain model stand out. But it would be helpful, it one can go one step further and try to figure out the underlying reason why that model is "best". What kind of feature of that model could possibly lead to the success?

Reply: MLP generally performed better than the other methods in estimating Ep. However, the accuracy of the other methods is also satisfactorily well. The advantages of each method were included in the methods section of the revised version.

L414. It is obvious that . . .. Throughout the paper, many places used this sentence structure "it is obvious that". Try to avoid if necessary, because it imply that if one can not immediately understand the results then he is stupid. Furthermore, sometime the results are not that obvious and it is always authors' responsibility to help the readers understand those results.

Reply: It was corrected.

---

## Author Comment (AC4) · 8 Aug 2016

The paper estimates pan evaporation by statistical models calibrated at 8 stations in China and briefly discusses the choice of climatic variables to drive the models. The paper can potentially be an interesting contribution to the field, but in my opinion it requires a revision to make it more attractive to the readers. I concur with most of the comments expressed by the other two reviewers, so I will not list any specific issues. A few things worth stressing on my view are:

(1) Statistical models. MLPs are in many cases used by default as the "soft computing" technique to statistically approximate geophysical relationships, so it is not a surprise to me that they came as the winner. Other methods may compete with the MLPs, but in my experience they are one of the best compromises between implementation

complexity and capacity to approximate mappings. The problem is that the MLPs are also well known for over-fitting issues, and as there is no cross-correlation tests in the paper (i.e., calibration and error performance on different stations), we may wonder if over-fitting may be playing a role here (other methods may be more robust in this sense, but will be penalized by showing a worst performance on the validation dataset). This needs to be discussed.

Reply: we further evaluated the applied models with full weather inputs by changing training and testing period. Please see Table 14 for the results. A Discussion section was added into the revised version.

(2) Climate drivers. It may have been better to start by looking at the linear correlation between the climate drivers and the Ep at the different stations, to rank the relative importance of the drivers in a simple way. That could have been used to justify why RH and WS are only tested as part of the final combination of drivers to the models, and perhaps be used to reduce the number of driver-combinations to be tested. In my experience, the mapping between drivers and geophysical parameter has to be very non-linear for the linear correlations to differ significantly from a "correlation" inferred by applying first a non-linear estimator.

Reply: we have also added two input combinations as iv) RH and v) Ws. Now, the effect of each variable on Ep can be clearly seen. The statement "In overall, soft computing models with full weather data (Rg, Ta, Hs, RH and Ws) generally had the best accuracy. This indicates that all these variables are required for better Ep estimation. It can be seen from the applications that adding RH or Ws inputs into the applied models generally increase their accuracies in predicting Ep in all stations even though these parameters have the lowest correlation with Ep (see Table 1). This indicates the non-linear relationship between RH (Ws) and Ep and linear R2 indicator cannot show this phenomenon. Two-input (Rg and Ta) soft computing models generally provide better accuracy than the SS model in predicting Ep in all stations in the testing stage. These simple models can be preferred in some areas (e.g, developing countries) where obtaining the data of the other parameters (RH and Ws) are difficult." was added at the end of the Discussion section.

(3) Applications. My reading of the paper is that I better use MLPs to statistically mode Ep, but perhaps not for all climates. OK, but not sure whether that is a clear message to pass. I any case, I would suggest to go a bit further and use the constructed database to provide a more general model that it is not restricted to a specific climate type. This could have been tested by investigating cross-correlations (i.e., how a model trained in a station performs at a different station), and/or by calibrating the model with a database containing data from all stations. It is quite likely that the more general model cannot outperform the individual-station best model, but if the differences are reasonable that single-model could potentially be used to generate Ep over most China when driven with remote sensing data. To me, that would be an excellent outcome of the paper and of more utility than just showing that at a specific station one statistical model performs better than the others.

Reply: the best generalized model (MLP9) was also tested at each site. A Discussion section was added into the revised version.

(4) Format. The paper, even with its current contents, needs a better way to present the results. There are currently more than 3000 numbers scattered around 12 table-sand around 70 scatter plots, with only one table ranking the models summarizing the main results. That material could be part of an appendix, but figures and/or tables synthesizing the main results are needed.

Reply: all tables and figures were changed. We agree with you that there are lots of tables and figures. To the authors, however, all the tables and figures indicate main results. We could not select any table or figure for appendix.
* * *